# Flow based approach for Dynamic Temporal Causal models with non-Gaussian or Heteroscedastic Noises

**Abdellah Rahmani**
LTS4, EPFL
Lausanne, Switzerland
abdellah.rahmani@epfl.ch

**Pascal Frossard**
LTS4, EPFL
Lausanne, Switzerland
pascal.frossard@epfl.ch

## Abstract

Understanding causal relationships in multivariate time series is crucial in many scenarios, such as those dealing with financial or neurological data. Many such time series exhibit multiple regimes, i.e., consecutive temporal segments with a priori unknown boundaries, with each regime having its own causal structure. Inferring causal dependencies and regime shifts is critical for analyzing the underlying processes. However, causal structure learning in this setting is challenging due to (1) non-stationarity, i.e., each regime can have its own causal graph and mixing function, and (2) complex noise distributions, which may be non-Gaussian or heteroscedastic. Existing causal discovery approaches cannot address these challenges, since generally assume stationarity or Gaussian noise with constant variance. Hence, we introduce FANTOM, a unified framework for causal discovery that handles non-stationary processes along with non-Gaussian and heteroscedastic noises. FANTOM simultaneously infers the number of regimes and their corresponding indices and learns each regime's Directed Acyclic Graph. It uses a Bayesian Expectation Maximization algorithm that maximizes the evidence lower bound of the data log-likelihood. On the theoretical side, we prove, under mild assumptions, that temporal heteroscedastic causal models, introduced in FANTOM's formulation, are identifiable in both stationary and non-stationary settings. In addition, extensive experiments on synthetic and real data show that FANTOM outperforms existing methods.

## 1 Introduction

Causal structure learning from multivariate time series (MTS) is a fundamental problem with diverse applications in traffic modeling [9], biology [44], climate science [43], or healthcare [47]. However, identifying causal relationships in MTS poses several challenges. First, real-world time series are often non-stationary, exhibiting multiple unknown regimes, each potentially governed by different causal relationships. Examples include changing dependencies across climate conditions [27], financial markets [21], and epileptic seizure stages [56]. Second, many MTS display complex noises, e.g., non-Gaussian or even heteroscedastic noise, whose variance depends on both instantaneous and lagged causes. This occurs in fMRI data [46], EEG measurements [22], or financial data [18].

Recent causal discovery methods capture linear and nonlinear interactions with instantaneous and lagged effects [41, 35, 51]. More recently, Gong et al. [15] and Wang et al. [55] explored structural equation models for a single stationary regime governed by a one causal graph with historically dependent noise, where noise variance depends solely on time-lagged variables, neglecting heteroscedasticity. Existing multi-regime methods include RPCMCI [45], which identifies only linear, time-lagged interactions and requires prior knowledge of regime numbers and transitions; and CD-NOD [21], which handles causal discovery from non-stationary MTS, but is limited to homoscedastic noise, cannot infer individual causal graph for each regime, and is incapable of

39th Conference on Neural Information Processing Systems (NeurIPS 2025).

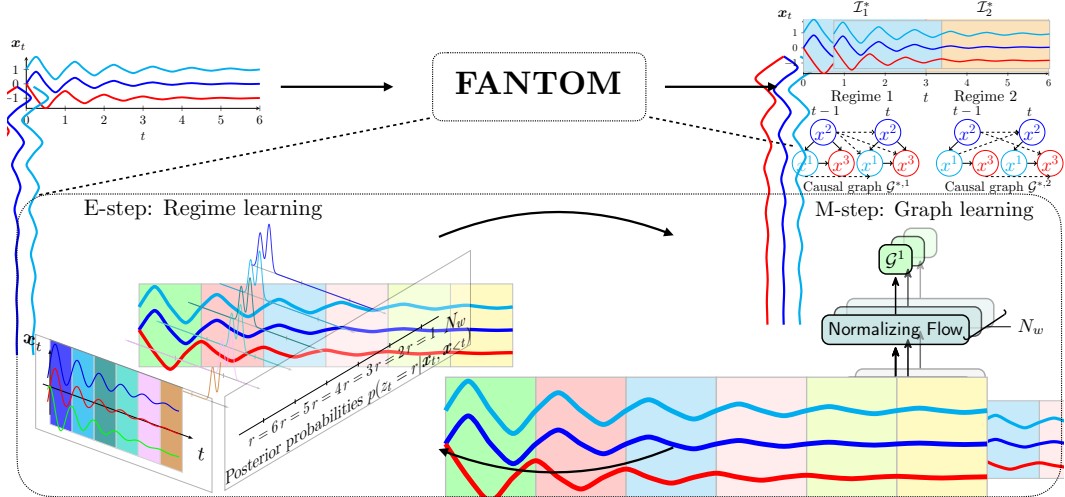

Figure 1: Illustration of FANTOM processing a MTS with two ground truth regimes ($K = 2$). The algorithm recovers the regime indices $\mathcal{I}_1^*$ and $\mathcal{I}_2^*$ and learns a temporal causal graph for each regime (dashed edges represent time-lagged links; solid arrows indicate instantaneous links). In the E-step, posterior probabilities $p\big(z_t = r \mid \boldsymbol{x}_t, \boldsymbol{x}_{<t}\big)$ are estimated, where $z_t = r$ means $\boldsymbol{x}_t$ belongs to regime $r$. The M-step then infers causal graphs within each regime. Here, $N_w$ is the number of regimes that converges to $K = 2$.

identifying recurring regimes. CASTOR [39], infers both regime indices and separate causal graphs per regime, accommodating instantaneous and lagged causal relationships, yet it is still restricted to normal noise. Consequently, the previous methods cannot jointly infer the number of regimes, their indices, their causal graphs, nor effectively manage either non-Gaussian or heteroscedastic noises.

To address these limitations, we propose FANTOM, a new framework for Structural Equation Models (SEMs) in multi-regime MTS under either non-Gaussian or heteroscedastic noises. FANTOM is, to the best of our knowledge, the first method to handle heteroscedasticity in both stationary and non-stationary MTS, as well as non-Gaussianity in non-stationary settings. Given a MTS with multiple regimes, FANTOM simultaneously learns each regime's causal graph and mixing function, determines the number of regimes, and infers their indices (Figure 1). It uses a Bayesian Expectation Maximization (BEM) [12] procedure to optimize the evidence lower bound (ELBO), alternatively assigning regime indices (Expectation step) and inferring causal relationships in each regime (Maximization step). Unlike Gaussian-based approaches, FANTOM employs conditional normalizing flows [12] to handle complex distributions and compute regime membership probabilities. It uses Bayesian structure learning that averages across all plausible graphs and naturally filters out spurious edges. Under mild assumptions, we prove that temporal heteroscedastic causal models are identifiable for both stationary and multi-regime MTS. Across extensive comparisons with existing multi-regime causal discovery methods, we show that FANTOM consistently achieves superior performance in structure learning and regime detection. Moreover, it outperforms stationary models, even when they are provided with ground-truth regime partitions, on synthetic and two real-world datasets. The main contributions of this paper can be summarized as follows:

- We introduce FANTOM, a unified framework for causal discovery in multi-regime MTS that handles both homoscedastic non-Gaussian and heteroscedastic noises while simultaneously discovering the number of regimes, their indices, and their corresponding causal graphs.

- Under mild assumptions in causal discovery, we prove identifiability of the temporal heteroscedastic causal models in the stationary case and show that the number of regimes, their indices, and their graphs are identifiable (up to permutation) in the non-stationary setting.

- We demonstrate, via extensive experiments, that FANTOM outperforms state-of-the-art methods on both synthetic and real-world datasets.

**Related work.** Many works tackle *causal structure learning from stationary MTS*, Granger causality is the primary approach used for this purpose [32, 8]. However, it is unable to accommodate

instantaneous effects. DYNOTEARS [35], learns instantaneous and time lagged structures and leverages the acyclicity constraint, introduced by Zheng et al. in [60], to turn the DAG learning problem to a purely continuous optimization problem. However, DYNOTEARS is limited to linear SEMs. Runge et al. [42] proposed a two-stage algorithm PCMCI+ that can scale to large time series, PCMCI+ is able also to handle non linear relationships. Neverthless, DYNOTEARS and PCMCI+ are restricted to homoscedastic noises where variance is a constant over time. For this reason, Rhino [15] and SCOTCH [55] introduced models that tackle stationary MTS with historical noise, where the noise variance is a function of solely time lagged parents. However Rhino, DYNOTEARS and PCMCI+, cannot handle heteroscedastic noise and are limited to stationary MTS.

Several studies have sought to tackle the challenge of *causal discovery in non-stationary MTS* [21, 17, 45, 33]. Remarkably, Huang et al. [21] address the setting of time series composed of different regimes by modulating causal relationships through a regime index. CD-NOD detects change points and outputs a single summary causal graph, but it overlooks recurring regimes and provides neither regime-specific graphs nor their count. RPCMCI [45] provides a graph for each regime, yet it assumes prior knowledge of the number of regimes, restricts edges to time-lagged links, and offers no identifiability guarantees. Balsells-Rodas et al. [5] establish identifiability for first-order, regime-dependent causal discovery in multi-regime MTS with Gaussian noise and offer a practical algorithm, but their framework allows only a single time-lagged edge and excludes instantaneous links. Finally, CASTOR [39] jointly infers regime labels and their causal graphs, capturing both instantaneous and lagged links, under an equivariant Gaussian noise assumption. However, none of these models can simultaneously learn the number of regimes, their indices, and their structures under non-Gaussian noise, nor do they offer identifiability guarantees. FANTOM fills this gap and even generalises to richer heteroscedastic noise settings while providing identifiability results. More detailed related work is provided in Appendix A).

## 2  Problem formulation

In this Section, we introduce our notation, define a temporal causal graph, and then we present a new Structural Equation Models (SEMs) for multi-regime MTS with non-Gaussian/Heteroscedastic noise.

**Notation.** Matrices, vectors, and scalars are denoted by uppercase bold $\boldsymbol{G}_\tau$, lowercase bold $\boldsymbol{x}_t$ and lowercase letters $x_{t-\tau}^i$, respectively. Ground-truth variables are indicated with an asterisk, such as $\mathcal{G}^*$. We assume that all distributions have densities $p(\boldsymbol{x}_t)$ w.r.t. the Lebesgue measure. The notation $[[0:L]]$ represents the set of integers $\{0,...,L\}$ and $|\cdot|$ denotes set cardinality. The notation $(\boldsymbol{x}_t)_{t\in\mathcal{T}} = (x_t^i)_{i\in\mathbf{V},t\in\mathcal{T}}$ represents a MTS of $|\mathbf{V}| = d$ components and length $|\mathcal{T}|$, $\mathcal{T}$ is the time index set and $\boldsymbol{x}_{<t}$ refers to $\{\boldsymbol{x}_{t-L},...,\boldsymbol{x}_{t-1}\}$.

**Definition 2.1** (Temporal Causal Graph [39])**.** The temporal causal graph, associated with the MTS $(\boldsymbol{x}_t)_{t\in\mathcal{T}}$, is defined by a Directed Acyclic Graph (DAG) $\mathcal{G} = (\mathbf{V}, E)$, represented by a collection of adjacency matrices $\boldsymbol{G}_{\tau\in[[0:L]]} = \{\boldsymbol{G}_0,\ldots,\boldsymbol{G}_L\}$, and a fixed maximum lag $L$. Its vertices $\mathbf{V}$ consist of the set of components $x_{t'}^1,\ldots,x_{t'}^d$ for each $t' \in [[t-L:t]]$. The edges $E$ of the graph are defined as follows: $\forall \tau \in [[1:L]]$ the variables $x_{t-\tau}^i$ and $x_t^j$ are connected by a lag-specific directed link $x_{t-\tau}^i \to x_t^j$ in $\mathcal{G}$ pointing forward in time if and only if $x^i$ at time $t-\tau$ causes $x^j$ at time $t$. Then the coefficient $[G_\tau]_{ij}$ associated with the adjacency matrix $\boldsymbol{G}_\tau \in \mathcal{M}_d(\mathbb{R})$ will be non-zero and $x^i \in \mathbf{Pa}_\mathcal{G}^j(<t)$; the lagged parents of node $i$ in $\mathcal{G}$. For instantaneous links ($\tau = 0$), we cannot have self loops i.e. $i \neq j$. If $\tau = 0$, we have an edge $x_t^i \to x_t^j$ and $x^i \in \mathbf{Pa}_\mathcal{G}^j(t)$; the instantaneous parents of $j$ at the current time $t$, if and only if $x^i$ at time $t$ causes $x^j$ at time $t$.

In many real world scenarios, a non-stationary MTS $(\boldsymbol{x}_t)_{t\in\mathcal{T}}$ may exhibit $K$ distinct, non-overlapping regimes, where non-stationarity is not modeled by continuous changes but rather by sequences of piecewise-constant regimes, as in climate science [27], finance [18], or epileptic recordings [56]. Every regime $r$ is a stationary MTS block, has its own temporal causal graph $\mathcal{G}^r$ (Definition 2.1). At each time $t = 1, 2, \ldots, |\mathcal{T}|$ there is a discrete latent state $z_t \in \{1, 2, \ldots, K\}$ that models the regime partition, i.e., $z_t = r$ means that $\boldsymbol{x}_t$ belongs to regime $r$ and we denote $\mathcal{I}_r = \{t|z_t = r, \forall t \in \mathcal{T}\}$ the set of all time indices at which regime $r$ appears. We gather these sets into $\mathcal{I} = (\mathcal{I}_r)_{r\in[[1:K]]}$, yielding a unique time partition of the MTS $(\boldsymbol{x}_t)_{t\in\mathcal{T}}$ composed of $K$ regimes. In addition, the observation $\boldsymbol{x}_t$ follows, a novel and general SEM that takes into account non stationarity, and handles both

non-Gaussian case and heteroscedastic setting, that we introduce as follows, $\forall r \in [|1 : K|], \forall t \in \mathcal{I}_r$:

$$x_t^i = f^{i,r}\left(\mathbf{Pa}_{\mathcal{G}^r}^i(<t), \mathbf{Pa}_{\mathcal{G}^r}^i(t)\right) + g^{i,r}\left(\mathbf{Pa}_{\mathcal{G}^r}^i(<t), \mathbf{Pa}_{\mathcal{G}^r}^i(t)\right) \cdot \epsilon_t^{i,r}, \tag{1}$$

where $f^{i,r}$ and $g^{i,r}$ are general differentiable functions, with $g^{i,r}$ strictly positive, and $\epsilon_t^{i,r}$ following an arbitrary probability density. We assume $\mathbb{E}(\epsilon_t^{i,r}) = 0$ and $\mathbb{E}((\epsilon_t^{i,r})^2) = 1$ without loss of generality. We denote the set of these temporal causal graphs as $\mathcal{G} = (\mathcal{G}^r)_{r \in [|1:K|]}$. In the case of non-stationary MTS, regimes appear sequentially with at least a minimum duration $\zeta$, and a subsequent regime $v$ (where $v = r + 1$) begins only after at least $\zeta$ time units from the start of regime $r$. Additionally, if regime $r$ reoccurs, its duration in the second appearance is also no less than $\zeta$ samples. We refer to the phenomenon, where each regime persists for at least $\zeta$ consecutive time steps, as the *regime persistent dynamics* assumption and we define it as follows:

**Assumption 2.2.** We consider a MTS with $K$ multiple regimes $(\boldsymbol{x}_t)_{t \in \mathcal{T}}$ where the SEM is defined in Eq.(1). Given variable $i$, we assume that a regime $r \in [|1 : K|]$ is $\zeta$-persistent if the parents $(\mathbf{Pa}_{\mathcal{G}^r}^i(<t), \mathbf{Pa}_{\mathcal{G}^r}^i(t))$ and functional dependencies $(f^{i,r}, g^{i,r}, \epsilon_t^{i,r})$ are stationary for $\zeta$ consecutive time steps $t$.

The persistence assumption permits to capture different regime dynamics, whether arising from changes in the causal graph across regimes, commonly observed in climate science [27] and epileptic recordings [56], or from shifts in functional dependencies, which correspond to soft interventions in causal discovery. Our newly introduced SEM in Eq.(1) generalizes several existing approaches in three novel aspects: (1) when $K = 1$, if $g^{i,1}(\mathbf{Pa}_{\mathcal{G}^1}^i(<t), \mathbf{Pa}_{\mathcal{G}^1}^i(t)) = 1$ for all $i \in [|1 : d|]$, we recover the classical additive noise models in causal discovery. Thus, allowing $g^{i,1}$ to be a strictly positive and differentiable function, *not only extends Rhino's SEM [15] but also yields a new, general SEM for stationary multivariate time series with heteroscedastic noise.* (2) When $K > 1$, if $g^{i,1}(\mathbf{Pa}_{\mathcal{G}^r}^i(<t), \mathbf{Pa}_{\mathcal{G}^r}^i(t)) = 1$, and $\epsilon_t^{i,r} \sim \mathcal{N}(0, \sigma_r)$ for all $i \in [|1 : d|]$, we recover the setting introduced in [39, 4]. Then, allowing $\epsilon_t^{i,r}$ to follow an arbitrary probability density then yields the first *SEM for non-stationary MTS composed of multiple regimes with non-Gaussian noise.* Finally, (3) permitting $g^{i,r}$ to be a strictly positive, differentiable function leads, to the best of our knowledge, to the first *general SEM for non-stationary MTS with heteroscedastic noise.*

## 3 Flow based approach for Dynamic Temporal Causal models

### 3.1 ELBO formulation

Many real-world time series e.g., EEG data [22, 40], climate data [27, 16], and financial data [18] are non-stationary and exhibit complex noise distributions. Existing causal discovery methods cannot jointly recover the number of regimes, their indices, and their structures under either non-Gaussian or heteroscedastic noises. With FANTOM, our objective is to close this gap, simultaneously learning the number of regimes $K$, their indices $\mathcal{I} = (\mathcal{I}_r)_{r \in [|1:K|]}$, and DAGs $\mathcal{G} = (\mathcal{G}^r)_{r \in [|1:K|]}$ in both homoscedastic non-Gaussian and general heteroscedastic settings. Because integrating over latent regimes makes the data log-likelihood intractable, we instead maximise its Evidence Lower BOund (ELBO). Proposition 3.1, proved in Appendix G, formalises this ELBO for $N_w > K$ provisional regimes. Section 3.2 outlines the initialization trick that instantiates these $N_w$ regimes, and the E-step that progressively merges them until $N_w$ settles at $K$.

**Proposition 3.1.** *Let $(\boldsymbol{x}_t)_{t \in \mathcal{T}}$ be a MTS composed of multiple regimes and following the SEM described in Eq.(1). The data likelihood admits the following evidence lower bound (ELBO):*

$$\log p_\Theta\left(\boldsymbol{x}_{t \in \mathcal{T}}\right) \geq \sum_{t=1}^{|\mathcal{T}|} \mathbb{E}_{q_\phi(\mathcal{G})}\left[\mathbb{E}_{p(z_t|\boldsymbol{x}_t,\boldsymbol{x}_{<t})}\left[\log p_{\theta^{z_t}}\left(\boldsymbol{x}_t \mid \boldsymbol{x}_{<t}, \mathcal{G}^{z_t}\right) + \log p\left(z_t\right)\right]\right.$$

$$\left. + H\left(p(z_t|\boldsymbol{x}_t, \boldsymbol{x}_{<t})\right)\right] + \sum_{r=1}^{N_w} \mathbb{E}_{q_{\phi^r}(\mathcal{G}^r)}\left[\log p\left(\mathcal{G}^r\right)\right] + H\left(q_{\phi^r}\left(\mathcal{G}^r\right)\right) \equiv \text{ELBO}(\Theta),$$

(2)

*where $\forall t \in \mathcal{T} : z_t \in [|1 : N_w|]$ are the discrete latent variables and $N_w$ is the number of regimes.*

Here, $\Theta$ are all the learnable parameters of our model (detailed in Section 3.2), $\log p_{\theta^{z_t}} (\boldsymbol{x}_t \mid \boldsymbol{x}_{<t}, \mathcal{G}^{z_t})$ represents the observational log-likelihood of $\boldsymbol{x}_t$ belonging to regime $z_t \in [\![1, N_w]\!]$, $q_{\phi^r}(\mathcal{G}^r)$ represents the variational distribution that approximates the intractable posterior $p_{\theta^r}(\mathcal{G}^r \mid \boldsymbol{x}_{t \in \mathcal{I}_r})$, and $p(z_t|\boldsymbol{x}_t, \boldsymbol{x}_{<t})$ represents the posterior distribution of the latent variables $z_t$. The distribution $p(\mathcal{G}^r)$ is the graph prior and $p(z_t)$ represents our prior belief about the membership of samples to the causal models; typically we model it as a time varying function.

## 3.2 Model parametrization

In this Section, we will define and motivate FANTOM design choices. FANTOM maximises the ELBO (Eq.(2)) with a Bayesian Expectation Maximization (BEM) scheme. Because BEM normally needs the number of regimes a priori, we first describe an initialisation trick that removes this requirement. Next, we motivate our prior over the latent regime indicator $z_t$ and show how a Temporal Graph Neural Network, combined with CNFs, models the regime-specific likelihood $p_{\theta^{z_t}}(\boldsymbol{x}_t \mid \boldsymbol{x}_{<t}, \mathcal{G}^{z_t})$.

**Initialization trick.** FANTOM initially divides the MTS into $N_w > K$ equal time windows in the initialisation step (the length of the initialized windows is greater than $\zeta$ minimum regime duration), where each window represents one initial regime estimate. Our initialization scheme builds some initial pure regimes (regimes composed of samples from the same ground truth regime) and other impure ones (regimes composed of samples from two neighboring ground truth regimes). After initialization, FANTOM alternates between two phases. The E-step (subsection 3.4) updates the regime indices $\mathcal{I} = (\mathcal{I}r)r \in [\![1 : N_w]\!]$ and removes regimes with too few samples, updating $N_w$. The M-step (subsection 3.3.1) learns the graphs $(\mathcal{G}r)r \in [\![1 : N_w]\!]$ and models heteroscedastic noise with a Bayesian structure-learning method that uses conditional normalizing flows (CNFs). These phases repeat until the algorithm reaches the maximum number of iterations.

**BEM choice motivation.** We argue that inferring regimes and learning their associated DAGs are interdependent tasks, making the BEM algorithm particularly well-suited for this problem. A two-step alternative, that detects change points with KCP [1], then runs causal discovery on each segment breaks down: First, change point detection methods like KCP [1] fail to detect regime shifts driven by changes in causal mechanisms, because those involve shifts in conditional distributions (See Appendix F.3.3). It also treats recurring regimes as distinct, forcing redundant model fits and raising computation costs. Second, heteroscedastic noise further degrades existing causal methods (Table 2). FANTOM addresses all three issues by coupling CNFs, which capture heteroscedasticity, with Bayesian structure learning that prunes spurious edges.

**Time varying weight modeling.** We use time-varying weights, initially proposed for financial data modeling [59, 57], as priors for the discrete latent variables $z_t$. To support smooth regime transitions consistent with our persistence assumption (Assumption 2.2), we adopt a flexible functional form based on the softmax transformation of learnable parameter $\boldsymbol{\omega}^r \in \mathbb{R}^2$ and time index $t$: $p(z_t = r) = \pi_t(\boldsymbol{\omega}^r) = \frac{\exp(\omega_1^r \cdot t + \omega_0^r)}{\sum_{j=1}^{N_w} \exp(\omega_1^j \cdot t + \omega_0^j)}$. This formulation encourages that, if $\boldsymbol{x}_t$ belongs to regime $r$ in the current iteration, it is only allowed to remain in the same regime $r$ or smoothly transition to neighboring regimes $(r-1, r+1)$ in the next iteration. See Section 3.4 and Appendix B for details.

**Bayesian structure learning.** Following [14, 15, 55], FANTOM employs Bayesian structure learning. We approximate the intractable posterior $p_\theta(\mathcal{G} \mid \boldsymbol{x}_{t \in \mathcal{T}})$ using the variational distribution $q_\phi(\mathcal{G}) = \prod_{r=1}^{N_w} q_{\phi^r}(\mathcal{G}^r) \delta(\theta^r)$, where $\delta$ denotes the Dirac delta function. Following [14, 15, 55], we model $q_{\phi^r}(\mathcal{G}^r)$ as a product of independent Bernoulli variables and compute its expectation with a single Monte Carlo sample using the Gumbel-Softmax trick [26]. Additional details are in Appendix C.

**Likelihood of SEM.** Using the functional form Eq.(1), we have

$$y_t^{i,r} = g^{i,r}\left(\mathbf{Pa}_{\mathcal{G}^r}^i(<t), \mathbf{Pa}_{\mathcal{G}^r}^i(t)\right) \cdot \epsilon_t^{i,r}$$
$$= x_t^i - f^{i,r}\left(\mathbf{Pa}_{\mathcal{G}^r}^i(<t), \mathbf{Pa}_{\mathcal{G}^r}^i(t)\right),$$

then we can write the observational likelihood:

$$p_{\theta^r}\left(x_t^i \mid \mathbf{Pa}_{\mathcal{G}^r}^i(<t), \mathbf{Pa}_{\mathcal{G}^r}^i(t)\right) = p_{\text{hetero}}\left(y_t^{i,r} \mid \mathbf{Pa}_{\mathcal{G}^r}^i(<t), \mathbf{Pa}_{\mathcal{G}^r}^i(t)\right) \quad (3)$$

where $p_{\text{hetero}}$ refers to the density function of the heteroscedastic conditions. To estimate $f^{i,r}$, we build upon the model formulation of [14], which uses neural networks to describe the functional

relationship $f_{\theta^r}^{i,r} : \mathbb{R}^d \to \mathbb{R}$. Specifically, we propose flexible functional designs for $f^{i,r}$, which must respect the relations encapsulated in $\mathcal{G}^r$. Namely, if $x_{t-\tau}^j \notin \mathbf{Pa}_{\mathcal{G}^r}^i(<t) \cup \mathbf{Pa}_{\mathcal{G}^r}^i(t)$, then $\partial f^{i,r}/\partial x_{t-\tau}^j = 0$. We design

$$f_{\theta^r}^{i,r}\left(\mathbf{Pa}_{\mathcal{G}^r}^i(<t), \mathbf{Pa}_{\mathcal{G}^r}^i(t)\right) = \boxed{\psi^r}\left(\boxed{\sum_{\tau=0}^{L}\sum_{j=1}^{d} G_{\tau,ji}^r}\,\boxed{\vartheta^r}\left(x_{t-\tau}^j, \boxed{e_{\tau,j}^r}\right), \boxed{e_{0,i}^r}\right), \qquad (4)$$

where $\psi^r$ and $\vartheta^r$ are neural network blocks illustrated in Figure 2 with all the other colored blocks. Instead of using a neural network block per node, we adopt a weight-sharing mechanism by using a trainable embeddings $e_{\tau,i}$ for $\tau \in [\![0:L]\!]$ and $i \in \{1, \cdots, d\}$. For the heteroscedastic term, we introduce an invertible mapping $\ell_{\theta^r}^{i,r} : \mathbb{R} \to \mathbb{R}$ such that:

$$\ell_{\theta^r}^{i,r}\left(g^{i,r}\left(\mathbf{Pa}_{\mathcal{G}^r}^i(<t), \mathbf{Pa}_{\mathcal{G}^r}^i(t)\right) \cdot \epsilon_t^{i,r}\right) = n_t^{i,r},$$

where $n_t^{i,r} \sim \mathcal{N}(0,1)$. The design of $\ell_{\theta^r}^{i,r}$ needs to properly balance the flexibility and tractability of the transformed noise density. We choose a conditional normalizing flows, for heteroscedastic noise, called conditional spline flow [10], that transforms our heteroscedastic noise distribution to a fixed normal noise $n_t^{i,r}$ for regime $r$. The spline parameters are predicted using a hyper-network with a similar form as Eq.(4) to incorporate heteroscedasticity. Due to the invertibility of $\ell_{\theta^r}^{i,r}$, the noise likelihood conditioned on all parents is:

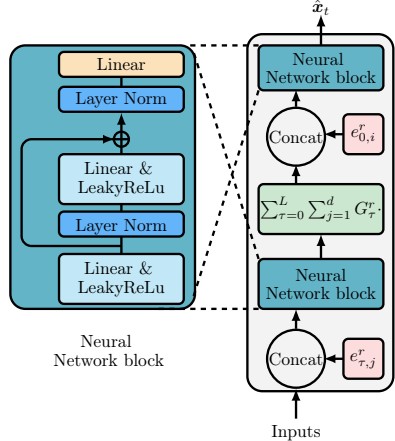

Figure 2: Temporal graph neural network (TGNN) used by FANTOM.

$$p_{\text{hetero}}\left(g^{i,r}\left(\mathbf{Pa}_{\mathcal{G}^r}^i(<t), \mathbf{Pa}_{\mathcal{G}^r}^i(t)\right) \cdot \epsilon_t^{i,r} \mid \mathbf{Pa}_{\mathcal{G}^r}^i(<t), \mathbf{Pa}_{\mathcal{G}^r}^i(t)\right) = p_{n^{i,r}}\left(n_t^{i,r}\right)\left|\frac{\partial \ell_{\theta^r}^{i,r}}{\partial \epsilon_t^{i,r}}\right|, \qquad (5)$$

where $p_{n^{i,r}}(\cdot)$ is the standard normal density. In the non-Gaussian, *non-heteroscedastic* case, FANTOM learns a base distribution based on a composite affine-spline transformation of a standard Gaussian. Finally, the system parameters comprise the learnable parameters of the time varying weights $\boldsymbol{\omega}^r$, of the variational inference $\phi^r$ and of the neural networks $\theta^r$. We use $\Theta$, introduced in Eq.(2), to note all the learnable parameters of FANTOM, and we have the set of parameters is: $\Theta = \{(\boldsymbol{\omega}^r, \phi^r, \theta^r)\}_{r=1}^{N_w}$.

### 3.3 BEM algorithm

#### 3.3.1 M step: graph learning

FANTOM applies BEM to maximize the ELBO described in Proposition 2. It begins by initializing the regime partitions $\beta_t^r = p(z_t = r|\boldsymbol{x}_t, \boldsymbol{x}_{<t})$ using equally sized windows, selected via a hyperparameter. Note that these binary regime indices $\beta_t^r$ are updated during the E-step. Then, in the M-step, FANTOM incorporates $\beta_t^r$ in the ELBO Eq.(2) to estimate the DAGs for each regime and learn the parameters $\boldsymbol{\omega}^r$ that align $\pi_t(\boldsymbol{\omega}^r)$ with $\beta_t^r$. We have:

$$\arg\max_{\Theta}\left\{\mathbb{E}_{q_\phi(\mathcal{G})}\left[\sum_{t=1}^{|\mathcal{T}|}\sum_{r=1}^{N_w}\beta_t^r \log p_{\theta_r}\left(\boldsymbol{x}_t \mid \boldsymbol{x}_{<t}, \mathcal{G}^r\right)\right] + \sum_{r=1}^{N_w}\mathbb{E}_{q_{\phi_r}(\mathcal{G}^r)}\left[\log p\left(\mathcal{G}^r\right)\right] + H\left(q_{\phi_r}\left(\mathcal{G}^r\right)\right)\right.$$
$$\left. + \sum_{t=1}^{|\mathcal{T}|}\sum_{r=1}^{N_w}\beta_t^r \log \pi_t(\boldsymbol{\omega}^r)\right\},$$

$$(6)$$

The maximization of the above equation can be decomposed into two distinct and separate maximization problems. The first problem, regime alignment, focuses on aligning $\pi_t(\boldsymbol{\omega}^r)$ with $\beta_t^r$. While the

second one, graph learning, involves estimating DAGs for every regime.

For the graph prior $p\left(\mathcal{G}^r\right)$ for all $r \in \llbracket 1 : N_w \rrbracket$ have to combine two components: DAG constraint and graph sparseness prior. Inspired by [60, 15, 14], we propose the following un-normalised prior $p\left(\mathcal{G}^r\right) \propto \exp\left(-\lambda_s \left\|\boldsymbol{G}_{0:K}^r\right\|_F^2 - \rho h^2\left(\boldsymbol{G}_0^r\right) - \alpha h\left(\boldsymbol{G}_0^r\right)\right)$. Using Eq.(3), we have $\log p_{\theta^r}\left(\boldsymbol{x}_t \mid \boldsymbol{x}_{<t}, \mathcal{G}^r\right) = \sum_{i=1}^d \log p_{\text{hetero}}\left(y_t^{i,r} \mid \mathbf{Pa}_{\mathcal{G}^r}^i(<t), \mathbf{Pa}_{\mathcal{G}^r}^i(t)\right)$, where $p_{\text{hetero}}$ and $y_t^{i,r}$ are defined in Eq.(5). The parameters $\theta^r, \phi^r$ are learned by maximizing the graph learning maximization problem Eq.(6), where the Gumbel-softmax gradient estimator is used [26]. We also leverage augmented Lagrangian training similar to [35, 39, 14], to anneal $\alpha, \rho$.

### 3.4 E step: Regime learning

In the E-step, FANTOM updates the posterior probability $\beta_t^r = p(z_t = r | \boldsymbol{x}_t, \boldsymbol{x}_{<t})$ (see Eq.(7), with derivation provided in Appendix B):

$$\beta_t^r = \frac{p\left(z_t = r\right) p\left(\boldsymbol{x}_t \mid \boldsymbol{x}_{<t}, z_t = r, \mathcal{G}^r\right)}{\sum_{j=1}^{N_w} p\left(z_t = j\right) p\left(\boldsymbol{x}_t \mid \boldsymbol{x}_{<t}, z_t = j, \mathcal{G}^j\right)} \tag{7}$$
$$\propto \pi_t(\boldsymbol{\omega}^r) p\left(\boldsymbol{x}_t \mid \boldsymbol{x}_{<t}, z_t = r, \mathcal{G}^r\right),$$

where $p(\boldsymbol{x}_t \mid \boldsymbol{x}_{<t}, z_t = r, \mathcal{G}^r)$ denotes the observational likelihood of $\boldsymbol{x}_t$ being generated by the SEM from Eq (1) for regime $r$. This probability is computed using the CNFs trained during the M-step, following the same reasoning as in Eq.(3).

The probability of $\boldsymbol{x}_t$ belonging to regime $r$ is influenced by two main factors: the observation's position within its current regime and whether that regime is designated as pure or impure. For example, if $\boldsymbol{x}_t$ is in a pure regime $r$ but is near the boundary in the current iteration, $\pi_t(\boldsymbol{\omega}^r)$ and $\pi_t(\boldsymbol{\omega}^{r+1})$ are nearly equal (e.g., $\pi_{t \in [1100,1500]}(\boldsymbol{\omega}^1)$ vs. $\pi_{t \in [1100,1500]}(\boldsymbol{\omega}^2)$ in Figure 3). Nonetheless, since regime $r$ was learned from pure data, $p(\boldsymbol{x}_t \mid \boldsymbol{x}_{<t}, z_t = r, \mathcal{G}^r)$ stays high, keeping $\beta_t^r$ at its maximum value and maintaining $\boldsymbol{x}_t$ in regime $r$ for the next iteration. In the other hand, if $\boldsymbol{x}_t$ is in an impure regime $r + 1$ near the boundary during the current iteration, $\pi_t(\boldsymbol{\omega}^r)$ and $\pi_t(\boldsymbol{\omega}^{r+1})$ are also close in value (e.g., $\pi_{t \in [1501,1800]}(\boldsymbol{\omega}^1)$ vs. $\pi_{t \in [1501,1800]}(\boldsymbol{\omega}^2)$ in Figure 3). However, because the causal graph for regime $r$ is more reliable (having been derived from pure data), $p(\boldsymbol{x}_t \mid \boldsymbol{x}_{<t}, z_t = r, \mathcal{G}^r) > p(\boldsymbol{x}_t \mid \boldsymbol{x}_{<t}, z_t = r + 1, \mathcal{G}^{r+1})$. As a result, $\boldsymbol{x}_t$ moves from regime $r + 1$ to $r$ in the next iteration.

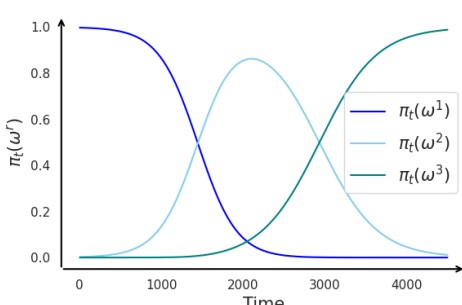

Figure 3: Illustration of $\pi_t\left(\boldsymbol{\omega}^r\right)$ after Fantom's first iteration with equal windows of 1500 samples for an MTS of 4500 samples with two ground-truth regimes: $\mathcal{I}_1^* = \llbracket 0 : 1999 \rrbracket$ and $\mathcal{I}_2^* = \llbracket 2000 : 4500 \rrbracket$.

For simplicity reasons, we explicit these cases from one border but the same thing happens in the other border which accelerates convergence. More details about other cases and Figures that illustrate the idea could be found in Appendix B.

After updating $\beta_t^r$, for each sample $\boldsymbol{x}_t$, FANTOM assigns a value of 1 to the most probable regime $r$ (with the highest $\beta_t^r$), and 0 to others. Additionally, FANTOM filters out regimes with insufficient samples (fewer than $\zeta$, the minimum regime duration, defined as a hyper-parameter). Discarded regime samples are then reassigned to the nearest regime in terms of probability $\beta_t^r$ in the subsequent iteration which is in general a neighboring regime ensured by the way we set up the probability $\beta_t^r \propto \pi_t(\boldsymbol{\omega}^r) p\left(\boldsymbol{x}_t \mid \boldsymbol{x}_{<t}, z_t = r, \mathcal{G}^r\right)$.

## 4 Identifiability results

Identifiability is an important statistical property to ensure that the causal discovery problem is meaningful. In causal analysis, the whole point is to find out which variable causes others; if the model is not identifiable, the analysis is not possible at all. This section proves that causal discovery from multi-regime MTS is identifiable in the FANTOM framework namely, when (i) the noise is

non-Gaussian or heteroscedastic and (ii) the latent variable $z_t$ has a time-varying-parent prior. We first formalize identifiability for this setting, then state three theorems covering both stationary and multi-regime MTS under the two noise assumptions.

> **Definition 4.1.** The conditional distribution of multi-regime MTS with a time varying prior is: $p(\boldsymbol{x}_t|\boldsymbol{x}_{<t}) = \sum_{r=1}^{K} \pi_t(\boldsymbol{\omega}^r) p_{\theta^r}(\boldsymbol{x}_t|\boldsymbol{x}_{<t}, \mathcal{G}^r)$. We say this model is identifiable up to permutation and translation, if:
> - $\forall r \in [|1:K|]$ the causal model $(\theta^r, \mathcal{G}^r)$ is identifiable.
> - For any two models with parameters $(\boldsymbol{\omega}^r, \theta^r, \mathcal{G}^r)_{r=1}^{K}$ and $(\tilde{\boldsymbol{\omega}}^r, \tilde{\theta}^r, \tilde{\mathcal{G}}^r)_{r=1}^{\tilde{K}}$, such that for any $t \in \mathcal{T}$: $p(\boldsymbol{x}_t|\boldsymbol{x}_{<t}) = \tilde{p}(\boldsymbol{x}_t|\boldsymbol{x}_{<t})$, we have $K = \tilde{K}$ and it exists a permutation $\sigma$ and translation function $\varrho : \mathbb{R}^2 \to \mathbb{R}^2$ such that $\theta^r = \tilde{\theta}^{\sigma(r)}$ and $\boldsymbol{\omega}^r = \varrho(\tilde{\boldsymbol{\omega}}^{\sigma(r)})$

Following [15, 39, 7], we use the common assumptions of causal discovery settings (Causal Markov property H.2, stationarity H.2, minimality H.2, sufficiency H.2), see Appendix H.1 for precise statements. We present our first theoretical results, Theorem 4.2, states that for any stationary MTS, composed of $K = 1$ regime and following Eq.(1) with $\epsilon_t^i \sim \mathcal{N}(0, 1)$ the ground truth solution $\mathcal{G}^*$ is uniquely identifiable, the detailed proof can be found in Appendix H.

> **Theorem 4.2** (Identifiability of Temporal Heteroscedastic Gaussian noise model (THGNM)). *Assume Causal Markov property, stationarity, minimality, sufficiency and let $(\boldsymbol{x}_t)_{t \in \mathcal{T}}$ be a MTS following a THGNM, $\forall t \in \mathcal{T}$:*
>
> $$x_t^i = f^i\left(\mathbf{Pa}_{\mathcal{G}}^i(<t), \mathbf{Pa}_{\mathcal{G}}^i(t)\right) + g^i\left(\mathbf{Pa}_{\mathcal{G}}^i(<t), \mathbf{Pa}_{\mathcal{G}}^i(t)\right) \cdot \epsilon_t^i, \tag{8}$$
>
> *where $f^i$ and $g^i$ are differentiable functions, with $g_i$ strictly positive and $\epsilon_t^i \sim \mathcal{N}(0, 1)$ are mutually independent normal noises. The THGNM is identifiable if $\frac{1}{g^i}$ is not a polynomial of degree two.*

**Identifiability of Temporal Restricted Heteroscedastic noise model (TRHNM).** We present and prove in Appendix H.3 our second identifiability results of a TRHNM (Theorem H.10), where $\epsilon_t^i$ can follow any arbitrary density distribution. We states the results for bivariate time series, in which we show that, if a backward model exists, a differential equation will always hold. Then inspired from Peters et al. [38], Immer et al. [25], and Strobl et al. [50], we define TRHNM and show its identifiability.

Our last theoretical result states that the mixture of identifiable temporal causal models with either non-Gaussian or heteroscedastic noises is also identifiable as defined in definition 4.1.

> **Theorem 4.3** (Identifiability of the mixture of identifiable temporal causal models). *Let $\mathcal{F}$ be a family of $K$ identifiable temporal causal models, $\mathcal{F} = (p_{\theta^r}(\cdot|\cdot, \mathcal{G}^r))_{r \in \mathbb{N}^*}$ that are linearly independent and let $\mathcal{M}_K$ be the family of all $K$-finite mixtures of elements from $\mathcal{F}$, i.e.,*
>
> $$\left\{ p(\boldsymbol{x}_t|\boldsymbol{x}_{<t}) = \sum_{r=1}^{K} \pi_t(\boldsymbol{\omega}^r) p_{\theta^r}(\boldsymbol{x}_t|\boldsymbol{x}_{<t}, \mathcal{G}^r), p_{\theta^r}(\cdot|\cdot, \mathcal{G}^r) \in \mathcal{F}, \forall t \in \mathcal{T} : \pi_t(\boldsymbol{\omega}^r) > 0 \text{ and } \sum_{r=1}^{K} \pi_t(\boldsymbol{\omega}^r) = 1 \right\}.$$
>
> *Then the family $\mathcal{M}_K$ is identifiable as defined in definition 4.1.*

Identifiability is important in causal discovery as shown in several papers [7, 15, 35, 5]. We extend these guarantees to FANTOM's settings by proving identifiability for stationary temporal models with heteroscedastic noise, showing that weaker assumptions suffice when the noise $\epsilon_t^i$ is simplified, recovering identifiability under explicit restrictions for arbitrary noise, and covering multi-regime MTS scenarios in both cases. Although convergence rates or finite-sample bounds remain elusive because the BEM objective is non-convex, our experiments demonstrate that FANTOM still converges in non-Gaussian and heteroscedastic settings. Further convergence rates or finite-data bounds are however extremely challenging due to the non-convexity of the acyclicity constraint in a BEM procedure. Yet, we empirically demonstrate, in the experiments section, that FANTOM converges in both non-Gaussian and heteroscedastic cases.

Table 1: Average SHD, F1 scores, NHD and Ratio for different models with $d = 10$ nodes and $K = 3$ regimes. *Split* denotes whether regime separation is automatic (✓) or manual (×), and *Type* classifies the graph as window (W) or summary (S). *Inst.* refers to instantaneous links, and *Lag* to time-lagged edges.

| Model | Split | Type | Homoscedastic non-Gaussian noise | | | | | | | | | Heteroscedastic noise | | | | | | | | |
| | | | Inst. | | | | Lag | | | | Regime | Inst. | | | | Lag | | | | Regime |
| | | | SHD↓ | F1↑ | NHD↓ | Ratio↓ | SHD↓ | F1↑ | NHD↓ | Ratio↓ | Acc. | SHD↓ | F1↑ | NHD↓ | Ratio↓ | SHD↓ | F1↑ | NHD↓ | Ratio↓ | Acc. |
| PCMCI+ | × | W | 17.5 | 74.9 | 0.02 | 0.24 | 14.5 | 88.3 | 0.01 | 0.11 | × | 46.1 | 11.1 | 0.05 | 0.88 | 46.0 | 19.0 | 0.05 | 0.80 | × |
| Rhino | × | W | 2.50 | 96.8 | 0.002 | 0.03 | **6.00** | **95.2** | **0.006** | **0.04** | × | 44.5 | 5.11 | 0.06 | 0.94 | 53.5 | 64.7 | 0.07 | 0.35 | × |
| DYNOTEARS | × | W | 42.0 | 54.4 | 0.06 | 0.45 | 21.5 | 82.1 | 0.02 | 0.17 | × | 89.5 | 31.5 | 0.14 | 0.68 | 118.0 | 37.5 | 0.17 | 0.61 | × |
| CASTOR | × | W | 22.0 | 66.2 | 0.03 | 0.33 | 17.0 | 84.4 | 0.01 | 0.15 | × | 104.0 | 23.4 | 0.19 | 0.76 | 133.5 | 34.8 | 0.24 | 0.64 | × |
| RPCMCI | ✓ | W | - | - | - | - | - | - | - | - | - | - | - | - | - | - | - | - | - | - |
| CASTOR | ✓ | W | 47.0 | 34.8 | 0.05 | 0.65 | 59.5 | 39.4 | 0.07 | 0.60 | 51.6 | - | - | - | - | - | - | - | - | - |
| FANTOM | ✓ | W | **0.33** | **99.5** | **0.00** | **0.00** | 12.5 | 89.1 | 0.01 | 0.10 | **99.4** | **5.67** | **93.3** | **0.006** | **0.06** | **12.3** | **90.9** | **0.012** | **0.08** | **97.1** |

| Model | Split | Type | SHD↓ | F1↑ | NHD↓ | Ratio↓ | Acc. | SHD↓ | F1↑ | NHD↓ | Ratio↓ | Acc. |
| --- | --- | --- | --- | --- | --- | --- | --- | --- | --- | --- | --- | --- |
| CD-NOD | × | S | 42.5 | 31.8 | 0.42 | 0.67 | × | 42 | 6.15 | 0.61 | 0.93 | × |
| FANTOM | ✓ | S | **4.5** | **95.6** | **0.04** | **0.04** | **96.6** | **4.5** | **96.5** | **0.04** | **0.03** | **96.8** |

# 5 Experiments

## 5.1 Synthetic data

**Data generation.** We conduct extensive experiments to evaluate FANTOM's performance on synthetic datasets. For ground truth graph generation, we use the Barabási-Albert model (degree 4) for instantaneous links and the Erdos–Rényi model (degree 1–2) for time-lagged relationships. For data generation process, $f_i^r, g_i^r$ are chosen to be randomly initialized MLPs with one hidden layer and activation functions randomly chosen from {Tanh, Exp}. We evaluate the different models on multiple complex noise distributions; non-stationary MTS with either (1) heteroscedastic noise or (2) non-Gaussian homoscedastic noise, details in Appendix E.1. We consider $L = 1$, while additional experiments with multiple lags are provided in the Appendix F.1. Regime durations are randomly selected from {1000, 1500, 2000, 2500}. We test different numbers of nodes {5, 10, 20, 40} and varying regime counts ($K \in \{2, 3\}$). Each combination of $K$ and $d$ nodes is repeated three times, resulting in over 24 distinct datasets [1].

**Benchmarks.** We benchmark our model against several baselines, including causal discovery methods for MTS with multiple regimes, such as CASTOR [39], CD-NOD [21] and RPCMCI [45]. Since CD-NOD returns a summary graph (see Appendix F.1), we compute a comparable summary graph from FANTOM's output for fair evaluation. FANTOM is also compared with models for single-regime MTS, including Rhino [15], PCMCI+ [42], DYNOTEARS [35]. *Given that these models cannot deal with multiple regimes, we put them in a more favorable position than ours and provide these models with the true regime partition information. This is done by training the aforementioned models on each pure regime separately (regime governed by the same causal model).*

**Evaluation Metrics.** We assess the performance of our proposed method for learning the DAGs using four key metrics: 1) F1 score, representing the harmonic mean of precision and recall; 2) Structural Hamming Distance (SHD), which counts discrepancies (e.g., reversed, missing, or redundant edges) between two DAGs; 3) Normalized Hamming Distance (NHD) measures how many edges differ normalized by the total number of possible edges; 4) Ratio NHD computes the ratio between the NHD and the baseline NHD of an output with the same number of edges but with all of them incorrect. For the regime detection task, we use Accuracy (Reg Acc) metric.

**Results and discussion.** Table 1 shows results on MTS with multiple regimes under heteroscedastic noise (right part of the table) and homoscedastic non-Gaussian noise (left part of the table). **In the homoscedastic, non-Gaussian scenario**, baselines generally perform better in the graph learning task, yet FANTOM still surpasses them on regime detection, with 99.4% accuracy, and instantaneous links, 99.5% F1. For the regime detection task, CASTOR succeeds to detect the regimes but with low accuracy (51.6%) compared to FANTOM (99.4%) and this due to the fact that CASTOR assumes equivariance normal noise with, however RPCMCI does not converge in this case too. For time-lagged connections, Rhino (95.2% F1), that has access to the ground-truth regime labels by training it on pure regime separately, slightly outperforms FANTOM (89.1% F1) that learns simultaneously the number of regimes, their indices and structures. **In the heteroscedastic setting**, FANTOM consistently outperforms both multi-regime baselines (CASTOR, RPCMCI, CD-NOD) and stationary approaches. It achieves the top scores on all metrics: for instantaneous links, an F1 of 93.3%, a 60%

---

[1]https://github.com/arahmani1/fantom.git

improvement over the second-best, and a ratio of 0.06, 0.64 lower than the next-best DYNOTEARS. For time-lagged links, an F1 of 90.9%, 25% higher than Rhino, and a ratio of 0.08. FANTOM also detects the correct number of regimes and their indices with over 96% accuracy. By contrast, RPCMCI struggles to converge due to its homoscedastic assumption and time-lag-only dependencies, CASTOR relies on Gaussian noise and cannot detect regime labels in the absence of ground truth, DYNOTEARS, PCMCI+, CD-NOD, and Rhino likewise fail in this heteroscedastic scenario, even when regime labels are given a priori. Although Rhino models history-dependent noise, it does not handle general heteroscedasticity. Overall, the table shows that FANTOM is the only method that remains robust across both noise scenarios, matching or surpassing specialised baselines while simultaneously discovering regimes and graphs.

Appendices F.2.1 and F.2.2 report additional results with standard deviations, confirming that FANTOM sustains its performance when scaled to graphs of 20 and 40 nodes. Ablation study in the Appendix F.1 further shows FANTOM's robustness towards the choice of initialized window and $\zeta$.

## 5.2 Real world data

**Wind Tunnel.** We use the wind tunnel datasets from Gamella et al. [13], featuring two controllable fans pushing air through a chamber, barometers measuring air pressure at various locations, and a hatch controlling an external opening. The dataset comprises 16 variables across two regimes of 10,000 samples each: the first is observational, while the second involves soft interventions on five variables (see Appendix E.4.1). We compare FANTOM to the aforementioned baselines, with results in Table 2. FANTOM is the only model that detects the regime with 99.9% accuracy and outperforms all baselines on the causal graph learning task, achieving 38.5% on F1 score. Notably, FANTOM surpasses all baselines even when they are given the ground-truth regime partitions.

Table 2: Performance on Wind Tunnel data evaluated on summary causal graph.

| | Split | SHD↓ | F1↑ | Ratio↓ | Reg Acc. |
|---|---|---|---|---|---|
| PCMCI+ | × | 37 | 22.9 | 0.77 | × |
| DYNOTEARS | × | 34 | 0 | 1 | × |
| CASTOR | × | 104 | 17.2 | 0.82 | × |
| CD-NOD | × | 40 | 20.0 | 0.80 | × |
| Rhino | × | 47 | 32.0 | 0.68 | × |
| CASTOR | ✓ | 120 | 19.5 | 0.80 | 49.9 |
| FANTOM | ✓ | **29** | **38.5** | **0,61** | **99.9** |

**Epilepsy detection.** Huizenga et al. [22] show that scalp potential fields are contaminated by heteroscedastic noise in EEG measurements. We evaluate FANTOM's performance in detecting epileptic regimes using EEG signals from 10 different patients in the Temple University Hospital EEG Seizure Corpus (TUSZ) dataset [40, 52]. We treat this as an unsupervised regime detection problem, analyzing roughly 100 seconds of recordings at a 250 Hz sampling rate for each patient, capturing both normal and seizure states (see Appendix E.4.2). The recordings consist of 19 electrodes, each considered a causal variable. FANTOM detects the correct regime partitions with an average 82.7% accuracy across all patients. The seizure regime's learned graph is denser and more connected than that of the normal state, which aligns with the generalized seizures affecting multiple brain regions. Full details and illustrations are provided in Appendix E.4.2.

## 6 Conclusion

We introduced FANTOM, a unified framework for multi-regime MTS that jointly infers *(i)* the number of regimes, *(ii)* their boundaries, and *(iii)* their corresponding causal DAG, under either non-Gaussian or heteroscedastic noises. Under mild assumptions in causal discovery, we prove identifiability of the temporal heteroscedastic causal models in the stationary case and show that the number of regimes, their indices, and their graphs are identifiable (up to permutation) in the non-stationary setting. Extensive experiments on synthetic and real-world data show consistent gains over strong baselines. FANTOM offers a principled means to uncover regime-specific causal dynamics, enhancing regime detection, and causal discovery with potential applications in various domains such as finance, climate science, and neuroscience.

## Acknowledgment

We thank Alessandro Favero, Nikolaos Dimitriadis, and Ortal Senouf for helpful feedbacks and comments. This work was supported by the SNSF Sinergia project 'PEDESITE: Personalized Detection of Epileptic Seizure in the Internet of Things (IoT) Era'

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

**Table of content**

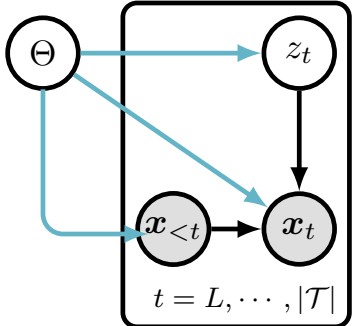

Figure 4: Graphical model of FANTOM. Observed variables ($\boldsymbol{x}_t$) are in gray, while latent variables ($z_t$) and parameters ($\Theta$) are in white. Blue edges represent parameter-variable interactions.

# A    Detailed related works

**Causal structure learning from IID data.** Causal structure learning has become an active area of research. hasan et al. [19] recently presented a comprehensive review of causal discovery methods for IID data and time series. For IID data, several approaches rely on conditional independence to infer causal relationships from observational data, such as the classical PC algorithm [49]. Additionally, some methods extend beyond observational data, incorporating interventional data to enhance causal inference, including COmbINE [53] and HEJ [23]. These approaches utilize data collected from controlled interventions to uncover causal relationships. A novel research direction introduced in [60] addresses the combinatorial challenges of structure learning by formulating it as a continuous constrained optimization problem, thus avoiding computationally costly combinatorial searches. Similarly, Zhu et al. [61] utilize the acyclicity constraint but employ reinforcement learning techniques for estimating directed acyclic graphs (DAGs). In contrast, Ke et al. [28] propose an approach that learns DAGs from interventional data by optimizing an unconstrained objective function. [7] provide a comprehensive analysis of continuous-constrained methods, offering a generalized framework applicable to interventional data scenarios. Another significant method, DiBS [31], estimates the full posterior distribution over Bayesian networks from limited observations, enabling quantification of uncertainty and assessment of confidence in causal discovery.

**Causal structure learning from stationary MTS.** The previously mentioned state-of-the-art methods primarily target independent observations rather than temporal dependencies. Assaad et al. [2] provide a comprehensive review of approaches specifically designed for causal discovery from MTS. To model causal relationships involving time dependencies, researchers frequently employ Dynamic Bayesian Networks (DBNs), which effectively capture discrete-time temporal dynamics within directed graphical frameworks. Some methods neglect contemporaneous (instantaneous) dependencies and focus exclusively on recovering time-lagged causal links [20, 48], and tsFCI [11], the latter adapting the Fast Causal Inference algorithm [49] for time series data. Runge et al. [43] introduced PCMCI, a scalable two-stage algorithm for time series, initially focusing only on time-lagged relationships. They subsequently extended it to PCMCI+ [42], enabling the identification of contemporaneous causal connections. Additionally, models addressing non-Gaussian instantaneous effects have been developed, such as VARLINGAM [24], which integrates non-Gaussian instantaneous models with autoregressive components. Another significant method is Time-series Models with Independent Noise (TiMINo) [36], which studies nonlinear and instantaneous effects using constrained SEMs. Pamfil et al. [35] recently proposed DYNOTEARS, leveraging an algebraic characterization of graph acyclicity from [60] to estimate both instantaneous and time-lagged relationships from time series data. DYNOTEARS utilizes a score-based DBN learning approach optimized via an augmented Lagrangian framework, enabling causal graph inference without assumptions on the underlying topology. In contrast, methods like NBCB [3], a noise-based/constraint-based approach, aim to learn a summary causal graph directly from observational time series data, going beyond Markov equivalence constraints even in the presence of instantaneous relationships. Rhino [15] introduced the first model that tackle stationary MTS with historical noise,where they assume that the noise variance changes over time as a function of solely time lagged

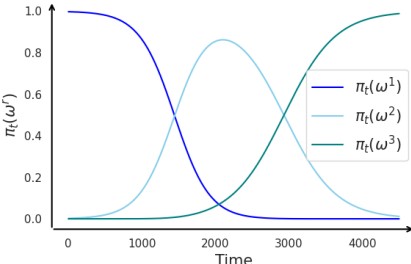

Figure 5: Illustration of $\pi_t(\omega^r)$ after Fantom's first iteration with equal windows of 1500 samples for an MTS of 4500 samples with two ground-truth regimes: $\mathcal{I}_1^* = [|0 : 1999|]$ and $\mathcal{I}_2^* = [|2000 : 4500|]$.

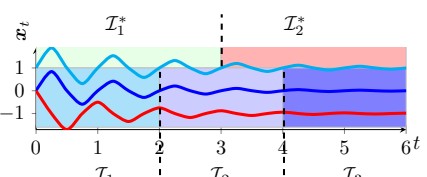

Figure 6: Example of initialization with $N_w = 3$ windows, $\mathcal{I}_1$ and $\mathcal{I}_3$ are pure regimes while $\mathcal{I}_2$ is impure (composed of samples from two ground-truth regimes $\mathcal{I}_1^*$ and $\mathcal{I}_2^*$, $K = 2$).

parents. All the aforementioned methods does not handle heteroscedastic setting and also fail short in the case of MTS with multiple regimes.

**Causal structure learning from MTS with multiple regimes.** Some research have aimed to address this challenge by developing methods for causal discovery in heterogeneous data. An example of such a method is CD-NOD [21], tackles time series with various regimes. By using the time stamp IDs as a surrogate variable, CD-NOD output one summary causal graph where the parents of each variable are identified as the union of all its parents in graphs from different regimes. Then it detects the change points by using a non-stationary driving force that estimates the variability of the conditional distribution $p(x_i|\text{union parents of } x_i)$ over the time index surrogate. While CD-NOD provides a summary graph capturing behavioral changes across regimes, it falls short in inferring individual causal graphs, also CD-NOD cannot handle either non-Gaussian or heteroscedastic noise. The overall summary graph does not effectively highlight changes between regimes. Additionally, CD-NOD detects the change points but fails to determine the regime indices, rendering it incapable of inferring the precise number of regimes. In scenarios involving recurring regimes, CD-NOD is unable to detect this crucial information. Another relevant work dealing with MTS composed of multiple regimes is RPCMCI [45]. In this paper, the model [45] learns a temporal graph for each regime. However, they focus initially on inferring only time-lagged relationships and require prior knowledge of the number of regimes and transitions between them. [5] addresses first-order regime-dependent causal discovery from MTS with multiple regimes. They proved that first-order Markov switching models with non-linear Gaussian transitions are identifiable up to permutations. Their work offers also a practical algorithms for regime-dependent causal discovery in time series data. However, its primary limitation is the assumption of solely time-lagged relationships, with the theory being restricted to a single time lag. CASTOR [39] learns regime indices and their corresponding causal graphs, including instantaneous and time-lagged relationships, under the assumption of normally distributed noise with equivariance. However, like other causal discovery methods for non-stationary MTS, it does not address non-Gaussian or heteroscedastic noise. [54] tackle a different setting in which they aim to discover a mixtures of Structural Causal Models from a datasets of MTS. They assume that they have different stationary MTS in the same dataset, one regime per MTS, and each one could be explained by one causal model in the mixture. In their case, they assume that every MTS in the dataset is stationary but the whole dataset is a mixture. In our case, we assume that we have only one non-stationary MTS and it is composed of different regimes where we do not know when the regime starts and ends, and our goal is to identify the regimes and the corresponding causal graphs.

# B  Expectation step: Derivation, intuition and illustration

$$\beta_t^r = p\left(z_r = 1 \mid \boldsymbol{x}_t, \boldsymbol{x}_{<t}, \boldsymbol{G}_{\{0:L\}}^r\right)$$
$$= \frac{p\left(z_t = r\right) p\left(\boldsymbol{x}_t \mid \boldsymbol{x}_{<t}, z_t = r, \mathcal{G}^r\right)}{\sum_{j=1}^{N_w} p\left(z_t = j\right) p\left(\boldsymbol{x}_t \mid \boldsymbol{x}_{<t}, z_t = j, \mathcal{G}^j\right)}$$
$$\propto \pi_t(\boldsymbol{\omega}^r) p\left(\boldsymbol{x}_t \mid \boldsymbol{x}_{<t}, z_t = r, \mathcal{G}^r\right),$$

where $p(\boldsymbol{x}_t \mid \boldsymbol{x}_{<t}, z_t = r, \mathcal{G}^r)$ denotes the likelihood of $\boldsymbol{x}_t$ being generated by the SEM from Equation (1) for regime $r$. This probability is computed using the normalizing flows trained during the M-step, following the same reasoning as in Eq(3).

The probability of $\boldsymbol{x}_t$ belonging to regime $r$ is influenced by two main factors: the observation's position within its current regime and whether that regime is designated as pure or impure. In order to clarify the intuition behind pure and impure regimes, Figure 6 shows an example of such case. It presents an initialization of three equal windows while the MTS is composed of two ground truth regimes presented by green color $\mathcal{I}_1^*$ and red one $\mathcal{I}_2^*$. In such case, the regimes $\mathcal{I}_1$ and $\mathcal{I}_3$ are pure, because they are composed of samples coming from the same ground truth regime ($\mathcal{I}_1^*$ for the regime $\mathcal{I}_1$ and $\mathcal{I}_2^*$ for the regime $\mathcal{I}_3$), while $\mathcal{I}_2$ is an impure regime and has samples from the two ground truth regimes.

We highlight all the different cases for a sample $\boldsymbol{x}_t$ either near to the border or not of regime $r$ and also either the causal graph learned for $r$ is on pure or impure data:

**Case 1:** If $\boldsymbol{x}_t$ is in a pure regime $r$ and is far from the boundary in the current iteration, $\pi_t(\boldsymbol{\omega}^r)$ takes a high value (for example, $\pi_{t \in [0,1000]}(\boldsymbol{\omega}^1)$ in Figure 3). Because regime $r$ was trained on pure data, its causal graph is more accurate, leading to a high likelihood $p(\boldsymbol{x}_t \mid \boldsymbol{x}_{<t}, z_t = r, \mathcal{G}^r)$. Consequently, $\beta_t^r \propto \pi_t(\boldsymbol{\omega}^r)\, p(\boldsymbol{x}_t \mid \boldsymbol{x}_{<t}, z_t = r, \mathcal{G}^r)$ remains dominant, causing $\boldsymbol{x}_t$ to stay in regime $r$ at the next iteration.

**Case 2:** If $\boldsymbol{x}_t$ is in a pure regime $r$ but is near the boundary in the current iteration, $\pi_t(\boldsymbol{\omega}^r)$ and $\pi_t(\alpha_{u+1})$ are nearly equal (e.g., $\pi_{t \in [1100,1500]}(\boldsymbol{\omega}^1)$ vs. $\pi_{t \in [1100,1500]}(\boldsymbol{\omega}^2)$ in Figure 3)). Nonetheless, since regime $r$ was learned from pure data, $p(\boldsymbol{x}_t \mid \boldsymbol{x}_{<t}, z_t = r, \mathcal{G}^r)$ stays high, keeping $\beta_t^r$ at its maximum value and maintaining $\boldsymbol{x}_t$ in regime $r$ for the next iteration.

**Case 3:** If $\boldsymbol{x}_t$ is in an impure regime $r+1$ near the boundary during the current iteration, $\pi_t(\boldsymbol{\omega}^r)$ and $\pi_t(\boldsymbol{\omega}^{r+1})$ are also close in value (e.g., $\pi_{t \in [1501,1800]}(\boldsymbol{\omega}^1)$ vs. $\pi_{t \in [1501,1800]}(\boldsymbol{\omega}^2)$ in Figure 3). However, because the causal graph for regime $r$ is more reliable (having been derived from pure data), $p(\boldsymbol{x}_t \mid \boldsymbol{x}_{<t}, z_t = r, \mathcal{G}^r) > p(\boldsymbol{x}_t \mid \boldsymbol{x}_{<t}, z_t = r+1, \mathcal{G}^{r+1})$. As a result, $\boldsymbol{x}_t$ moves from regime $r+1$ to $r$ in the next iteration.

**Case 4:** $\boldsymbol{x}_t$ belongs to impure regime $r+1$ and is far from the border (e.g., $t \in [1801, 2500]$ in Figure 3). In this case, it's uncertain whether $\boldsymbol{x}_t$ will switch regimes in the next iteration. However, as the pure regime $r$ expands with each iteration, $\boldsymbol{x}_t$ will eventually be near the border of regime $r+1$, bringing us back to Case 3.

## C   Maximization step:

### C.1   Mathematical Derivation: Equation (2) to (6)

We perform the maximization of ELBO presented in proposition 3.1, using a BEM procedure, where we alternate between E-step (updating posterior probabilities while fixing all the parameters $\Theta = \{\theta^r, \phi^r, \omega^r\}_{r=1}^{N_w}$) and M step (Updating the parameters while using the posteriors learned in the E-step), we can summarize the process as follows:

- In the Estep: we learn $\beta_t^r = p\left(z_t \mid x_t, x_{<t}, \Theta_{\text{old}}\right)$, where $\Theta_{\text{old}}$ is $\Theta$ of the previous iteration.

- In the M-step: we fix the learned posterior probabilities to the values $\beta_t^r$ and we update $\Theta$.

- By fixing the posterior probabilities in the M-step, the entropy of these probabilities $H\left(p\left(z_t \mid x_t, x_{<t}, \Theta_{\text{old}}\right)\right)$ is a constant, which allow us to discard it.

The detail derivation from Eq.(2) to Eq.(6) is the following:

$$
\begin{aligned}
\text{ELBO}(\Theta) = & \sum_{t=1}^{|T|} \mathbb{E}_{q_\phi(\mathcal{G})} \Big[ \mathbb{E}_{p(z_t \mid x_t, x_{<t})} \left( \log p_{\theta_{z_t}} \left( x_t \mid x_{<t}, \mathcal{G}^{z_t} \right) + \log p\left( z_t \right) \right) + H\left( p\left( z_t \mid x_t, x_{<t} \right) \right) \Big] \\
& + \sum_{r=1}^{N_w} \mathbb{E}_{q_{\phi_r}(\mathcal{G}^r)} \left[ \log p\left( \mathcal{G}^r \right) \right] + H\left( q_{\phi_r}\left( \mathcal{G}^r \right) \right) \\
= & \sum_{t=1}^{|T|} \mathbb{E}_{q_\phi(\mathcal{G})} \Bigg[ \sum_{z_t} p\left( z_t \mid x_t, x_{<t}, \Theta_{\text{old}} \right) \left( \log p_{\theta_{z_t}} \left( x_t \mid x_{<t}, \mathcal{G}^{z_t} \right) + \log p\left( z_t \right) \right) \Bigg] \\
& + \underbrace{H\left( p\left( z_t \mid x_t, x_{<t}, \Theta_{\text{old}} \right) \right)}_{Cte} + \sum_{r=1}^{N_w} \mathbb{E}_{q_{\phi_r}(\mathcal{G}^r)} \left[ \log p\left( \mathcal{G}^r \right) \right] + H\left( q_{\phi_r}\left( \mathcal{G}^r \right) \right)
\end{aligned}
\tag{9}
$$

We replace $p\left( z_t \mid x_t, x_{<t}, \Theta_{\text{old}} \right)$ by $\beta_t^r$, $p\left( z_t \right)$ by our prior choice $\pi_t\left( \omega^r \right)$, and we discard the entropy of the posterior because it is a constant in these steps. Hence, we got the Eq.(6):

$$
\begin{aligned}
\text{ELBO}(\Theta) = & = \mathbb{E}_{q_\phi(\mathcal{G})} \Bigg[ \sum_{t=1}^{|T|} \sum_{r=1}^{N_w} \beta_t^r \left( \log p_{\theta_r} \left( x_t \mid x_{<t}, \mathcal{G}^r \right) + \log \pi_t\left( \omega^r \right) \right) \Bigg] \\
& + \sum_{r=1}^{N_w} \mathbb{E}_{q_{\phi_r}(\mathcal{G}^r)} \left[ \log p\left( \mathcal{G}^r \right) \right] + H\left( q_{\phi_r}\left( \mathcal{G}^r \right) \right).
\end{aligned}
\tag{10}
$$

After the maximization of Eq.(6), FANTOM update the posteriors $p\left( z_t \mid x_t, x_{<t}, \theta_{\text{old}} \right)$ following Eq.(7).

## C.2 Variational Inference Details

We provide the detailed formulations for $q_{\phi^r}\left( \mathcal{G}^r \right)$. In order to model the temporal adjacency matrices $\mathbf{G}_\tau^r$ where $\tau \in [|1:K|]$, we use two learnable matrices $\mathbf{U}_\tau, \mathbf{Q}_\tau \in \mathbf{R}^{d \times d}$ such that:

$$
p_{k,ij} = \frac{\exp\left( u_{k,ij} \right)}{\exp\left( u_{k,ij} \right) + \exp\left( q_{k,ij} \right)}
\tag{11}
$$

For instantaneous graphs $\mathbf{G}_0^r$, we used the same trick as in [15, 54], in which we employ three lower triangular learnable matrices $\mathbf{U}_0, \mathbf{Q}_0, \mathbf{E}_0 \in \mathbf{R}^{d \times d}$ to characterise three scenarios: (1) $i \rightarrow j$; (2) $j \rightarrow i$; (3) no edge between them. For node $i > j$:

$$
\begin{aligned}
p(i \rightarrow j) &= \frac{\exp\left( u_{ij} \right)}{\exp\left( u_{ij} \right) + \exp\left( q_{ij} \right) + \exp\left( e_{ij} \right)} \\
p(j \rightarrow i) &= \frac{\exp\left( q_{ij} \right)}{\exp\left( u_{ij} \right) + \exp\left( q_{ij} \right) + \exp\left( e_{ij} \right)} \\
p(\text{ no edge }) &= \frac{\exp\left( e_{ij} \right)}{\exp\left( u_{ij} \right) + \exp\left( q_{ij} \right) + \exp\left( e_{ij} \right)}.
\end{aligned}
\tag{12}
$$

With this formulation, the instantaneous adjacency matrix is free of self-loops, eliminating any length-1 cycles.

## D   Limitations & Risk of spurious causality

FANTOM's performance deteriorates when a regime contains only a handful of samples or is recorded at an extremely low sampling rate. This shortcoming is not surprising, estimating a separate causal graph for each regime is intrinsically difficult in the presence of multiple regimes and heteroscedastic noise. Yet this ability to pinpoint which edges vanish or emerge from one regime to the next is what makes FANTOM valuable in domains such as healthcare and climate science, where regime shifts

carry substantive meaning. Importantly, in realistic settings where each regime offers sufficient data, for example, epileptic seizures that last several minutes at 250 Hz, FANTOM delivers strong results and provides insights unattainable with existing methods.

FANTOM requires a suitable initial segmentation to effectively learn the regime indices. Our ablation studies demonstrate that selecting a reasonable initial window size establishes a basis for accurate regime detection, which is crucial for achieving high performance. However, we highlight a critical limitation regarding the subsequent pruning step: if the initial window is too high (over-pruning), it can drastically reduce the number of initial regimes below the ground truth. This initialization leads to a loss of essential information and a significant deterioration of FANTOM's final performance.

Spurious causality is a practical risk, so we advocate using the learned graphs as decision support rather than as ground truth. Before acting on them, especially in medical or financial settings, practitioners should adopt safeguards: (i) expert vetting of proposed edges and mechanisms; (ii) robustness checks (e.g., stability under resampling or perturbations, alternative specifications); and (iii) interventional validation when feasible.

# E  Data generation and Baselines

## E.1  Synthetic data

We employ the Erdos–Rényi (ER) [34] model with mean degrees of 1 or 2 to generate lagged graphs, and the Barabasi–Albert (BA) [6] model with mean degrees 4 for instantaneous graphs. The maximum number of lags, $L$, is set at 1. We experiment with varying numbers of nodes $\{10, 20, 40\}$ and different numbers of regimes $\{2, 3\}$, each representing diverse causal graphs or mixing functions. The length of each regime is randomly sampled from the set $\{1000, 1500, 2000, 2500, 3000\}$.

- **Heteroscedastic case.** In heteroscedastic settings, noise variance shifts across both variables and observations, making the underlying DAG much harder to recover from data. Given a random set if directed acyclic graphs $\mathcal{G} = (\mathcal{G}^r)_{r \in [|1:K|]}$, we generate observations from the SEMs in Eq 1 as follows:

$$\forall r \in \{1, ..., K\}, \forall t \in \mathcal{I}_r : x_t^i = f^{i,r}\left(\mathbf{Pa}_{\mathcal{G}^r}^i(<t), \mathbf{Pa}_{\mathcal{G}^r}^i(t)\right) + exp(g^{i,r}\left(\mathbf{Pa}_{\mathcal{G}^r}^i(<t), \mathbf{Pa}_{\mathcal{G}^r}^i(t)\right)) \cdot \epsilon_t^{i,r},$$

  where $f^{i,r}, g^{i,r}$ are chosen to be randomly initialized MLPs with one hidden layer of size number of nodes and `tanh` activation functions. $\epsilon_t^{i,r}$ follows either a normal distribution $\mathcal{N}(0,1)$ or a more complex one obtained by transforming samples from a standard Gaussian with an MLP with random weights and `sin` activation function.

- **Homoscedastic non-Gaussian case.** The formulation used to generated the data is:

$$\forall r \in \{1, ..., K\}, \forall t \in \mathcal{I}_r : x_t^i = f^{i,r}\left(\mathbf{Pa}_{\mathcal{G}^r}^i(<t), \mathbf{Pa}_{\mathcal{G}^r}^i(t)\right) + \epsilon_t^{i,r},$$

  where $f^{i,r}$ is a general differentiable non-linear function. The function $f^{i,r}$ is a random combination between a linear transformation and a randomly chosen function from the set: $\{\texttt{Tanh}, \texttt{Exp}\}$. $\epsilon_t^{i,r}$ follows either a Triangular distribution or a more complex one obtained by transforming samples from a standard Gaussian with an MLP with random weights and `sin` activation function.

## E.2  Baselines

**DYNOTEARS [35].** DYNOTEARS formulates causal discovery for multivariate time series through a linear vector autoregressive (VAR) model that simultaneously captures lagged *and* instantaneous causal effects. Its key innovation is the *DAGness* penalty a smooth, continuously differentiable relaxation of the acyclicity constraint optimized via an augmented Lagrangian scheme alongside a mean squared error loss. DYNOTEARS emerges as the special case of FANTOM obtained by setting $K = 1$, using linear component functions $f^{i,1}$, fixing the noise scaling to $g^{i,1} = 1$ and $\epsilon_t^{i,1} \sim \mathcal{N}(0,1)$ in Eq(1). For comparing with this model, we use publicly available package `causalnex`[2].

**PCMCI+ [42].** a scalable two-stage algorithm for time series, enabling the identification of contemporaneous causal connection. As DYNOTEARS, PCMCI+ is a special case of FANTOM, obtained

---

[2]https://causalnex.readthedocs.io/en/latest/

by setting $K = 1$, using linear or non linear component functions $f^{i,1}$, fixing the noise scaling to $g^{i,1} = 1$ and allowing $\epsilon_t^{i,1}$ to follow any distribution. For the comparison, we use publicly available package `Tigramite`[3].

**Rhino [15].** Gong et al. propose the first structural equation models with historically dependent noise, where noise variance depends solely on time-lagged variables, the Rhino's SEM is as follow:

$$x_t^i = f^i \left( \mathbf{Pa}_G^i(< t), \mathbf{Pa}_G^i(t) \right) + g^i \left( \mathbf{Pa}_G^i(< t), \epsilon_t^i \right), \tag{13}$$

where $\epsilon_t^i \sim \mathcal{N}(0, 1)$. Rhino neglects heteroscedasticity, and assumes a single stationary regime governed by a one causal graph. By our SEM proposed in Eq(1), we can recover the Rhino's SEM by setting $K = 1$ and making $g^{i,1}$ a function of only time lagged parents. In Rhino, they took a non linear transformation of normal noise which is equivalent in our case to allowing $\epsilon_t^{i,1}$ to follow any distribution. To compare with Rhino, we used the open package `causica`[4].

**RPCMCI [45].** RPCMCI learns regime indices and time lagged causal relationships from multi-regime MTS. FANTOM's SEM in Eq(1) generalize it. We can recover RPCMCI settings by making $f^{i,r}$ depends only on time lagged relations and $g^{i,r} = 1$. For the comparison, we use publicly available package `Tigramite`[5].

**CASTOR [39].** CASTOR learns number of regimes their indices and also their corresponding DAGs including instantaneous and time lagged causal relationships from multi-regime MTS. But they assume that they only have gaussian noise with equivariance. FANTOM's SEM in Eq(1) generalize it. We can recover CASTOR settings by making $g^{i,r} = 1$ and $\epsilon_t^{i,r} \sim \mathcal{N}(0, 1)$. For the comparison, we use publicly available code `CASTOR`[6].

### E.3 Optimization parameters

**Heteroscedastic settings.** Unless noted (i.e., in the synthetic-data study), we set the model lag to the true value of 1 and allow FANTOM to capture instantaneous effects. The variational posterior $q_{\phi^r}(\mathcal{G}^r)$ is initialized to prefer sparse graphs (edge probability $< 0.5$). Heteroscedastic noise is modeled with conditional normalizing flows (CNFs). Every neural block is a two-layer MLP with 32 hidden units, residual connections, and layer normalization.

Gradients for discrete edges are estimated via the Gumbel–Softmax trick, using a hard forward pass and a soft backward pass with temperature 0.25. All spline flows employ 128 bins, and each transformation uses an embedding dimension equal to the number of nodes.

The sparsity penalty is fixed at $\lambda_s = 50$. For graphs with 10 or 20 nodes we use $\rho = 1$ and $\alpha = 0$, whereas for 40 nodes we set $\rho = 0.001$. Models are optimized with Adam [30] at a learning rate of 0.005. We establish $\zeta = 900$ as the minimum regime duration, and we use 1000 as initial window size.

**Homoscedastic non-Gaussian settings.** Unless noted (i.e., in the synthetic-data study), we set the model lag to the true value of 1 and allow FANTOM to capture instantaneous effects. We use the same parameters as the heteroscedastic settings. The main difference is the use of a composite of affine-spline transformation. All spline flows employ 16 bins, and each transformation uses an embedding dimension equal to the number of nodes.

The sparsity penalty is fixed at $\lambda_s = 5$. For graphs with 10 or 20 nodes we use $\rho = 1$ and $\alpha = 0$, whereas for 40 nodes we set $\rho = 0.001$. Models are optimized with Adam [30] at a learning rate of 0.005.

### E.4 Real world data

#### E.4.1 Causal Chambers data

We use the wind tunnel datasets from Gamella et al. [13], featuring two controllable fans pushing air through a chamber, barometers measuring air pressure at various locations, and a hatch controlling an

---

[3] https://jakobrunge.github.io/tigramite/
[4] https://github.com/microsoft/causica/blob/main/README.md
[5] https://jakobrunge.github.io/tigramite/
[6] https://github.com/arahmani1/CASTOR

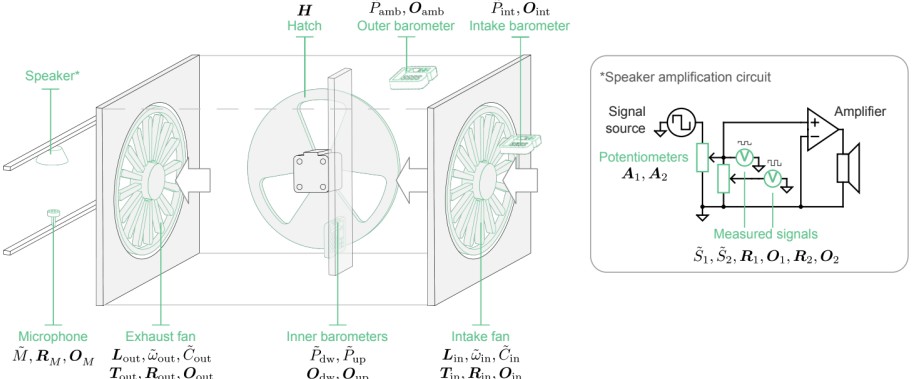

Figure 7: Figure taken from Gamella et al. [13]. Diagrams of the wind tunel causal chamber and its main components, including the amplification circuit that drives the speaker of the wind tunnel. The variables measured by the chamber are displayed in black math print. Sensor measurements are denoted by a tilde. Manipulable variables, that is, actuators and sensor parameters, are shown in bold symbols.

external opening. The tunnel is a chamber with two controllable fans that push air through it and barometers that measure air pressure at different locations. A hatch precisely controls the area of an additional opening to the outside (see Figure 7). The dataset comprises 16 variables: controllable load of the two fans $L_{in}, L_{out}$, their measurable speed ($\tilde{\omega}_{in}$, $\bar{\omega}_{out}$), the current draw by the fans $\left(\tilde{C}_{in}, \tilde{C}_{out}\right)$ ($\tilde{C}_{in}, \tilde{C}_{out}$), the resulting air pressure inside the chamber ($\tilde{P}_{dw}, \tilde{P}_{up}$) or at its intake ($\tilde{P}_{int}$), and the hatch $H$. In the circuit that drives the speaker, we can manipulate the potentiometers ($A_1, A_2$) that control the amplification, monitoring the resulting signal at different points of the circuit $\left(\tilde{S}_1, \tilde{S}_2\right)$ and through the microphone output $(\tilde{M})$. $\tilde{P}_{amb}$ is the ambient pressure measure by the outer barometer.

We evaluate all the models on two regimes of 10,000 samples each: the first is observational, while the second involves soft interventions on five variables:

- $T_{out}$, the resolution of the tachometer timer that measures the elapsed time between successive revolutions of the fan. Choosing microseconds yields a higher resolution in the fan-speed measurement. Hence intervention on $T_{out}$ yields to a change on $\bar{\omega}_{out}$.

- $O_{up}$ the oversampling rates when taking measurements of the current ($\tilde{C}_{in}$, $\tilde{C}_{out}$), amplifier ($\tilde{S}_1, \tilde{S}_2$) and microphone signals $(M)$, and of air pressure at the different barometers $\left(\tilde{P}_{up}, \tilde{P}_{dw}, \tilde{P}_{amb}, \tilde{P}_{int}\right)$.

- $R_{in}$, $R_{out}$, $R_2$ the reference voltages, in volts, of the sensors used to measure the current ($\tilde{C}_{in}$, $\tilde{C}_{out}$), and amplifier ($\tilde{S}_2$), respectively.

For the training procedure, we start by a window of 6000 samples, which gives us three different initial regimes then FANTOM converges smoothly to the exact number of regimes $K = 2$. We set our lag to 8 time lagged and we allow the presence of instantaneous parents. Regarding the parameters, we use a sparsity coefficient equal to 50, spline of 8 bins and MLPs of size 32.

We compare FANTOM to the baselines mentioned in the main text, with results in Table 2. FANTOM is the only model that detects the regime with 99.9% accuracy and outperforms all baselines on the graph learning task, achieving 38.5% on F1 score. Notably, FANTOM surpasses all the models tailored for stationary MTS, even when they are given the ground-truth regime partitions.

### E.4.2   Epilepsy data

Huizenga et al. [22] show that scalp potential fields are contaminated by heteroscedastic noise in EEG measurements. We evaluate FANTOM's performance in detecting epileptic regimes using EEG

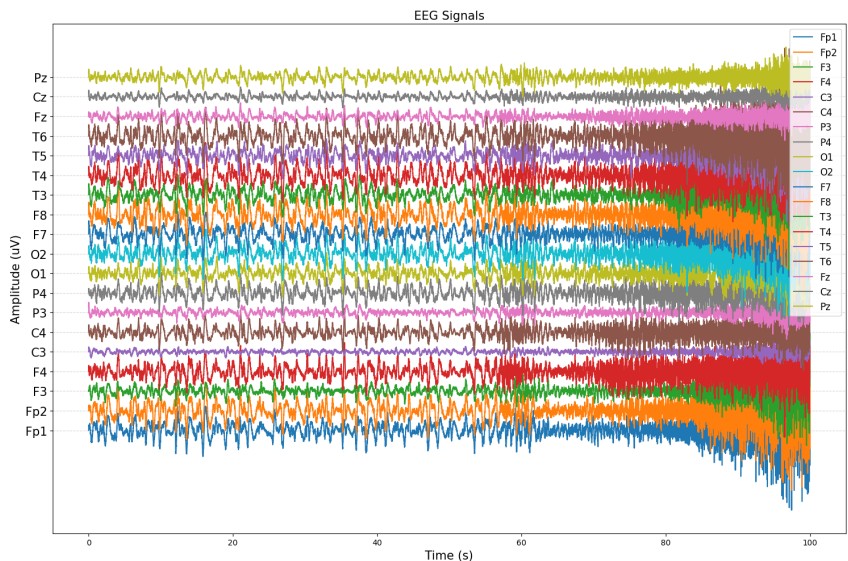

Figure 8: EEG signals for patient of id 7170, session 1 in the TUSZ data.

signals from 10 different patients in the Temple University Hospital EEG Seizure Corpus (TUSZ) dataset [52]. The dataset encompasses multiple seizure types; in this study, we focus on generalized seizures, which engage the entire brain. Each patient's record contains scalp EEG signals from 19 channels (Figure 8), each considered a causal variable.

State-of-the-art methods [52] use graph neural networks (GNNs) for seizure detection, typically building a fixed distance graph and feeding Fast Fourier Transform (FFT) coefficients from 10–12 seconds EEG windows as node features. These approaches (i) train a single model for all patients, offering no personalization; (ii) cannot operate in zero-shot or unsupervised settings; and (iii) reuse an identical graph for both seizure and normal periods. FANTOM addresses these limitations: it detects seizures in a personalized manner without any training data and learns distinct temporal causal graphs for normal and seizure states.

**Preprocessing.** We apply FANTOM to 10 different patients from TUSZ dataset, we treat this as an unsupervised regime detection problem, analyzing roughly 100 seconds of recordings at a 250 Hz sampling rate for each patient, capturing both normal and seizure states. Before running FANTOM, we filter out the EEG signals by a band pass filter of order 6, the lower frequency is 0.5Hz while the highest frequency is 50Hz.

Table 3: Seizure detection accuracy using FANTOM for 10 different patients

| Patient id | Regime Acc |
|------------|------------|
| 0002 | 84.8 |
| 0021 | 80.8 |
| 0302 | 76.6 |
| 0492 | 86.6 |
| 6440 | 86.1 |
| 6520 | 80.2 |
| 7128 | 94.1 |
| 7170 | 81.1 |
| 7936 | 82.0 |
| 8303 | 75.1 |
| Avg | $82.7_{\pm 5.2}$ |

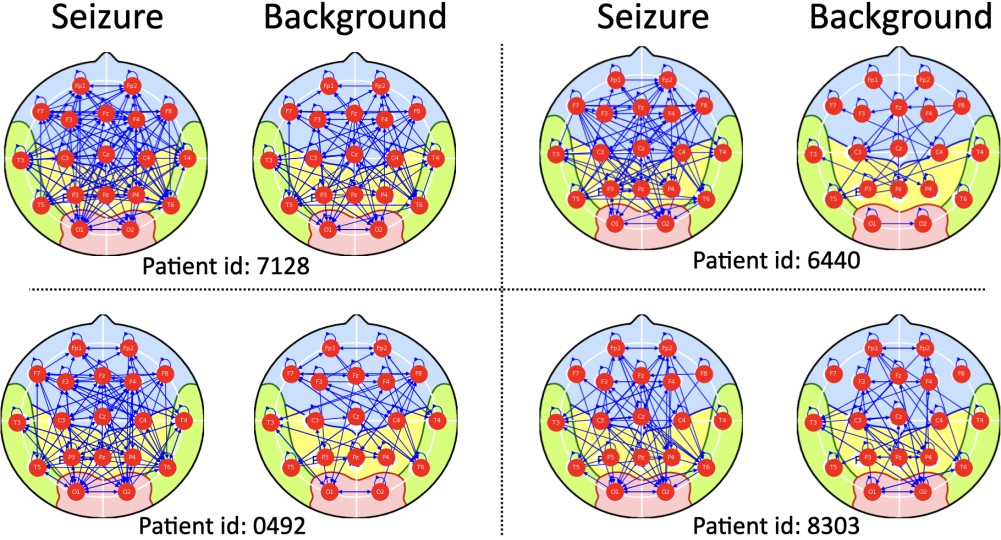

| Seizure | Background | Seizure | Background |

Patient id: 7128 — Patient id: 6440

Patient id: 0492 — Patient id: 8303

Figure 9: Figure illustrates the summary causal graph per regime learned by FANTOM for different patients

We segment each EEG recording into fixed 12 s windows (3000 samples). FANTOM is initialized with eight initial regimes and reliably converges to the two ground-truth states; quantitative results appear in Table 3. We employ a temporal lag of eight samples and allow instantaneous parental links. Averaged over all patients, FANTOM achieves 82.7% regime-assignment accuracy. The graph learned for the seizure state is substantially denser and more interconnected than that of the normal state (Figure 9), consistent with the widespread neural involvement of generalized seizures.

# F   Additional Experiments

## F.1   Ablation studies

### F.1.1   Impact of Heteroscedastic Modeling and Flow Architecture

To assess the contribution of heteroscedastic modelling and the flow architecture, we ran an ablation on MTS comparing three FANTOM variants: (i) Gaussian output (no flow, FANTOM-Gauss), (ii) a simple coupling-spline flow (FANTOM-Spline), and (iii) the full model with a conditional normalizing flow (CNF, FANTOM). In the single-regime setting, removing or simplifying the CNF consistently degrades performance. In the multi-regime setting, only the CNF variant remains stable and converges to the two ground truth regimes; the alternatives collapse to one regime. Results are reported in the table below:

Table 4: Importance of heteroscedastic modelling and flow architecture for 10 nodes. - means the method collapses to 1 regime.

| | | Heteroscedastic noise | | | | |
|---|---|---|---|---|---|---|
| K | model | Inst | | Lag | | Regime |
| | | SHD | F1 | SHD | F1 | Accuracy |
| 1 | FANTOM-Gauss | 39 | 0.0 | 44 | 42.5 | - |
| | FANTOM-Spline | 17 | 0.0 | 18 | 0 | - |
| | FANTOM | **0** | **100** | **0** | **100** | - |
| 2 | FANTOM-Gauss | - | - | - | - | - |
| | FANTOM-Spline | - | - | - | - | - |
| | FANTOM | **1** | **97.8** | **1** | **98.9** | **98.6** |

### F.1.2 Robustness to the pruning step, minimum regime duration, and regime initialization

We evaluated robustness to the pruning step, minimum regime duration ($\zeta$), and regime initialization. Across these variations, FANTOM's predictive performance remains stable. We find that changing (window size, $\zeta$) primarily affects runtime, especially at long horizons, without materially impacting graph and regime learning performances. To show efficiency under long-horizon conditions, we used regimes with lengths [2000, 3000, 2000, 2500].

Table 5: Performance of FANTOM with varying hyperparameters (window size and $\zeta$) on 10 node graphs with 2, 3, and 4 regimes.

| | | | | Heteroscedastic noise | | | | |
|---|---|---|---|---|---|---|---|---|
| (window, $\zeta$) | K | Numb | Running | Inst. | | Lag | | Regime |
| | | | | SHD | F1 | SHD | F1 | Acc. |
| (1000,900) | 2 | 4 | 17'8s | 5 | 91.2 | 13 | 85.7 | 96.5 |
| (1500,1200) | 2 | 4 | 10'12s | 1 | 97.9 | 9 | 89.8 | 95.1 |
| (1000,900) | 3 | 4 | 30'33s | 5 | 94.2 | 8 | 94.4 | 96.4 |
| (1500,1200) | 3 | 4 | 16'2s | 3 | 96.7 | 10 | 91.9 | 91.8 |
| (1000,900) | 4 | 4 | 40'2s | 5 | 95.8 | 12 | 92.7 | 92.9 |
| (1500,1200) | 4 | 4 | 25'38s | 3 | 97.5 | 16 | 90.5 | 91.5 |

Table 6: Performance of FANTOM with varying hyperparameters (window size and $\zeta$) on 40 node graphs with 2, 3, and 4 regimes.

| | | | | Heteroscedastic noise | | | | |
|---|---|---|---|---|---|---|---|---|
| (window, $\zeta$) | K | Numb | Running | Inst. | | Lag | | Regime |
| | | | | SHD | F1 | SHD | F1 | Acc. |
| (1000,900) | 2 | 4 | 22'15s | 33 | 81.9 | 27 | 91.9 | 100 |
| (1500,1200) | 2 | 4 | 10'20s | 35 | 83.8 | 25 | 91.7 | 100 |
| (1000,900) | 3 | 4 | 34'25s | 49 | 85.6 | 58 | 88.6 | 99.8 |
| (1500,1200) | 3 | 4 | 15'52s | 43 | 87.3 | 72 | 87.3 | 99.9 |
| (1000,900) | 4 | 5 | 66'30s | 34 | 86.5 | 53 | 91.9 | 99.7 |
| (1500,1200) | 4 | 4 | 21'38s | 32 | 87.1 | 54 | 91.8 | 99.6 |

From the table, FANTOM scales to 40 nodes and converges to four regimes under different initializations. With a window initialization of 1500 and $\zeta = 1200$, it starts with six initial regimes and converges smoothly to four with 99.6% accuracy in 21min38s. With a window initialization of

1000 and $\zeta = 900$, it starts with nine initial regimes and converges to four with 99.7% accuracy in 66min30s. All rebuttal experiments were run on a Tesla V100-SXM2 (26GB). These results indicate that FANTOM scales without requiring large GPU memory.

To test scalability, we ran FANTOM on a 40 node dataset with four regimes and long horizons (lengths 2000, 3000, 2000, 2500). As expected, additional regimes and longer horizons increase runtime. With 40 nodes and two regimes, training finishes in 10 min 20 s, whereas the four regime setting requires 21 min 38 s.

We showed in the table above (Table 5 and 6) that FANTOM is robust to the initialization parameter (window size that impacts directly the initial regime count and the pruning threshold), e.g. in the case of 40 nodes with 4 regimes, with a window initialization of 1500 and $\zeta = 1200$, it starts with six initial regimes and converges smoothly to four in 21min38s. With a window initialization of 1000 and $\zeta = 900$, it starts with nine initial regimes and converges to four with 99.6% accuracy in 66 min 30s.

### F.1.3 Robustness to data standardization

We investigated FANTOM's robustness to data standardization, a common challenge for most optimization-based causal discovery models. The results in Table 7 clearly demonstrate FANTOM's resilience. This robustness stems from its use of Bayesian structure learning to estimate a distribution over plausible graphs and conditional normalizing flows to effectively model complex data distributions. In contrast, models such as CASTOR are sensitive to standardization and even fail when provided with ground-truth regime labels.

Table 7: Robustness of FANTOM to data standardization (10 nodes).

| | | | Heteroscedastic noise | | | | |
|---|---|---|---|---|---|---|---|
| model | K | data | Inst. | | Lag | | Regime |
| | | | SHD | F1 | SHD | F1 | Acc. |
| CASTOR | 2 | Raw | 90 | 26.9 | 28 | 29.7 | x |
| | | standardized | 28 | 0.0 | 33 | 0.0 | x |
| FANTOM | 2 | Raw | 4 | 93.3 | 7 | 90.9 | 97.1 |
| | | standardized | 5 | 91.5 | 7 | 90.9 | 91.1 |
| CASTOR | 3 | Raw | 135 | 24.1 | 135 | 29.6 | x |
| | | standardized | 37 | 0.0 | 48 | 0.0 | x |
| FANTOM | 3 | Raw | 3 | 96.1 | 10 | 91.5 | 92.2 |
| | | standardized | 2 | 97.3 | 9 | 92.3 | 92.2 |

From the Table 7, FANTOM achieves the same results on raw and standardized data while CASTOR fails in both cases even when we give it access to the ground truth regime labels. In the case of standardized data, CASTOR predicts adjacency matrices full of zeros due to its incapability of handling such scenarios.

## F.2 Additional results on synthetic data

### F.2.1 Heteroscedastic noise with different number of nodes and regimes

Table 8: Average SHD, F1 scores, NHD and Ratio for different models with $d = 10$ nodes and $K = 2$ regimes. *Split* denotes whether regime separation is automatic (✓) or manual (×). *Inst.* refers to instantaneous links, and *Lag* to time-lagged edges.

| Model | Split | Inst. | | | | Lag | | | | Regime |
|---|---|---|---|---|---|---|---|---|---|---|
| | | SHD↓ | F1↑ | NHD↓ | Ratio↓ | SHD↓ | F1↑ | NHD↓ | Ratio↓ | Acc. |
| PCMCI+ | × | $34.3_{\pm3.7}$ | $14.2_{\pm2.5}$ | $0.09_{\pm0.0}$ | $0.85_{\pm0.1}$ | $26.0_{\pm4.3}$ | $25.6_{\pm4.8}$ | $0.07_{\pm0.0}$ | $0.74_{\pm0.0}$ | × |
| Rhino | × | $28.0_{\pm5.6}$ | $8.56_{\pm4.9}$ | $0.09_{\pm0.0}$ | $0.92_{\pm0.0}$ | $38.5_{\pm6.3}$ | $58.4_{\pm11.7}$ | $0.13_{\pm0.0}$ | $0.43_{\pm0.0}$ | × |
| DYNOTEARS | × | $58.5_{\pm4.9}$ | $31.2_{\pm2.8}$ | $0.22_{\pm0.0}$ | $0.68_{\pm0.0}$ | $79.5_{\pm3.5}$ | $33.9_{\pm4.4}$ | $0.27_{\pm0.0}$ | $0.66_{\pm0.0}$ | × |
| CASTOR | × | $90.0_{\pm0.0}$ | $28.1_{\pm2.9}$ | $0.38_{\pm0.0}$ | $0.74_{\pm0.0}$ | $90.5_{\pm3.2}$ | $35.6_{\pm3.4}$ | $0.42_{\pm0.0}$ | $0.64_{\pm0.0}$ | × |
| RPCMCI | ✓ | - | - | - | - | - | - | - | - | - |
| CASTOR | ✓ | - | - | - | - | - | - | - | - | - |
| FANTOM | ✓ | $\mathbf{4.67}_{\pm1.5}$ | $\mathbf{91.7}_{\pm3.2}$ | $\mathbf{0.005}_{\pm0.0}$ | $\mathbf{0.04}_{\pm0.0}$ | $\mathbf{11.6}_{\pm1.1}$ | $\mathbf{88.2}_{\pm2.1}$ | $\mathbf{0.02}_{\pm0.0}$ | $\mathbf{0.08}_{\pm0.0}$ | $\mathbf{96.6}_{\pm1.1}$ |

Table 9: Average SHD, F1 scores, NHD and Ratio for different models with $d = 10$ nodes and $K = 3$ regimes. *Split* denotes whether regime separation is automatic (✓) or manual (×). *Inst.* refers to instantaneous links, and *Lag* to time-lagged edges.

| Model | Split | Inst. | | | | Lag | | | | Regime |
|---|---|---|---|---|---|---|---|---|---|---|
| | | SHD↓ | F1↑ | NHD↓ | Ratio↓ | SHD↓ | F1↑ | NHD↓ | Ratio↓ | Acc. |
| PCMCI+ | × | $46.1_{\pm4.7}$ | $11.1_{\pm2.1}$ | $0.05_{\pm0.0}$ | $0.88_{\pm0.0}$ | $46.0_{\pm0.8}$ | $19.0_{\pm3.3}$ | $0.05_{\pm0.0}$ | $0.80_{\pm0.0}$ | × |
| Rhino | × | $44.5_{\pm6.5}$ | $5.11_{\pm1.9}$ | $0.06_{\pm0.0}$ | $0.94_{\pm0.0}$ | $53.5_{\pm1.5}$ | $64.7_{\pm4.6}$ | $0.07_{\pm0.0}$ | $0.35_{\pm0.0}$ | × |
| DYNOTEARS | × | $89.5_{\pm3.5}$ | $31.5_{\pm0.4}$ | $0.14_{\pm0.0}$ | $0.68_{\pm0.0}$ | $118.0_{\pm4.0}$ | $37.5_{\pm1.2}$ | $0.17_{\pm0.0}$ | $0.61_{\pm0.0}$ | × |
| CASTOR | × | $104._{\pm3.7}$ | $23.4_{\pm0.9}$ | $0.19_{\pm0.0}$ | $0.76_{\pm0.0}$ | $133.5_{\pm2.8}$ | $34.8_{\pm1.2}$ | $0.24_{\pm0.0}$ | $0.64_{\pm0.0}$ | × |
| RPCMCI | ✓ | - | - | - | - | - | - | - | - | - |
| CASTOR | ✓ | - | - | - | - | - | - | - | - | - |
| FANTOM | ✓ | $\mathbf{5.67}_{\pm3.3}$ | $\mathbf{93.3}_{\pm4.2}$ | $\mathbf{0.006}_{\pm0.0}$ | $\mathbf{0.06}_{\pm0.0}$ | $\mathbf{12.3}_{\pm6.3}$ | $\mathbf{90.9}_{\pm1.0}$ | $\mathbf{0.012}_{\pm0.0}$ | $\mathbf{0.08}_{\pm0.0}$ | $\mathbf{97.1}_{\pm0.0}$ |

Under heteroscedastic conditions, $d = 10$ node graphs with either two or three regimes, FANTOM outperforms every baseline. Among methods explicitly designed for multi-regime MTS, it is the only one that converges to the true regime partitions, attaining 97.1% (Table 9) in regime detection accuracy. CASTOR and RPCMCI break down in heteroscedastic settings.

To give stationary MTS baselines the best possible chance, we supply them with the ground truth regime partitions. This is done by training the aforementioned models on each pure regime separately (regime governed by the same causal model). Yet, FANTOM still dominates the structure learning task, while learning the number of regime and their indices as well, achieving an F1 of 93.3 % and an NHD of 0.006 (Table 9). Because NHD penalizes every missing, extra, or mis-oriented edge, a value of 0.006 implies that FANTOM not only recovers the graph skeleton but orients edges with high precision.

Table 10: Average SHD, F1 scores, NHD and Ratio for different models with $d = 20$ nodes and $K = 2$ regimes. *Split* denotes whether regime separation is automatic (✓) or manual (×). *Inst.* refers to instantaneous links, and *Lag* to time-lagged edges.

| Model | Split | Inst. | | | | Lag | | | | Regime |
|---|---|---|---|---|---|---|---|---|---|---|
| | | SHD↓ | F1↑ | NHD↓ | Ratio↓ | SHD↓ | F1↑ | NHD↓ | Ratio↓ | Acc. |
| PCMCI+ | × | $84.0_{\pm17.}$ | $39.7_{\pm2.6}$ | $0.03_{\pm0.0}$ | $0.6_{\pm0.0}$ | $42.5_{\pm12.}$ | $76.9_{\pm1.0}$ | $0.14_{\pm0.0}$ | $0.22_{\pm0.0}$ | × |
| Rhino | × | $70.0_{\pm6.0}$ | $1.26_{\pm1.2}$ | $0.04_{\pm0.0}$ | $0.98_{\pm0.0}$ | $188.0_{\pm30.}$ | $26.3_{\pm9.8}$ | $0.14_{\pm0.0}$ | $0.73_{\pm0.0}$ | × |
| DYNOTEARS | × | $221.5_{\pm9.5}$ | $26.9_{\pm1.2}$ | $0.22_{\pm0.0}$ | $0.72_{\pm0.0}$ | $45.0_{\pm7.0}$ | $61.5_{\pm3.9}$ | $0.02_{\pm0.0}$ | $0.38_{\pm0.0}$ | × |
| CASTOR | × | $377.5_{\pm1.5}$ | $14.9_{\pm0.1}$ | $0.41_{\pm0.0}$ | $0.84_{\pm0.0}$ | $379.5_{\pm1.5}$ | $19.1_{\pm1.0}$ | $0.41_{\pm0.0}$ | $0.80_{\pm0.0}$ | × |
| RPCMCI | ✓ | - | - | - | - | - | - | - | - | - |
| CASTOR | ✓ | - | - | - | - | - | - | - | - | - |
| FANTOM | ✓ | $\mathbf{9.00}_{\pm3.0}$ | $\mathbf{89.1}_{\pm5.7}$ | $\mathbf{0.006}_{\pm0.0}$ | $\mathbf{0.10}_{\pm0.0}$ | $\mathbf{26.0}_{\pm5.0}$ | $\mathbf{85.4}_{\pm1.3}$ | $\mathbf{0.01}_{\pm0.0}$ | $\mathbf{0.14}_{\pm0.0}$ | $\mathbf{97.8}_{\pm0.2}$ |

Table 11: Average SHD, F1 scores, NHD and Ratio for different models with $d = 20$ nodes and $K = 3$ regimes. *Split* denotes whether regime separation is automatic (✓) or manual (×). *Inst.* refers to instantaneous links, and *Lag* to time-lagged edges.

| Model | Split | Inst. | | | | Lag | | | | Regime |
|---|---|---|---|---|---|---|---|---|---|---|
| | | SHD↓ | F1↑ | NHD↓ | Ratio↓ | SHD↓ | F1↑ | NHD↓ | Ratio↓ | Acc. |
| PCMCI+ | × | $83.5_{\pm 15.}$ | $39.9_{\pm 1.6}$ | $0.03_{\pm 0.0}$ | $0.59_{\pm 0.0}$ | $38.0_{\pm 2.0}$ | $77.8_{\pm 0.9}$ | $0.01_{\pm 0.0}$ | $0.22_{\pm 0.0}$ | × |
| Rhino | × | $108.5_{\pm 11.}$ | $1.6_{\pm 1.2}$ | $0.02_{\pm 0.0}$ | $0.98_{\pm 0.0}$ | $292.0_{\pm 62.}$ | $25.8_{\pm 8.9}$ | $0.09_{\pm 0.0}$ | $0.74_{\pm 0.0}$ | × |
| DYNOTEARS | × | $325._{\pm 14.}$ | $27.6_{\pm 1.0}$ | $0.14_{\pm 0.0}$ | $0.72_{\pm 0.0}$ | $55.5_{\pm 0.5}$ | $69.8_{\pm 7.6}$ | $0.01_{\pm 0.0}$ | $0.29_{\pm 0.0}$ | × |
| CASTOR | × | $412.3_{\pm 2.5}$ | $13.8_{\pm 0.0}$ | $0.20_{\pm 0.0}$ | $0.86_{\pm 0.0}$ | $570._{\pm 2.0}$ | $17.5_{\pm 0.4}$ | $0.28_{\pm 0.0}$ | $0.81_{\pm 0.0}$ | × |
| RPCMCI | ✓ | - | - | - | - | - | - | - | - | - |
| CASTOR | ✓ | - | - | - | - | - | - | - | - | - |
| FANTOM | ✓ | $\mathbf{13.5}_{\pm 6.5}$ | $\mathbf{88.2}_{\pm 6.2}$ | $\mathbf{0.05}_{\pm 0.0}$ | $\mathbf{0.11}_{\pm 0.0}$ | $\mathbf{46.5}_{\pm 9.5}$ | $\mathbf{84.6}_{\pm 2.5}$ | $\mathbf{0.01}_{\pm 0.0}$ | $\mathbf{0.14}_{\pm 0.0}$ | $\mathbf{97.8}_{\pm 1.4}$ |

Under heteroscedastic conditions, $d = 20$ node graphs with either two or three regimes, FANTOM outperforms all the baselines. Among methods explicitly designed for multi-regime MTS, it is the only one that converges to the true regime partitions, attaining 97.8% in regime detection accuracy (Table 10). CASTOR and RPCMCI break down in heteroscedastic settings, also in the case of 20 nodes.

To give stationary MTS baselines the best possible chance, we supply them with the ground truth regime partitions. This is done by training the aforementioned models on each pure regime separately (regime governed by the same causal model). Yet, FANTOM still dominates the structure learning task, while learning the number of regime and their indices as well, achieving an F1 of 89.1 % and an NHD of 0.006 for instantaneous links and an F1 of 85.4% and and an NHD 0.01% for time lagged (Table 10).

Table 12: Average SHD, F1 scores, NHD and Ratio for different models with $d = 40$ nodes and $K = 2$ regimes. *Split* denotes whether regime separation is automatic (✓) or manual (×). *Inst.* refers to instantaneous links, and *Lag* to time-lagged edges.

| Model | Split | Inst. | | | | Lag | | | | Regime |
|---|---|---|---|---|---|---|---|---|---|---|
| | | SHD↓ | F1↑ | NHD↓ | Ratio↓ | SHD↓ | F1↑ | NHD↓ | Ratio↓ | Acc. |
| PCMCI+ | × | $146.5_{\pm 5.0}$ | $25.8_{\pm 0.1}$ | $0.02_{\pm 0.0}$ | $0.74_{\pm 0.0}$ | $109.0_{\pm 15.}$ | $58.0_{\pm 4.5}$ | $0.01_{\pm 0.0}$ | $0.42_{\pm 0.0}$ | × |
| Rhino | × | $137.0_{\pm 7.1}$ | $0.0_{\pm 0.0}$ | $0.02_{\pm 0.0}$ | $1.00_{\pm 0.0}$ | $700.0_{\pm 87.}$ | $28.2_{\pm 5.7}$ | $0.12_{\pm 0.0}$ | $0.71_{\pm 0.0}$ | × |
| DYNOTEARS | × | $137.5_{\pm 9.2}$ | $17.1_{\pm 1.4}$ | $0.020_{\pm 0.0}$ | $0.82_{\pm 0.0}$ | $129.0_{\pm 15.}$ | $36.6_{\pm 4.9}$ | $0.01_{\pm 0.0}$ | $0.63_{\pm 0.0}$ | × |
| CASTOR | × | $151.0_{\pm 11.}$ | $0.0_{\pm 0.0}$ | $0.03_{\pm 0.0}$ | $1.00_{\pm 0.0}$ | $333.0_{\pm 9.3}$ | $19.6_{\pm 5.6}$ | $0.050_{\pm 0.0}$ | $0.80_{\pm 0.0}$ | × |
| RPCMCI | ✓ | - | - | - | - | - | - | - | - | - |
| CASTOR | ✓ | - | - | - | - | - | - | - | - | - |
| FANTOM | ✓ | $\mathbf{27.00}_{\pm 6.0}$ | $\mathbf{85.6}_{\pm 3.7}$ | $\mathbf{0.005}_{\pm 0.0}$ | $\mathbf{0.14}_{\pm 0.0}$ | $\mathbf{26.5}_{\pm 0.5}$ | $\mathbf{91.9}_{\pm 1.0}$ | $\mathbf{0.004}_{\pm 0.0}$ | $\mathbf{0.08}_{\pm 0.0}$ | $\mathbf{99.9}_{\pm 0.1}$ |

Table 13: Average SHD, F1 scores, NHD and Ratio for different models with $d = 40$ nodes and $K = 3$ regimes. *Split* denotes whether regime separation is automatic (✓) or manual (×). *Inst.* refers to instantaneous links, and *Lag* to time-lagged edges.

| Model | Split | Inst. | | | | Lag | | | | Regime |
|---|---|---|---|---|---|---|---|---|---|---|
| | | SHD↓ | F1↑ | NHD↓ | Ratio↓ | SHD↓ | F1↑ | NHD↓ | Ratio↓ | Acc. |
| PCMCI+ | × | $222.0_{\pm 1.4}$ | $25.2_{\pm 0.4}$ | $0.01_{\pm 0.0}$ | $0.75_{\pm 0.0}$ | $142.0_{\pm 17.}$ | $62.4_{\pm 1.0}$ | $0.01_{\pm 0.0}$ | $0.37_{\pm 0.0}$ | × |
| Rhino | × | $210.5_{\pm 9.2}$ | $0.0_{\pm 0.0}$ | $0.01_{\pm 0.0}$ | $1.00_{\pm 0.0}$ | $1005._{\pm 146.}$ | $29.3_{\pm 5.4}$ | $0.08_{\pm 0.0}$ | $0.7_{\pm 0.0}$ | × |
| DYNOTEARS | × | $203.0_{\pm 14.}$ | $15.2_{\pm 2.2}$ | $0.01_{\pm 0.0}$ | $0.85_{\pm 0.0}$ | $161.0_{\pm 1.4}$ | $51.3_{\pm 15.}$ | $0.01_{\pm 0.0}$ | $0.49_{\pm 0.1}$ | × |
| CASTOR | × | $224.0_{\pm 10.2}$ | $0.00_{\pm 0.0}$ | $0.01_{\pm 0.0}$ | $1.00_{\pm 0.0}$ | $501.0_{\pm 21.}$ | $23.1_{\pm 1.2}$ | $0.03_{\pm 0.0}$ | $0.76_{\pm 0.0}$ | × |
| RPCMCI | ✓ | - | - | - | - | - | - | - | - | - |
| CASTOR | ✓ | - | - | - | - | - | - | - | - | - |
| FANTOM | ✓ | $\mathbf{52.0}_{\pm 9.0}$ | $\mathbf{82.2}_{\pm 5.1}$ | $\mathbf{0.004}_{\pm 0.0}$ | $\mathbf{0.19}_{\pm 0.0}$ | $\mathbf{65.0}_{\pm 7.0}$ | $\mathbf{87.9}_{\pm 0.6}$ | $\mathbf{0.004}_{\pm 0.0}$ | $\mathbf{0.11}_{\pm 0.0}$ | $\mathbf{99.8}_{\pm 0.0}$ |

Under heteroscedastic conditions, $d = 40$ node graphs with either two or three regimes, FANTOM outperforms every baselineand shows that it can scale to large graphs even in this complex setting. Among methods explicitly designed for multi-regime MTS, it is the only one that converges to the true regime partitions, attaining 99.9% in regime detection accuracy. CASTOR and RPCMCI break

down in heteroscedastic settings (Table 12). Large scale graphs helps FANTOM to differentiate between the different regimes and increases the regime detection accuracy by 2% compared to 20 node graphs settings.

To give stationary MTS baselines the best possible chance, we supply them with the ground truth regime partitions. This is done by training the aforementioned models on each pure regime separately (regime governed by the same causal model). Yet, FANTOM still dominates the structure learning task, while learning the number of regime and their indices as well, achieving an F1 of 85.6 % and an NHD of 0.14 (Table 12).

### F.2.2 Non-Gaussian noise with different number of nodes and regimes

Table 14: Average SHD, F1 scores, NHD and Ratio for different models with $d = 10$ nodes and $K = 2$ regimes. *Split* denotes whether regime separation is automatic ($\checkmark$) or manual ($\times$). *Inst.* refers to instantaneous links, and *Lag* to time-lagged edges.

| | | | Inst. | | | | Lag | | | | Regime |
|---|---|---|---|---|---|---|---|---|---|---|---|
| Model | Split | Type | SHD↓ | F1↑ | NHD↓ | Ratio↓ | SHD↓ | F1↑ | NHD↓ | Ratio↓ | Acc. |
| PCMCI+ | × | W | $6.00_{\pm0.0}$ | $88.1_{\pm0.5}$ | $0.01_{\pm0.0}$ | $0.12_{\pm0.0}$ | $8.00_{\pm5.0}$ | $90.7_{\pm6.1}$ | $0.01_{\pm0.0}$ | $0.09_{\pm0.0}$ | × |
| Rhino | × | W | $2.50_{\pm0.5}$ | $96.0_{\pm0.1}$ | $0.004_{\pm0.0}$ | $0.04_{\pm0.0}$ | $\mathbf{4.00}_{\pm1.0}$ | $\mathbf{96.0}_{\pm0.4}$ | $\mathbf{0.006}_{\pm0.0}$ | $\mathbf{0.04}_{\pm0.0}$ | × |
| DYNOTEARS | × | W | $42.0_{\pm11.}$ | $51.8_{\pm7.9}$ | $0.12_{\pm0.0}$ | $0.48_{\pm0.0}$ | $8.00_{\pm1.0}$ | $86.8_{\pm0.5}$ | $0.02_{\pm0.0}$ | $0.12_{\pm0.0}$ | × |
| CASTOR | × | W | $17.0_{\pm3.0}$ | $62.6_{\pm8.5}$ | $0.055_{\pm0.0}$ | $0.37_{\pm0.0}$ | $11.0_{\pm1.0}$ | $85.2_{\pm0.7}$ | $0.02_{\pm0.0}$ | $0.15_{\pm0.0}$ | × |
| RPCMCI | $\checkmark$ | W | - | - | - | - | - | - | - | - | - |
| CASTOR | $\checkmark$ | W | $34.0_{\pm20.}$ | $42.5_{\pm27.}$ | $0.09_{\pm0.0}$ | $0.57_{\pm0.2}$ | $45.0_{\pm31.}$ | $47.0_{\pm25.}$ | $0.13_{\pm0.1}$ | $0.52_{\pm0.2}$ | $77.0_{\pm13.}$ |
| FANTOM | $\checkmark$ | W | $\mathbf{1.00}_{\pm0.0}$ | $\mathbf{98.2}_{\pm0.1}$ | $\mathbf{0.002}_{\pm0.0}$ | $\mathbf{0.01}_{\pm0.0}$ | $8.00_{\pm0.0}$ | $91.0_{\pm4.9}$ | $0.02_{\pm0.0}$ | $0.08_{\pm0.05}$ | $\mathbf{98.6}_{\pm0.2}$ |

| | | | SHD↓ | F1↑ | NHD↓ | Ratio ↓ | Acc. |
|---|---|---|---|---|---|---|---|
| CD-NOD | × | S | $33.0_{\pm3.0}$ | $47.0_{\pm5.9}$ | $0.41_{\pm0.0}$ | $0.52_{\pm0.0}$ | × |
| FANTOM | $\checkmark$ | S | $\mathbf{7.5}_{\pm2.5}$ | $\mathbf{93.1}_{\pm2.2}$ | $\mathbf{0.07}_{\pm0.0}$ | $\mathbf{0.06}_{\pm0.0}$ | $\mathbf{99.6}_{\pm0.0}$ |

Table 15: Average SHD, F1 scores, NHD and Ratio for different models with $d = 10$ nodes and $K = 3$ regimes. *Split* denotes whether regime separation is automatic ($\checkmark$) or manual ($\times$). *Inst.* refers to instantaneous links, and *Lag* to time-lagged edges.

| | | | Inst. | | | | Lag | | | | Regime |
|---|---|---|---|---|---|---|---|---|---|---|---|
| Model | Split | Type | SHD↓ | F1↑ | NHD↓ | Ratio↓ | SHD↓ | F1↑ | NHD↓ | Ratio↓ | Acc. |
| PCMCI+ | × | W | $17.5_{\pm2.1}$ | $74.9_{\pm5.0}$ | $0.02_{\pm0.0}$ | $0.24_{\pm0.0}$ | $14.5_{\pm2.1}$ | $88.3_{\pm1.6}$ | $0.01_{\pm0.0}$ | $0.11_{\pm0.0}$ | × |
| Rhino | × | W | $2.50_{\pm0.7}$ | $96.8_{\pm1.3}$ | $0.002_{\pm0.0}$ | $0.03_{\pm0.0}$ | $\mathbf{6.00}_{\pm1.4}$ | $95.2_{\pm1.6}$ | $\mathbf{0.006}_{\pm0.0}$ | $\mathbf{0.04}_{\pm0.0}$ | × |
| DYNOTEARS | × | W | $42.0_{\pm33.}$ | $54.4_{\pm11.2}$ | $0.06_{\pm0.0}$ | $0.45_{\pm0.1}$ | $21.0_{\pm1.4}$ | $82.1_{\pm1.6}$ | $0.02_{\pm0.0}$ | $0.17_{\pm0.0}$ | × |
| CASTOR | × | W | $22.0_{\pm2.8}$ | $66.2_{\pm2.5}$ | $0.030_{\pm0.0}$ | $0.33_{\pm0.0}$ | $17.0_{\pm4.2}$ | $84.4_{\pm1.8}$ | $0.01_{\pm0.0}$ | $0.15_{\pm0.0}$ | × |
| RPCMCI | $\checkmark$ | W | - | - | - | - | - | - | - | - | - |
| CASTOR | $\checkmark$ | W | $47.0_{\pm17.}$ | $34.8_{\pm30.}$ | $0.05_{\pm0.0}$ | $0.65_{\pm0.2}$ | $59.5_{\pm31.}$ | $39.4_{\pm19.}$ | $0.07_{\pm0.0}$ | $0.60_{\pm0.2}$ | $51.6_{\pm8.9}$ |
| FANTOM | $\checkmark$ | W | $\mathbf{0.33}_{\pm0.4}$ | $\mathbf{99.5}_{\pm0.6}$ | $\mathbf{0.00}_{\pm0.0}$ | $\mathbf{0.00}_{\pm0.0}$ | $12.5_{\pm6.8}$ | $89.1_{\pm3.7}$ | $0.01_{\pm0.0}$ | $0.10_{\pm0.0}$ | $\mathbf{99.4}_{\pm0.1}$ |

| | | | SHD↓ | F1↑ | NHD↓ | Ratio ↓ | Acc. |
|---|---|---|---|---|---|---|---|
| CD-NOD | × | S | $42.5_{\pm2.5}$ | $31.8_{\pm1.1}$ | $0.42_{\pm0.0}$ | $0.67_{\pm0.0}$ | × |
| FANTOM | $\checkmark$ | S | $\mathbf{4.5}_{\pm2.0}$ | $\mathbf{95.6}_{\pm1.5}$ | $\mathbf{0.04}_{\pm0.0}$ | $\mathbf{0.04}_{\pm0.0}$ | $\mathbf{96.6}_{\pm0.2}$ |

Under homoscedastic, non-Gaussian noise with $d = 10$ node graphs and either two or three regimes, FANTOM outperforms every baseline on instantaneous-link inference, reaching an F1 of 98.2 % and an NHD of 0.002. Among methods explicitly designed for multi-regime MTS, both FANTOM and CASTOR recover the exact number of regimes, whereas RPCMCI fails to converge to the true partitions. FANTOM further surpasses CASTOR in regime detection (98.6 % vs. 77.0 %) and in DAG learning (F1 = 98.2 % vs. 42.5 %).

To give the stationary-MTS baselines their best chance, we supply them with the ground-truth regime labels and train each model on the corresponding pure regime. Even in this favorable setting, only Rhino exceeds FANTOM on time-lagged links, achieving an F1 of 96.0 % compared with FANTOM's 91.0 %.

Table 16: Average SHD, F1 scores, NHD and Ratio for different models with $d = 20$ nodes and $K = 2$ regimes. *Split* denotes whether regime separation is automatic (✓) or manual (×). *Inst.* refers to instantaneous links, and *Lag* to time-lagged edges.

| | | | Inst. | | | | Lag | | | | Regime |
|---|---|---|---|---|---|---|---|---|---|---|---|
| Model | Split | Type | SHD↓ | F1↑ | NHD↓ | Ratio↓ | SHD↓ | F1↑ | NHD↓ | Ratio↓ | Acc. |
| PCMCI+ | × | W | $46.0_{\pm 2.8}$ | $54.9_{\pm 0.6}$ | $0.03_{\pm 0.0}$ | $0.45_{\pm 0.0}$ | $17.0_{\pm 4.2}$ | $88.7_{\pm 0.5}$ | $0.009_{\pm 0.0}$ | $0.11_{\pm 0.0}$ | × |
| Rhino | × | W | $17.5_{\pm 14.}$ | $82.5_{\pm 16.}$ | $0.007_{\pm 0.0}$ | $0.17_{\pm 0.1}$ | $29.5_{\pm 3.5}$ | $83.5_{\pm 2.1}$ | $0.01_{\pm 0.0}$ | $0.16_{\pm 0.0}$ | × |
| DYNOTEARS | × | W | $44.5_{\pm 3.5}$ | $44.9_{\pm 3.1}$ | $0.03_{\pm 0.0}$ | $0.55_{\pm 0.0}$ | $46.5_{\pm 12.}$ | $55.8_{\pm 5.2}$ | $0.025_{\pm 0.0}$ | $0.44_{\pm 0.0}$ | × |
| CASTOR | × | W | $139.5_{\pm 13.}$ | $41.8_{\pm 5.3}$ | $0.10_{\pm 0.0}$ | $0.58_{\pm 0.0}$ | $186.0_{\pm 28.}$ | $40.1_{\pm 2.3}$ | $0.13_{\pm 0.0}$ | $0.60_{\pm 0.0}$ | × |
| RPCMCI | ✓ | W | - | - | - | - | - | - | - | - | - |
| CASTOR | ✓ | W | $68.0_{\pm 8.5}$ | $37.2_{\pm 12.}$ | $0.04_{\pm 0.0}$ | $0.62_{\pm 0.1}$ | $86.0_{\pm 9.9}$ | $37.4_{\pm 7.8}$ | $0.06_{\pm 0.0}$ | $0.64_{\pm 0.0}$ | $46.6_{\pm 19.}$ |
| FANTOM | ✓ | W | $\mathbf{3.50}_{\pm 1.5}$ | $\mathbf{97.2}_{\pm 1.2}$ | $\mathbf{0.002}_{\pm 0.0}$ | $\mathbf{0.02}_{\pm 0.0}$ | $\mathbf{11.5}_{\pm 2.5}$ | $\mathbf{93.1}_{\pm 1.6}$ | $\mathbf{0.006}_{\pm 0.0}$ | $\mathbf{0.06}_{\pm 0.0}$ | $\mathbf{100.}_{\pm 0.0}$ |
| | | | SHD↓ | | F1↑ | | NHD↓ | | Ratio ↓ | | Acc. |
| CD-NOD | × | S | $106._{\pm 4.0}$ | | $26.0_{\pm 4.1}$ | | $0.31_{\pm 0.0}$ | | $0.73_{\pm 0.0}$ | | × |
| FANTOM | ✓ | S | $\mathbf{4.0}_{\pm 0.0}$ | | $\mathbf{98.3}_{\pm 0.0}$ | | $\mathbf{0.01}_{\pm 0.0}$ | | $\mathbf{0.01}_{\pm 0.0}$ | | $\mathbf{100.}_{\pm 0.0}$ |

Under homoscedastic, non-Gaussian noise with $d = 20$ node graphs and two regimes, FANTOM outperforms every baseline on instantaneous-link and time lagged link inference, reaching an F1 of 97.2 % and an NHD of 0.002 for instantaneous links and an F1 of 93.1% on time lagged relationships. Among methods explicitly designed for multi-regime MTS, both FANTOM and CASTOR recover the exact number of regimes, whereas RPCMCI fails to converge to the true partitions. FANTOM further surpasses CASTOR in regime detection (100. % vs. 46.6 %) and in DAG learning (F1 = 97.2 % vs. 37.2 %).

To give the stationary-MTS baselines their best chance, we supply them with the ground-truth regime labels and train each model on the corresponding pure regime. Even in this favorable setting, FANTOM out performs all the baselines, achieving an F1 of 97.2 %.

Table 17: Average SHD, F1 scores, NHD and Ratio for different models with $d = 20$ nodes and $K = 3$ regimes. *Split* denotes whether regime separation is automatic (✓) or manual (×). *Inst.* refers to instantaneous links, and *Lag* to time-lagged edges.

| | | | Inst. | | | | Lag | | | | Regime |
|---|---|---|---|---|---|---|---|---|---|---|---|
| Model | Split | Type | SHD↓ | F1↑ | NHD↓ | Ratio↓ | SHD↓ | F1↑ | NHD↓ | Ratio↓ | Acc. |
| PCMCI+ | × | W | $66.5_{\pm 2.1}$ | $55.1_{\pm 2.5}$ | $0.02_{\pm 0.0}$ | $0.45_{\pm 0.0}$ | $\mathbf{24.5}_{\pm 12.}$ | $\mathbf{88.9}_{\pm 3.0}$ | $\mathbf{0.007}_{\pm 0.0}$ | $\mathbf{0.11}_{\pm 0.0}$ | × |
| Rhino | × | W | $27.5_{\pm 14.}$ | $82.1_{\pm 11.}$ | $0.007_{\pm 0.0}$ | $0.17_{\pm 0.1}$ | $50.5_{\pm 3.5}$ | $81.8_{\pm 1.4}$ | $0.01_{\pm 0.0}$ | $0.18_{\pm 0.0}$ | × |
| DYNOTEARS | × | W | $69.0_{\pm 4.2}$ | $43.6_{\pm 3.7}$ | $0.02_{\pm 0.0}$ | $0.56_{\pm 0.0}$ | $54.5_{\pm 10.6}$ | $66.3_{\pm 2.4}$ | $0.01_{\pm 0.0}$ | $0.34_{\pm 0.0}$ | × |
| CASTOR | × | W | $167.0_{\pm 9.9}$ | $38.3_{\pm 4.6}$ | $0.05_{\pm 0.0}$ | $0.61_{\pm 0.0}$ | $264.5_{\pm 34.}$ | $41.2_{\pm 1.5}$ | $0.08_{\pm 0.0}$ | $0.58_{\pm 0.0}$ | × |
| RPCMCI | ✓ | W | - | - | - | - | - | - | - | - | - |
| CASTOR | ✓ | W | $121.5_{\pm 27.}$ | $34.3_{\pm 9.6}$ | $0.03_{\pm 0.0}$ | $0.65_{\pm 0.1}$ | $181.5_{\pm 17.}$ | $33.7_{\pm 7.8}$ | $0.05_{\pm 0.0}$ | $0.66_{\pm 0.0}$ | $79.8_{\pm 4.3}$ |
| FANTOM | ✓ | W | $\mathbf{7.00}_{\pm 0.0}$ | $\mathbf{96.1}_{\pm 0.0}$ | $\mathbf{0.001}_{\pm 0.0}$ | $\mathbf{0.03}_{\pm 0.0}$ | $45.5_{\pm 23.0}$ | $85.2_{\pm 6.1}$ | $0.01_{\pm 0.0}$ | $0.14_{\pm 0.0}$ | $\mathbf{100.}_{\pm 0.0}$ |
| | | | SHD↓ | | F1↑ | | NHD↓ | | Ratio ↓ | | Acc. |
| CD-NOD | × | S | $133._{\pm 1.5}$ | | $31.5_{\pm 1.8}$ | | $0.43_{\pm 0.0}$ | | $0.61_{\pm 0.0}$ | | × |
| FANTOM | ✓ | S | $\mathbf{6.0}_{\pm 1.0}$ | | $\mathbf{98.2}_{\pm 0.2}$ | | $\mathbf{0.01}_{\pm 0.0}$ | | $\mathbf{0.01}_{\pm 0.0}$ | | $\mathbf{100.}_{\pm 0.0}$ |

Under homoscedastic, non-Gaussian noise with $d = 20$ node graphs and three regimes, FANTOM outperforms every baseline on instantaneous-link inference, reaching an F1 of 96.1 % and an NHD of 0.001. Among methods explicitly designed for multi-regime MTS, both FANTOM and CASTOR recover the exact number of regimes, whereas RPCMCI fails to converge to the true partitions. FANTOM further surpasses CASTOR in regime detection (100. % vs. 79.8 %) and in DAG learning (F1 = 96.1 % vs. 34.3 %).

To give the stationary-MTS baselines their best chance, we supply them with the ground-truth regime labels and train each model on the corresponding pure regime. For this scenario of 20 nodes and 3 regimes and benefiting from regime labels, PCMCI+ exceeds FANTOM and Rhino on time-lagged links, achieving an F1 of 88.9 % compared with FANTOM's 85.2%.

Table 18: Average SHD, F1 scores, NHD and Ratio for different models with $d = 40$ nodes and $K = 2$ regimes. *Split* denotes whether regime separation is automatic (✓) or manual (×). *Inst.* refers to instantaneous links, and *Lag* to time-lagged edges.

| Model | Split | Type | Inst. | | | | Lag | | | | Regime |
|---|---|---|---|---|---|---|---|---|---|---|---|
| | | | SHD↓ | F1↑ | NHD↓ | Ratio↓ | SHD↓ | F1↑ | NHD↓ | Ratio↓ | Acc. |
| PCMCI+ | × | W | 91.0±2.8 | 56.7±0.1 | 0.01±0.0 | 0.43±0.0 | **25.0**±1.4 | **91.9**±0.8 | **0.003**±0.0 | **0.08**±0.0 | × |
| Rhino | × | W | **24.5**±2.1 | **90.7**±0.9 | **0.004**±0.0 | **0.09**±0.0 | 34.5±5.0 | 89.8±0.9 | 0.005±0.0 | 0.10±0.0 | × |
| DYNOTEARS | × | W | 100.0±5.7 | 40.8±3.7 | 0.015±0.0 | 0.59±0.0 | 58.5±17. | 76.4±5.4 | 0.009±0.0 | 0.21±0.0 | × |
| CASTOR | × | W | 94.0±46. | 67.3±5.0 | 0.010±0.0 | 0.33±0.0 | 100.0±49. | 78.6±3.4 | 0.01±0.0 | 0.21±0.0 | × |
| RPCMCI | ✓ | W | - | - | - | - | - | - | - | - | - |
| CASTOR | ✓ | W | - | - | - | - | - | - | - | - | - |
| FANTOM | ✓ | W | 29.5±1.5 | 88.1±0.2 | 0.004±0.0 | 0.11±0.0 | 32.3±0.5 | 90.8±0.2 | 0.005±0.0 | 0.08±0.0 | **100.**±0.0 |
| | | | SHD↓ | | F1↑ | | NHD↓ | | Ratio ↓ | | Acc. |
| CD-NOD | × | S | 260.±5.0 | | 17.8±3.6 | | 0.18±0.0 | | 0.82±0.0 | | × |
| FANTOM | ✓ | S | **39.5**±4.5 | | **92.4**±0.6 | | **0.02**±0.0 | | **0.07**±0.0 | | **100.**±0.0 |

Under homoscedastic, non-Gaussian noise with $d = 40$-node graphs and two regimes, FANTOM is the only method that recovers the correct number of regimes, whereas CASTOR and RPCMCI fail to converge. FANTOM achieves 100% regime-detection accuracy and an 88.1% F1 score in DAG recovery.

For a fair comparison with stationary-MTS baselines, we provide these models with the ground-truth regime labels and train them separately on each pure regime. In this advantaged setting, Rhino surpasses FANTOM on instantaneous links (F1 = 90.7% vs. 88.1%), while PCMCI+ leads on time-lagged links (F1 = 91.9% vs. 90.8%).

Table 19: Average SHD, F1 scores, NHD and Ratio for different models with $d = 40$ nodes and $K = 3$ regimes. *Split* denotes whether regime separation is automatic (✓) or manual (×). *Inst.* refers to instantaneous links, and *Lag* to time-lagged edges.

| Model | Split | Type | Inst. | | | | Lag | | | | Regime |
|---|---|---|---|---|---|---|---|---|---|---|---|
| | | | SHD↓ | F1↑ | NHD↓ | Ratio↓ | SHD↓ | F1↑ | NHD↓ | Ratio↓ | Acc. |
| PCMCI+ | × | W | 141.5±12. | 56.5±4.5 | 0.008±0.0 | 0.43±0.0 | **37.5**±9.2 | **91.6**±2.8 | **0.003**±0.0 | **0.08**±0.0 | × |
| Rhino | × | W | 53.5±26. | 85.5±7.4 | 0.004±0.0 | 0.14±0.0 | 52.0±5.7 | 89.2±2.2 | 0.004±0.0 | 0.11±0.0 | × |
| DYNOTEARS | × | W | 152.0±4.2 | 40.8±3.7 | 0.01±0.0 | 0.59±0.0 | 91.0±17. | 75.8±3.4 | 0.006±0.0 | 0.24±0.0 | × |
| CASTOR | × | W | 93.5±2.1 | 65.9±7.6 | 0.009±0.0 | 0.34±0.0 | 94.0±8.5 | 78.6±4.5 | 0.009±0.0 | 0.21±0.0 | × |
| RPCMCI | ✓ | W | - | - | - | - | - | - | - | - | - |
| CASTOR | ✓ | W | - | - | - | - | - | - | - | - | - |
| FANTOM | ✓ | W | **35.3**±0.7 | **90.9**±0.4 | **0.002**±0.0 | **0.09**±0.0 | 46.3±0.3 | 90.7±1.3 | 0.003±0.0 | 0.09±0.0 | **100.**±0.0 |
| | | | SHD↓ | | F1↑ | | NHD↓ | | Ratio ↓ | | Acc. |
| CD-NOD | × | S | 342.±4.3 | | 17.1±3.2 | | 0.24±0.0 | | 0.82±0.0 | | × |
| FANTOM | ✓ | S | **57.0**±2.5 | | **92.8**±0.7 | | **0.03**±0.0 | | **0.07**±0.0 | | **100.**±0.0 |

## F.3 Additional experiments for L=2

Table 20: Performance of FANTOM on $L = 2$ time series under varying node counts $d \in \{10, 20, 40\}$ and regime settings $K = 2$ and 3 .

| | | Heteroscedastic noise | | | | |
|---|---|---|---|---|---|---|
| k | d | Inst. | | Lag | | Regime |
| | | SHD | F1 | SHD | F1 | Acc. |
| 2 | 10 | 1 | 97.7 | 0 | 100.0 | 94.3 |
| | 20 | 14 | 87.0 | 4 | 98.7 | 99.0 |
| | 40 | 15 | 94.5 | 3 | 99.5 | 98.9 |
| 3 | 10 | 1 | 98.5 | 1 | 99.5 | 91.5 |
| | 20 | 5 | 97.5 | 11 | 97.4 | 97.9 |
| | 40 | 43 | 88.6 | 3 | 99.0 | 99.0 |

FANTOM maintains similar performance with a larger lag ($L = 2$). It achieves an F1 score above 87% on instantaneous links for 10, 20, and 40 nodes in MTS with 2 or 3 regimes. For time-lagged links, we evaluate using the summary graph of lagged relations; the table shows that FANTOM detects them effectively.

### F.3.1   Illustrations of learned graphs

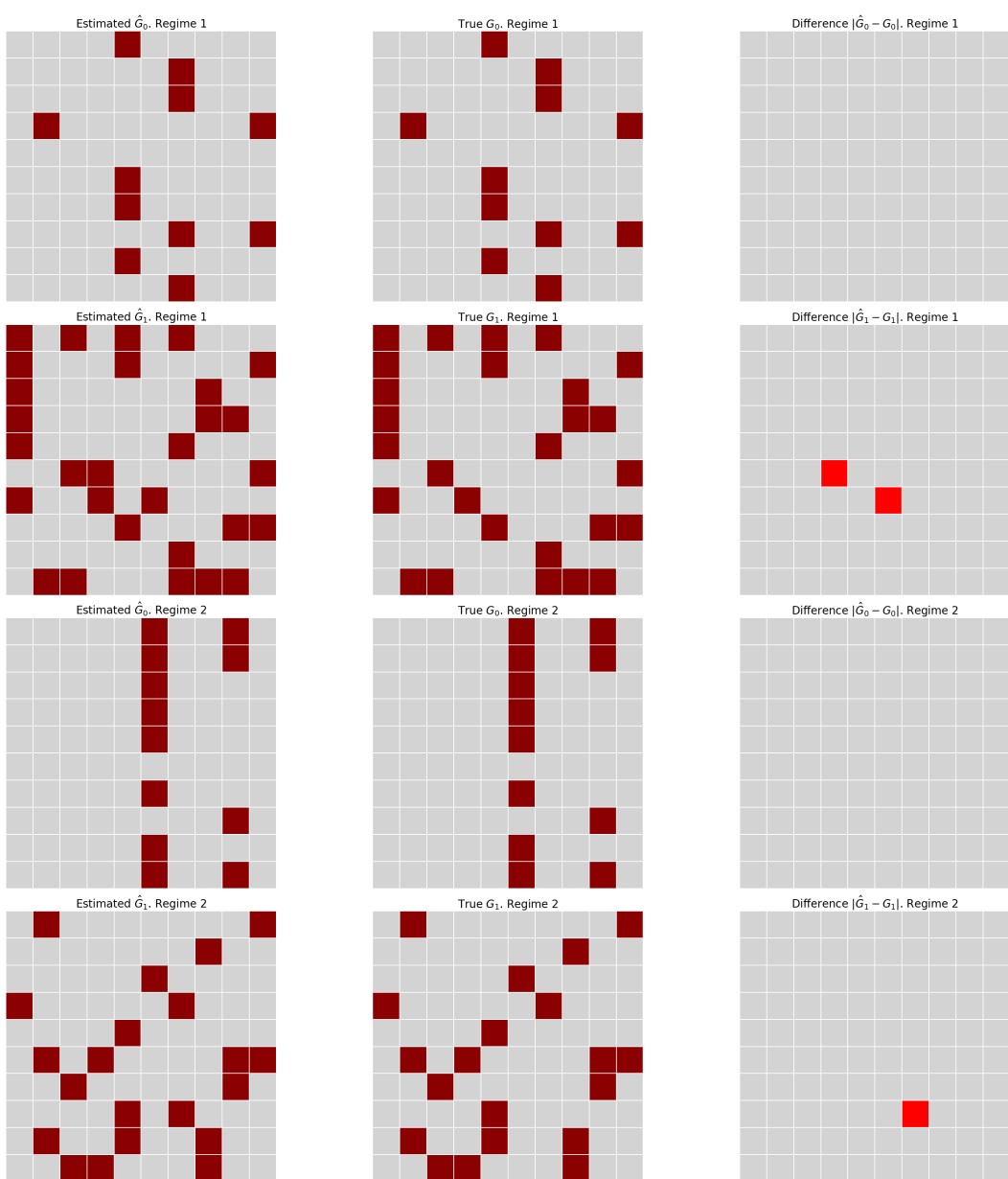

Figure 10: The estimated temporal causal graphs for two regimes, in Heteroscedastic case, consist of one matrix of 10 rows and 10 columns representing instantaneous links and another of 10 rows and 10 columns delineating time-lagged relations (with a maximum lag $L = 1$ in this case). Dark red indicates a value of one (presence of an edge), while gray symbolizes a value of 0 (absence of an edge). The second column displays the ground-truth causal graphs, and the final column highlights the difference between the estimated and true graphs.

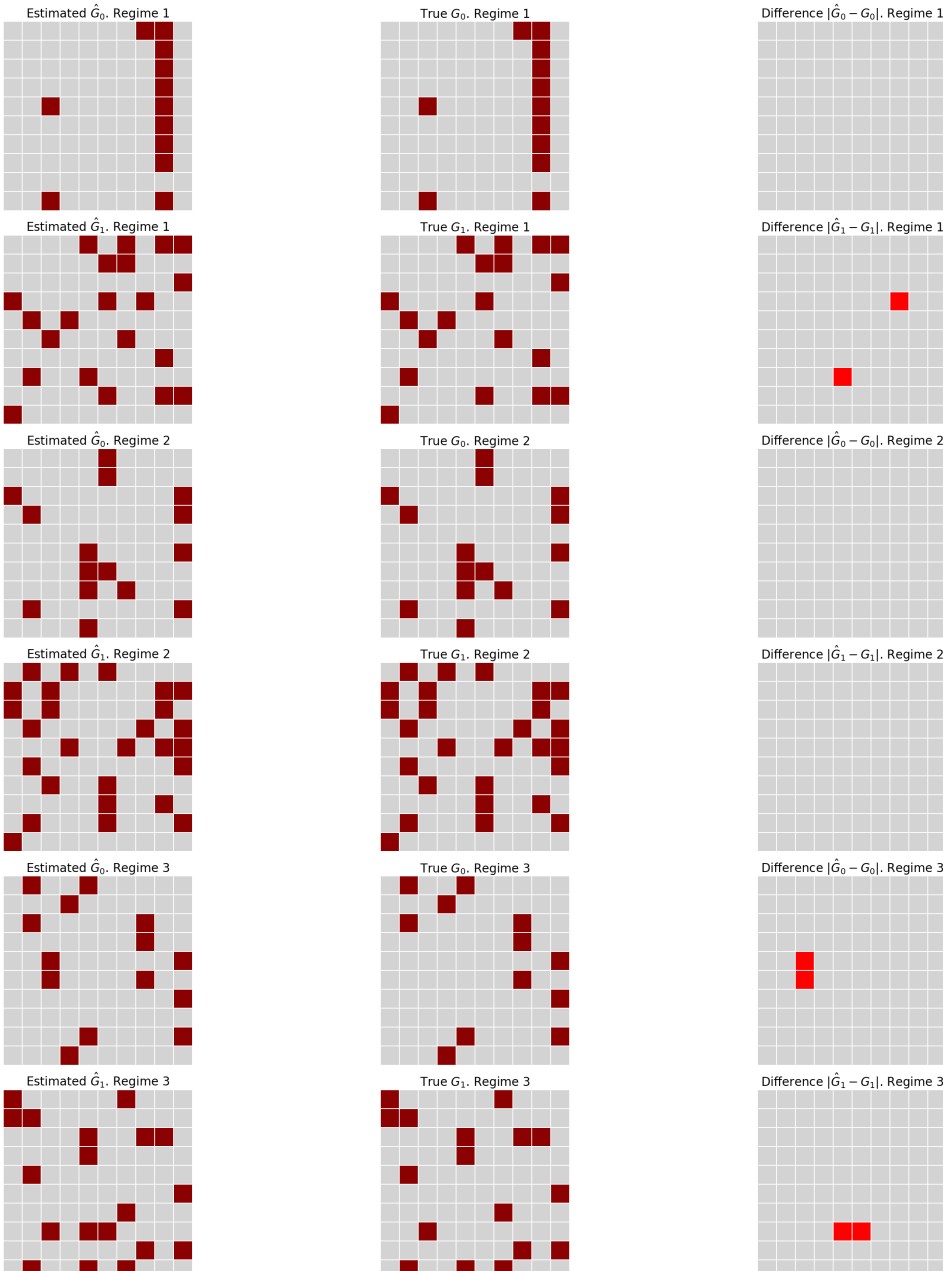

Figure 11: The estimated temporal causal graphs for three regimes, in Heteroscedastic case, consist of one matrix of 10 rows and 10 columns representing instantaneous links and another of 10 rows and 10 columns delineating time-lagged relations (with a maximum lag $L = 1$ in this case). Dark red indicates a value of one (presence of an edge), while gray symbolizes a value of 0 (absence of an edge). The second column displays the ground-truth causal graphs, and the final column highlights the difference between the estimated and true graphs.

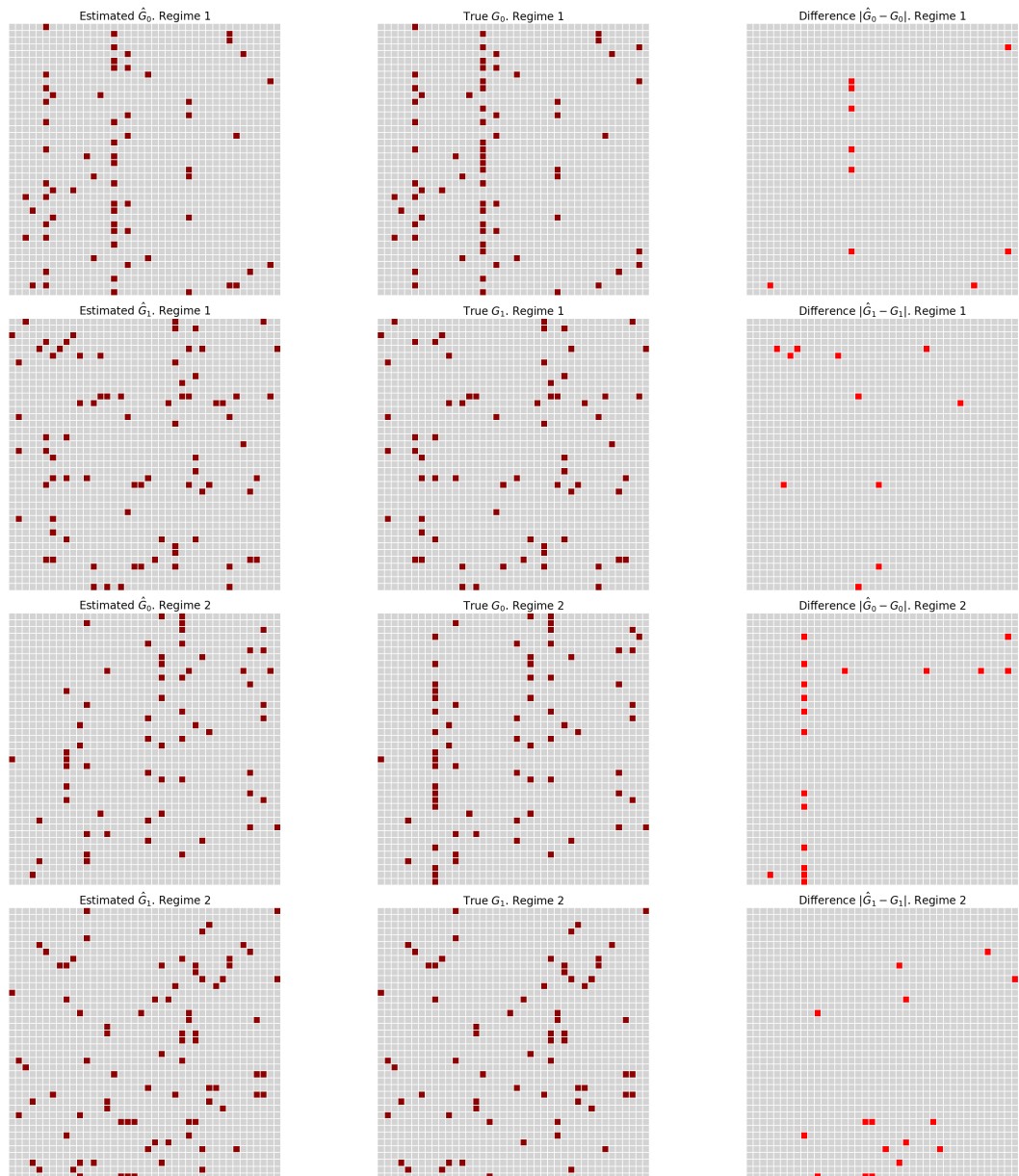

Figure 12: The estimated temporal causal graphs for three regimes, in Heteroscedastic case, consist of one matrix of 40 rows and 40 columns representing instantaneous links and another of 40 rows and 40 columns delineating time-lagged relations (with a maximum lag $L = 1$ in this case). Dark red indicates a value of one (presence of an edge), while gray symbolizes a value of 0 (absence of an edge). The second column displays the ground-truth causal graphs, and the final column highlights the difference between the estimated and true graphs.

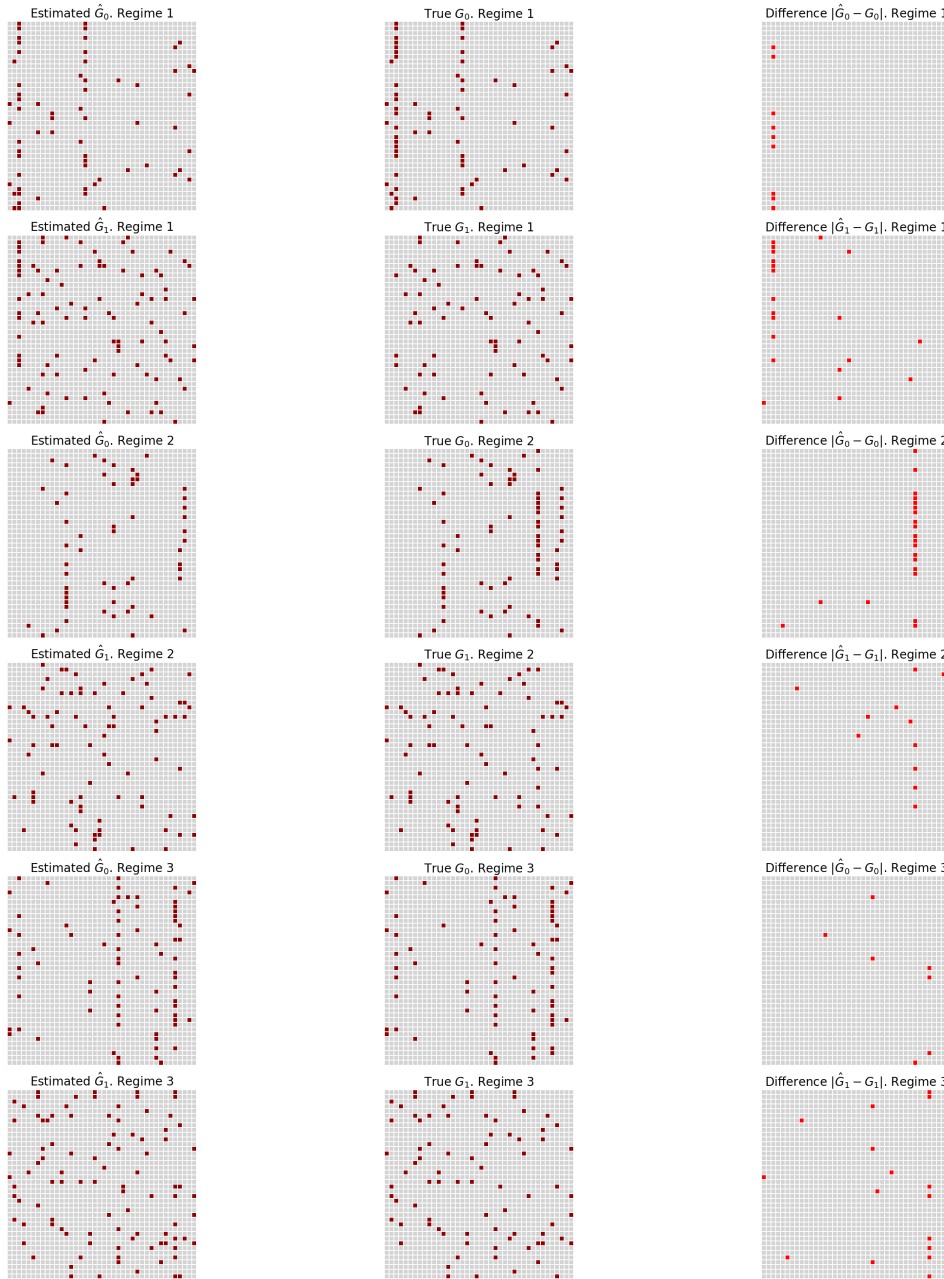

Figure 13: The estimated temporal causal graphs for three regimes, in Non-Gaussian case, consist of one matrix of 40 rows and 40 columns representing instantaneous links and another of 40 rows and 40 columns delineating time-lagged relations (with a maximum lag $L = 1$ in this case). Dark red indicates a value of one (presence of an edge), while gray symbolizes a value of 0 (absence of an edge). The second column displays the ground-truth causal graphs, and the final column highlights the difference between the estimated and true graphs.

### F.3.2 Time complexity analysis

We start first by computing the time complexity of our Temporal Graph Neural network illustrated in Figure 2. We note $d$ the input size, $e$ the embedding size used for $e_{ij}$ and $h$ hidden layer size. After the first NN block,

$$T_{\text{NN}-1} = \mathcal{O}((d+e)h + hd) + \mathcal{O}(d) = \mathcal{O}(h(d+e))$$

then after the matrix multiplication block and the second NN block, we have :

$$T_{\text{forward}} = \mathcal{O}\left(Ld^2 + 2h(d+e)\right),$$

where $L$ is the maximum lag.

Using the same architecture for a Conditional normalizing flow has a time complexity of :

$$T_{\text{CNF}} = \mathcal{O}\left(Ld^2 + h\left(e+dK\right)\right).$$

The complexity of FANTOM per iteration is $\mathcal{O}\left(N_w|\mathcal{T}|(2Ld^2 + h\left(e+dK\right)\right)$, where $K$ is the number of bins, $N_w$ is the number of regimes, and $|\mathcal{T}|$ is the number of samples.

### F.3.3 Regime detection experiments

We compare FANTOM to CASTOR [39] and KCP [1] in the task of regime detection. KCP is a multiple change-point detection method designed to handle univariate, multivariate, or complex data. Being non-parametric, KCP does not necessitate knowing the true number of change points in advance. It detects abrupt changes in the complete distribution of the data by employing a characteristic kernel.

CASTOR is a causal discovery model specifically designed for multi-regime MTS. CASTOR is learns number of regime and their indices and their corresponding causal graphs without any prior knowledge. But it is limited to normal noise with equivariance. FANTOM learns the regime indices, while handling heteroscedastic noises.

We opted to perform regime detection with 10 nodes and three different regimes. For a fair comparison, we chose three regimes without re-occurrence, as KCP only detect change points and cannot identify the re-occurrence of a specific regime.

Regarding the models employed, we use the open-source code of CASTOR implemented in Python by the authors[7]. For KCP, we employ the `Rupture` package[8].

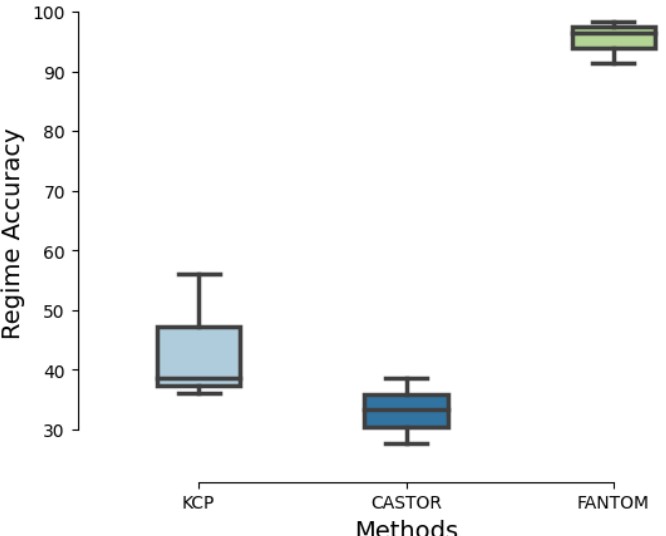

Figure 14: Comparison between FANTOM, CASTOR and KCP on regime detection for a MTS with heteroscedastic noise and composed of 3 regimes using accuracy metric. The number of nodes is $d = 10$.

From Figure 14, it is evident that FANTOM outperform CASTOR and the change-point detection method KCP. This outcome can be attributed to the limitation of KCP in detecting changing points within causal mechanisms that are represented by conditional distributions. FANTOM outperforms

---

[7]https://github.com/arahmani1/CASTOR
[8]https://centre-borelli.github.io/ruptures-docs/

CASTOR in detecting regime indices. This result can be explained by the fact that CASTOR fails in handling heteroscedastic noises and fails to learn meaningful graphs which also lead to poor performance in regime detection.

From this analysis and the other experiments shown in the different tables, we can conclude that in scenarios involving MTS with multiple regimes with non-Gaussian or Heteroscedastic noises, FANTOM offers a robust solution. Additionally, employing other methods to split the regimes and learn the causal graph through traditional causal discovery methods may not be an optimal solution:

- We demonstrate that regime indices are not well recoverable by other state-of-the-art change point detection method KCP. Therefore, employing KCP to learn the regimes and subsequently using methods like DYNOTEARS, PCMCI+, or Rhino to learn the graph may not constitute an optimal solution.

- In cases of regime recurrence, the aforementioned methods are unable to accurately detect the exact number of regimes. Therefore, if a user employs KCP and subsequently uses the regime partitions revealed by KCP as an input to a causal discovery method (such as PCMCI+, DYNOTEARS, Rhino, etc.), the running time will be significantly high.

- We show throughout all our experiments, that FANTOM outperforms all causal discovery method in DAG learning task, even when these models are in more favorable scenarios, by having access to the regime labels beforehand.

# G   Proof of proposition

In this section, we are going to provide the entire proof of our proposition 3.1. As we state before, we note $\mathcal{G} = (\mathcal{G}^r)_{r \in [\![1:N_w]\!]}$.

*Proof.* we have:

$$
\log p_\Theta \left( \boldsymbol{x}_{t \in \mathcal{T}} \right) = \sum_{t=1}^{|\mathcal{T}|} \log p_\Theta \left( \boldsymbol{x}_t | \boldsymbol{x}_{<t} \right)
$$

$$
= \sum_{t=1}^{|\mathcal{T}|} \log \sum_{\mathcal{G}} p_\Theta \left( \boldsymbol{x}_t | \boldsymbol{x}_{<t}, \mathcal{G} \right) p(\mathcal{G}) \frac{q_\phi(\mathcal{G})}{q_\phi(\mathcal{G})}
$$

$$
\geq \sum_{t=1}^{|\mathcal{T}|} \mathbb{E}_{q_\phi(\mathcal{G})} \left[ \log p_\Theta \left( \boldsymbol{x}_t | \boldsymbol{x}_{<t}, \mathcal{G} \right) + \log p(\mathcal{G}) - \log q_\phi(\mathcal{G}) \right]
$$

Let's focus on the first term of our last inequality $\log p_\Theta \left( \boldsymbol{x}_t | \boldsymbol{x}_{<t}, \mathcal{G} \right)$, we have:

$$
\log p_\Theta \left( \boldsymbol{x}_t | \boldsymbol{x}_{<t}, \mathcal{G} \right) = \log \sum_{z_t} p_{\theta^{z_t}} \left( \boldsymbol{x}_t | \boldsymbol{x}_{<t}, \mathcal{G}, z_t \right) p(z_t) \frac{p(z_t | \boldsymbol{x}_t, \boldsymbol{x}_{<t})}{p(z_t | \boldsymbol{x}_t, \boldsymbol{x}_{<t})}
$$

$$
\geq \mathbb{E}_{p(z_t | \boldsymbol{x}_t, \boldsymbol{x}_{<t})} \left[ \log p_{\theta^{z_t}} \left( \boldsymbol{x}_t | \boldsymbol{x}_{<t}, \mathcal{G}^{z_t} \right) + \log p(z_t) - \log p(z_t | \boldsymbol{x}_t, \boldsymbol{x}_{<t}) \right]
$$

$$
\geq \mathbb{E}_{p(z_t | \boldsymbol{x}_t, \boldsymbol{x}_{<t})} \left[ \log p_{\theta^{z_t}} \left( \boldsymbol{x}_t | \boldsymbol{x}_{<t}, \mathcal{G}^{z_t} \right) + \log p(z_t) \right] + H \left( p(z_t | \boldsymbol{x}_t, \boldsymbol{x}_{<t}) \right)
$$

Including this result in the previous equation gives us the following:

$$
\log p_\Theta \left( \boldsymbol{x}_{t \in \mathcal{T}} \right) \geq \sum_{t=1}^{|\mathcal{T}|} \mathbb{E}_{q_\phi(\mathcal{G})} \left[ \mathbb{E}_{p(z_t | \boldsymbol{x}_t, \boldsymbol{x}_{<t})} \left[ \log p_{\theta^{z_t}} \left( \boldsymbol{x}_t | \boldsymbol{x}_{<t}, \mathcal{G}^{z_t} \right) + \log p(z_t) \right] + H \left( p(z_t | \boldsymbol{x}_t, \boldsymbol{x}_{<t}) \right) \right]
$$

$$
+ \sum_{r=1}^{N_w} \mathbb{E}_{q_{\phi^r}(\mathcal{G}^r)} \left[ \log p(\mathcal{G}^r) \right] + H \left( q_{\phi^r}(\mathcal{G}^r) \right)
$$

$$
\equiv \mathrm{ELBO}(\Theta),
$$

we note that the priors $p(\mathcal{G}^r)$ and the variational estimations $q_{\phi^r}(\mathcal{G}^r)$ are independents.

# H  Proofs of our theoretical contributions

In this section, we concentrate on establishing the identifiability of regimes and causal graphs within the FANTOM framework. Before diving into the details, let us set and clarify the required assumptions.

## H.1  Assumptions

**Definition H.1** (Causal Stationarity, [41]). A stationary time series process $(\boldsymbol{x}_t)_{t \in \mathcal{T}}$ with graph $\mathcal{G}$ is called causally stationary over a time index set $\mathcal{T}$ if and only if for all links $x_{t-\tau}^i \to x_t^j$ in the graph

$$x_{t-\tau}^i \not\perp\!\!\!\perp x_t^j \mid \boldsymbol{x}_{<t} \setminus \{x_{t-\tau}^i\}.$$

This elucidates the inherent characteristics of the time-series data generation mechanism, thereby validating the choice of the auto-regressive model.

**Assumption H.2** (Causal Stationarity for MTS with multiple regime). A MTS $(\boldsymbol{x}_t)_{t \in \mathcal{T}}$ with $K$ regimes, graph set $(\mathcal{G}^u)_{u \in [|1:K|]}$, and regime partition $\mathcal{E} = (\mathcal{E}_u)_{u \in [|1:K|]}$ is **causally stationary** over the time index set $\mathcal{T}$ if, for each regime $u \in [|1:K|]$, the sub-series $(\boldsymbol{x}_t)_{t \in \mathcal{E}_u}$ is causally stationary with graph $\mathcal{G}^u$ as defined in definition H.1.

**Definition H.3.** (Causal Markov Property, [37]). Given a DAG $\mathcal{G}$ and a joint distribution $p$, this distribution is said to satisfy causal Markov property w.r.t. the DAG $\mathcal{G}$ if each variable is independent of its non-descendants given its parents.

This is a common assumptions for the distribution induced by an SEM. With this assumption, one can deduce conditional independence between variables from the graph.

**Assumption H.4** (Causal Markov Property (CMP)). A set of joint distributions $(p(\cdot|\mathcal{G}^r))_{r \in [|1:K|]}$ satisfies the **CMP** with respect to the DAGs $(\mathcal{G}^r)_{r \in [|1:K|]}$ if, for each $r \in [|1:K|]$, the distribution $p(\cdot|\mathcal{G}^r)$ satisfies the CMP relative to the DAG $\mathcal{G}^r$. Specifically, in every regime $r$, each variable is independent of its non-descendants given its parents.

**Assumption H.5** (Causal Minimality). Given a set of DAGs $(\mathcal{G}^r)_{r \in [|1:K|]}$ and a set of joint distribution $(p(\cdot|\mathcal{G}^r))_{r \in [|1:K|]}$, we say that this set of distributions satisfies causal minimality w.r.t. the set of DAGs $(\mathcal{G}^r)_{r \in [|1:K|]}$ if for every $r$: $p(\cdot|\mathcal{G}^r)$ is Markovian w.r.t the DAG $\mathcal{G}^r$ but not to any proper subgraph of $\mathcal{G}^r$.

**Assumption H.6** (Causal Sufficiency). A set of observed variables $\boldsymbol{V}$ is causally sufficient for a process $\boldsymbol{x}_t$ if and only if in the process every common cause of any two or more variables in $\boldsymbol{V}$ is in $\boldsymbol{V}$ or has the same value for all units in the population.

This assumption implies there are no latent confounders present in the time-series data.

|  | Causal graph | Causal Markov | Causal sufficiency | Faithfulness / Minimality | Heteroscedastic noise | Stationarity per regime |
|---|---|---|---|---|---|---|
| DYNOTEARS | W | ✓ | ✓ |  | ✗ | ✗ |
| PCMCI+ | W | ✓ | ✓ | F | ✗ | ✗ |
| RPCMCI | W | ✓ | ✓ | F | ✗ | ✓ |
| Rhino | W | ✓ | ✓ | M | ✗ | ✗ |
| CD-NOD | S | ✓ | ✓ | F | ✗ | ✓ |
| CASTOR | W | ✓ | ✓ | M | ✗ | ✓ |
| FANTOM | W | ✓ | ✓ | M | ✓ | ✓ |

Table 21: Summary of the main assumptions of algorithms considered in the paper. For causal graphs, S means that the algorithm provides a summary causal graph and W means that the algorithm provides a window causal graph; F corresponds to faithfulness and M to minimality. An empty cell mean that the information given in the corresponding column was not discussed by the authors of the corresponding algorithm.

The table 21 illustrates that most assumptions (causal sufficiency, causal Markov, faithfulness/minimality) are commonly shared among various state-of-the-art models in causal discovery.

However, FANTOM, CASTOR, RPCMCI, and CD-NOD relax the assumption of stationarity and instead assume that the MTS (Multivariate Time Series) are composed of different regimes. While CD-NOD predicts only a summary causal graph, FANTOM, CASTOR and RPCMCI predict a window causal graph, which can subsequently be used to reconstruct a summary graph. FANTOM is the only model that can handle heteroscedastic noise.

## H.2 Proof of theorem 4.2

We start first by proving theorem 4.2. To do so, we will prove identifiability in the case of bivariate time series, Lemma H.7. Then we will prove identifiability in the case of MTS.

**Lemma H.7.** *Assume Causal Markov property, minimality, stationarity, sufficiency and $(x_t^1, x_t^2)_{t \in \mathcal{T}}$ be a bivariate time series such that $(x_t^1, x_t^2)_{t \in \mathcal{T}}$ following Eq(1) where $K = 1$ and $\epsilon_t^i \sim \mathcal{N}(0, 1)$. We have $x_t^1, x_t^2$ follow Gaussian distribution, if $f^1, f^2$ are non linear and $\frac{1}{g^1}, \frac{1}{g^2}$ are not a polynomial of degree two then the bivariate Temporal heteroscedastic Gaussian noise (THGNM) model is identifiable.*

*Proof of the Lemma.* Let's assume we have two temporal causal graph $\mathcal{G}$ and $\mathcal{G}'$ for the bivariate TGHNM.

**Disagreement in time lagged relationships.** Assume that $\mathcal{G}$ and $\mathcal{G}'$ do not differ in the instantaneous effects. $\forall i \in \{1, 2\}$: $\mathbf{Pa}_{\mathcal{G}}^i(t) = \mathbf{Pa}_{\mathcal{G}'}^i(t)$. Hence and Wlog, there is some $k > 0$ and an edge $x_{t-k}^1 \to x_t^2$ in $\mathcal{G}$ but not in $\mathcal{G}'$. From $\mathcal{G}'$ and the Causal Markov property, we have that $x_{t-k}^1 \perp\!\!\!\perp x_t^2 \mid \mathcal{S}$, where $\mathcal{S} = (\{x_{t-l}^i, 1 \leq l \leq L, i \in \{1, 2\}\} \cup \mathbf{ND}_t) \setminus \{x_{t-k}^1, x_t^2\}$, and $\mathbf{ND}_t$ are all $X_t^i$ that are non-descendants (wrt instantaneous effects) of $x_t^2$. Applied to $\mathcal{G}$, causal minimality leads to a contradiction because $x_{t-k}^1 \not\perp\!\!\!\perp x_t^2 \mid \mathcal{S}$ in $\mathcal{G}$, and the above reasoning shows that it exists a subgraph $\mathcal{G}'$ of $\mathcal{G}$ that is Markovian to the joint distribution of the data.

**Disagreement on instantaneous parents.** Now, let's assume we have a forward model, $\forall t \in \mathcal{T}$:

$$x_t^1 = f^1 \left( \mathbf{Pa}_{\mathcal{G}}^1(< t), x_t^2 \right) + g^1 \left( \mathbf{Pa}_{\mathcal{G}}^1(< t), x_t^2 \right) \cdot \epsilon_t^1,$$

We will prove by contradiction that a backward model

$$x_t^2 = f^2 \left( \mathbf{Pa}_{\mathcal{G}}^2(< t), x_t^1 \right) + g^2 \left( \mathbf{Pa}_{\mathcal{G}}^2(< t), x_t^1 \right) \cdot \epsilon_t^2,$$

can not exists.
We note $\mathbf{h}_t = \mathbf{Pa}_{\mathcal{G}}^1(< t) \cup \mathbf{Pa}_{\mathcal{G}}^2(< t)$. We know that $\epsilon_t^2 \perp\!\!\!\perp (x_t^1, \mathbf{Pa}_{\mathcal{G}}^1(< t), \mathbf{Pa}_{\mathcal{G}}^2(< t))$ and $\epsilon_t^1 \perp\!\!\!\perp (x_t^2, \mathbf{Pa}_{\mathcal{G}}^1(< t), \mathbf{Pa}_{\mathcal{G}}^2(< t))$, using Lemma 36 in Peters et al. [38], we have:

$$x_t^1|_{\mathbf{h}_t} = f^1 \left( pa_{<t}^1, x_t^2|_{\mathbf{h}_t} \right) + g^1 \left( pa_{<t}^1, x_t^2|_{\mathbf{h}_t} \right) \cdot \epsilon_t^1,$$

$$x_t^2|_{\mathbf{h}_t} = f^2 \left( pa_{<t}^2, x_t^1|_{\mathbf{h}_t} \right) + g^2 \left( pa_{<t}^2, x_t^1|_{\mathbf{h}_t} \right) \cdot \epsilon_t^2,$$

where $x_t^i|_{\mathbf{h}_t}$ is $x_t^i$ conditioned on $\mathbf{h}_t$. This last result contradicts the theorem states by Khemakhem et al. [29]. Hence, our bivariate TGHNM is identifiable.

Let's prove this results in the case of MTS (Theorem 4.2). In the case of Disagreement in time lagged relationships, we can use the same proof for the bivariate case.

**Disagreement on instantaneous parents.** Let's assume we have two temporal causal graph $\mathcal{G}$ and $\mathcal{G}'$ such that $\mathcal{G} \neq \mathcal{G}'$. According to the Proposition 28 in Peters et al [38], for $\mathcal{G}$ and $\mathcal{G}'$ be two different DAGs over a set of variables $\mathbf{V}$, such that $\mathbf{x}_t$ is generated by our HNM and satisfies the Markov condition and causal minimality with respect to $\mathcal{G}$ and $\mathcal{G}'$. Then there are variables $x_t^1, x_t^2 \in \mathbf{V}$ such that for the set $\mathbf{Q} := \mathbf{Pa}_{\mathcal{G}}^1(t) \setminus \{x_t^2\}$, $\mathbf{Y} := \mathbf{Pa}_{\mathcal{G}'}^2(t) \setminus \{x_t^1\}$ and $\mathbf{S} := \mathbf{Q} \cup \mathbf{Y}$, we have: 1) $x_t^2 \to x_t^1$ in $\mathcal{G}$ and $x_t^1 \to x_t^2$ in $\mathcal{G}'$. 2) $\mathbf{S} \subseteq \mathbf{ND}_{x_t^1}^{\mathcal{G}} \setminus \{x_t^2\}$ and $\mathbf{S} \subseteq \mathbf{ND}_{x_t^2}^{\mathcal{G}'} \setminus \{x_t^1\}$. $\mathbf{Pa}_{\mathcal{G}}^1(t)$ is the set of parent variables of $x_t^1$ in graph $\mathcal{G}$. $\mathbf{ND}_{x_t^1}^{\mathcal{G}}$ is the set of non-descendant (wrt instantaneous effects) of $x_t^1$ in graph $\mathcal{G}$.

We consider $\mathbf{S} = \mathbf{s}$ with $p(\mathbf{s}) > 0$. Denote $x_t^{1,*} := x_t^1 \mid \mathbf{S} = \mathbf{s}$ and $x_t^{2,*} := x_t^2 \mid \mathbf{S} = \mathbf{s}$. Lemma 37 in Peters et al. [38] states that if $p(\mathbf{x}_t)$ is generated according to the SEM models as follows:

$$x_t^i = f_i \left( \mathbf{Pa}_{\mathcal{G}}^i(< t), \mathbf{Pa}_{\mathcal{G}}^i(t), \epsilon_t^i \right), i \in \{1, 2, \cdots, d\}, x_t^i \in \boldsymbol{V}$$

with corresponding DAG $\mathcal{G}$, then for a variable $x_t^i \in \boldsymbol{V}$, if $\boldsymbol{S} \subseteq \mathbf{ND}_{x_t^i}^{\mathcal{G}}$ then $\epsilon_t^i \perp\!\!\!\perp \boldsymbol{S}$. Our TGHNM can be viewed one specific class of the SEM in the aforementioned equation. Hence, Lemma 37 holds under our TGHNM and renders $\epsilon_t^1 \perp\!\!\!\perp (x_t^2, \boldsymbol{S})$ and $\epsilon_t^2 \perp\!\!\!\perp (x_t^1, \boldsymbol{S})$, using Lemma 36 in Peters et al. [38], we have:

$$x_t^{1,*}|_{\boldsymbol{h}_t} = f^1 \left( \boldsymbol{q}, pa_{<t}^1, x_t^{2,*}|_{\boldsymbol{h}_t} \right) + g^1 \left( \boldsymbol{q}, pa_{<t}^1, x_t^{2,*}|_{\boldsymbol{h}_t} \right) \cdot \epsilon_t^1,$$

$$x_t^{2,*}|_{\boldsymbol{h}_t} = f^2 \left( \boldsymbol{y}, pa_{<t}^2, x_t^{1,*}|_{\boldsymbol{h}_t} \right) + g^2 \left( \boldsymbol{y}, pa_{<t}^2, x_t^{1,*}|_{\boldsymbol{h}_t} \right) \cdot \epsilon_t^2,$$

where $x_t^i|_{\boldsymbol{h}_t}$ is $x_t^i$ conditioned on $\boldsymbol{h}_t$. This results contradict our previous proved Lemma, then THGNM is identifiable model under the conditions stated in the theorem.

**Theorem H.8** (Identifiability of Temporal Non Gaussian noise model (TNGNM)). *Assume Causal Markov property, stationarity, minimality, sufficiency and let $(\boldsymbol{x}_t)_{t \in \mathcal{T}}$ be a MTS following a TNGNM, $\forall t \in \mathcal{T}$:*

$$x_t^i = f^i \left( \mathbf{Pa}_{\mathcal{G}}^i(< t), \mathbf{Pa}_{\mathcal{G}}^i(t) \right) + \epsilon_t^i, \tag{14}$$

*where $f^i$ is a differentiable function, and $\epsilon_t^i$ are mutually independent noises and follow a non Gaussian distribution. The TNGNM is identifiable.*

*Proof.* The proof of this theorem could be concluded from theorem 1 in Rhino [15]. Eq(14) is a special case of Rhino SEMs.

## H.3 Identifiability results in the case of Temporal General Heteroscedastic Noise Models

In this section, we will present our identifiability results for the case of Temporal General Heteroscedastic Noise, where a MTS has the following SEM :$\forall t \in \mathcal{T}$:

$$x_t^i = f^i \left( \mathbf{Pa}_{\mathcal{G}}^i(< t), \mathbf{Pa}_{\mathcal{G}}^i(t) \right) + g^i \left( \mathbf{Pa}_{\mathcal{G}}^i(< t), \mathbf{Pa}_{\mathcal{G}}^i(t) \right) \cdot \epsilon_t^i, \tag{15}$$

where $f^i$ and $g^i$ are differentiable functions, with $g_i$ strictly positive and $\epsilon_t^i$ are mutually independent normal noises and can have any arbitrary density distribution. We assume $\mathbb{E}(\epsilon_t^i) = 0$ and $\mathbb{E}((\epsilon_t^i)^2) = 1$ without loss of generality.

We will start first by showing that if backward model, respects to instantaneous links, exists in the bivariate case then, the data generating mechanism must fulfill the a Partial Differential Equation (PDE). Then, following Peters et al. [38] and Strobl et al. [50] for defining Restricted SEM on iid, we will define a Temporal Restricted Heteroscedastic Noise model and show its identifiability.

**Lemma H.9.** *Assume Causal Markov property, minimality, stationarity, sufficiency and $(x_t^1, x_t^2)_{t \in \mathcal{T}}$ be a bivariate time series. Then we have time lagged parents are identifiable, and a backward model with respect to instantaneous links can be fit i.e. $\forall t \in \mathcal{T}$:*

$$\begin{cases} \tilde{x}_t^1 = x_t^1|_{\boldsymbol{h}_t} = f^1 \left( \tilde{x}_t^2 \right) + g^1 \left( \tilde{x}_t^2 \right) \cdot \epsilon_t^1 \\ \tilde{x}_t^2 = x_t^2|_{\boldsymbol{h}_t} = f^2 \left( \tilde{x}_t^1 \right) + g^2 \left( \tilde{x}_t^1 \right) \cdot \epsilon_t^2. \end{cases} \tag{16}$$

*We note $\boldsymbol{h}_t = \mathbf{Pa}^1(< t) \cup \mathbf{Pa}^2(< t)$, and let $\nu_1(\cdot)$ and $\nu_2(\cdot)$ be the twice differentiable log densities of $\tilde{X}_t^1$ and $\epsilon_t^2$ respectively. For compact notation, define*

$$\nu_{\tilde{X}_t^2|\tilde{X}_t^1}(\tilde{x}_t^2 \mid \tilde{x}_t^1) = \log \left( p_{\tilde{X}_t^2|\tilde{X}_t^1}(\tilde{x}_t^2 \mid \tilde{x}_t^1) \right)$$

$$= \log \left( p_{\epsilon_t^2} \left( \frac{\tilde{x}_t^2 - f^2(\tilde{x}_t^1)}{g^2(\tilde{x}_t^1)} \right) / g^2(\tilde{x}_t^1) \right)$$

$$= \nu_2 \left( \frac{\tilde{x}_t^2 - f^2(\tilde{x}_t^1)}{g^2(\tilde{x}_t^1)} \right) - \log(g^2(\tilde{x}_t^1)) \quad and$$

$$G(\tilde{x}_t^2, \tilde{x}_t^1) = g^1(\tilde{x}_t^2)(f^1)'(\tilde{x}_t^2) + (g^1)'(\tilde{x}_t^2)[\tilde{x}_t^1 - f^1(\tilde{x}_t^2)].$$

*Assume that $f^1$, $g^1$, $f^2$, and $g^2$ are twice differentiable. Then, the data generating mechanism must fulfill the following PDE for all $(\tilde{x}_t^2, \tilde{x}_t^1)$ with $G(\tilde{x}_t^2, \tilde{x}_t^1) \neq 0$.*

$$
\begin{aligned}
0 = {} & \nu_1''(\tilde{x}_t^1) + \frac{(g^1)'(\tilde{x}_t^2)}{G(\tilde{x}_t^2, \tilde{x}_t^1)} \nu_1'(\tilde{x}_t^1) + \frac{\partial^2}{\partial(\tilde{x}_t^1)^2} \nu_{\tilde{X}_t^2|\tilde{X}_t^1}(\tilde{x}_t^2 \mid \tilde{x}_t^1) + \\
& \frac{g^1(\tilde{x}_t^2)}{G(\tilde{x}_t^2, \tilde{x}_t^1)} \frac{\partial^2}{\partial \tilde{x}_t^1 \partial \tilde{x}_t^2} \nu_{\tilde{X}_t^2|\tilde{X}_t^1}(\tilde{x}_t^2 \mid \tilde{x}_t^1) + \frac{(g^1)'(\tilde{x}_t^2)}{G(\tilde{x}_t^2, \tilde{x}_t^1)} \frac{\partial}{\partial \tilde{x}_t^1} \nu_{\tilde{X}_t^2|\tilde{X}_t^1}(\tilde{x}_t^2 \mid \tilde{x}_t^1).
\end{aligned}
\tag{17}
$$

We drop the time-lagged parent in Eq (16) to simplify the notation, since conditioning on the history makes it redundant.

*Proof of the Lemma.* Let's assume we have two temporal causal graph $\mathcal{G}$ and $\mathcal{G}'$ for the bivariate temporal heteroscedastic causal models where the noise distribution could follow any arbitrary distribution.

**Disagreement in time lagged relationships.** Assume that $\mathcal{G}$ and $\mathcal{G}'$ do not differ in the instantaneous effects. $\forall i \in \{1,2\}$: $\mathbf{Pa}_{\mathcal{G}}^i(t) = \mathbf{Pa}_{\mathcal{G}'}^i(t)$. Hence and Wlog, there is some $k > 0$ and an edge $x_{t-k}^1 \to x_t^2$ in $\mathcal{G}$ but not in $\mathcal{G}'$. From $\mathcal{G}'$ and the Causal Markov property, we have that $x_{t-k}^1 \perp\!\!\!\perp x_t^2 \mid \mathcal{S}$, where $\mathcal{S} = (\{x_{t-l}^i, 1 \leq l \leq L, i \in \{1,2\}\} \cup \mathbf{ND}_t) \setminus \{x_{t-k}^1, x_t^2\}$, and $\mathbf{ND}_t$ are all $X_t^i$ that are non-descendants (wrt instantaneous effects) of $x_t^2$. Applied to $\mathcal{G}$, causal minimality leads to a contradiction because $x_{t-k}^1 \not\perp\!\!\!\perp x_t^2 \mid \mathcal{S}$ in $\mathcal{G}$, and the above reasoning shows that it exists a subgraph $\mathcal{G}'$ of $\mathcal{G}$ that is Markovian to the joint distribution of the data.

**Disagreement in instantaneous parents.** Now, let's assume we have a forward model, after conditioning on $\boldsymbol{h}_t = \mathbf{Pa}_{\mathcal{G}}^1(< t) \cup \mathbf{Pa}_{\mathcal{G}}^2(< t)$, $\forall t \in \mathcal{T}$:

$$
\tilde{x}_t^1 = x_t^1|_{\boldsymbol{h}_t} = f^1(\tilde{x}_t^2) + g^1(\tilde{x}_t^2) \cdot \epsilon_t^1
$$

We want to prove that if a backward model

$$
\tilde{x}_t^2 = x_t^2|_{\boldsymbol{h}_t} = f^2(\tilde{x}_t^1) + g^2(\tilde{x}_t^1) \cdot \epsilon_t^2
$$

exists then the PDE in Eq(17) is fulfilled.

Our conditioning trick on time lagged parents makes the use of Immer et al. [25] theorem 1 feasible in our case. We employ the change of variables from $\{\tilde{x}_t^2, \epsilon_t^1\}$ to $\{\tilde{x}_t^1, \epsilon_t^2\}$ and the proof will be the same as Immer et al.. Hence, we leverage Theorem 1 of Immer el al. and we conclude that if a backward model exists the PDE Eq(17) is verified.

**Theorem H.10** (Identifiability of Temporal Restricted Heteroscedastic noise model (TRHNM)). *Assume Causal Markov property, minimality, sufficiency and let $(\boldsymbol{x}_t)_{t \in \mathcal{T}}$ be a MTS following* Eq(1) *where $K = 1$. The graph $\mathcal{G}$ is uniquely identified if $\forall i \in [[1 : d]], \forall j : x_t^j \in \mathbf{Pa}_{\mathcal{G}}^i(t)$ and $\boldsymbol{S}$ such that $\left(\mathbf{Pa}_{\mathcal{G}}^i(t) \backslash x_t^j\right) \subseteq \boldsymbol{S} \subseteq \left(\mathrm{Nd}\left(x_t^i\right) \backslash x_t^j\right)$, there exists $\boldsymbol{S} = \boldsymbol{s}$ where $p(\boldsymbol{s}) > 0$ $\boldsymbol{h}_t = \mathbf{Pa}_{\mathcal{G}}^i(< t) \cup \mathbf{Pa}_{\mathcal{G}}^j(< t)$ and $p\left(x_t^i, x_t^j \mid \boldsymbol{s}, \boldsymbol{h}_t\right)$ do not satisfy PDE of Equation 17, and we call the model that verify this condition, the Temporal Restricted Heteroscedastic noise model.*

*Proof of the theorem.* We will follow the same steps in the proof of theorem 4.2. Let's assume we have two temporal causal graph $\mathcal{G}$ and $\mathcal{G}'$ for the multivariate TRHNM. We assume also that $\forall i \in [[1 : d]], \forall j : x_t^j \in \mathbf{Pa}_{\mathcal{G}}^i(t)$ and $\boldsymbol{S}$ such that $\left(\mathbf{Pa}_{\mathcal{G}}^i(t) \backslash x_t^j\right) \subseteq \boldsymbol{S} \subseteq \left(\mathrm{Nd}\left(x_t^i\right) \backslash x_t^j\right)$, there exists $\boldsymbol{S} = \boldsymbol{s}$ where $p(\boldsymbol{s}) > 0$ $\boldsymbol{h}_t = \mathbf{Pa}_{\mathcal{G}}^i(< t) \cup \mathbf{Pa}_{\mathcal{G}}^j(< t)$ and $p\left(x_t^i, x_t^j \mid \boldsymbol{s}, \boldsymbol{h}_t\right)$ do not satisfy PDE of Equation 17.

We will start by showing that time lagged parents are identifiable. Same reasoning in the bivariate case.

**Disagreement in time lagged relationships.** Assume that $\mathcal{G}$ and $\mathcal{G}'$ do not differ in the instantaneous effects. $\forall i \in \{1,2\}$: $\mathbf{Pa}_{\mathcal{G}}^i(t) = \mathbf{Pa}_{\mathcal{G}'}^i(t)$. Hence and Wlog, there is some $k > 0$ and an edge $x_{t-k}^1 \to x_t^2$ in $\mathcal{G}$ but not in $\mathcal{G}'$. From $\mathcal{G}'$ and the Causal Markov property, we have that $x_{t-k}^1 \perp\!\!\!\perp x_t^2 \mid \mathcal{S}$, where $\mathcal{S} = (\{x_{t-l}^i, 1 \leq l \leq L, i \in \{1,2\}\} \cup \mathbf{ND}_t) \setminus \{x_{t-k}^1, x_t^2\}$, and $\mathbf{ND}_t$ are all $X_t^i$ that are non-descendants (wrt instantaneous effects) of $x_t^2$. Applied to $\mathcal{G}$, causal minimality leads to a contradiction

because $x^1_{t-k} \not\perp\!\!\!\perp x^2_t \mid S$ in $\mathcal{G}$, and the above reasoning shows that it exists a subgraph $\mathcal{G}'$ of $\mathcal{G}$ that is Markovian to the joint distribution of the data.

**Disagreement on instantaneous parents.** Let's now assume we have two temporal causal graph $\mathcal{G}$ and $\mathcal{G}'$ such that $\mathcal{G} \neq \mathcal{G}'$. According to the Propostion 29 in Peters et al [38], for $\mathcal{G}$ and $\mathcal{G}'$ be two different DAGs over a set of variables $V$, such that $x_t$ is generated by our TRHNM and satisfies the Markov condition and causal minimality with respect to $\mathcal{G}$ and $\mathcal{G}'$. Then there are variables $x^1_t, x^2_t \in V$ such that for the set $Q := \mathbf{Pa}^1_{\mathcal{G}}(t)\backslash\{x^2_t\}, Y := \mathbf{Pa}^2_{\mathcal{G}'}(t)\backslash\{x^1_t\}$ and $S := Q \cup Y$, we have: 1) $x^2_t \to x^1_t$ in $\mathcal{G}$ and $x^1_t \to x^2_t$ in $\mathcal{G}'$. 2) $S \subseteq \mathbf{ND}^{\mathcal{G}}_{x^1_t}\backslash\{x^2_t\}$ and $S \subseteq \mathbf{ND}^{\mathcal{G}'}_{x^2_t}\backslash\{x^1_t\}$ . $\mathbf{Pa}^1_{\mathcal{G}}(t)$ is the set of parent variables of $x^1_t$ in graph $\mathcal{G}$. $\mathbf{ND}^{\mathcal{G}}_{x^1_t}$ is the set of non-descendant (wrt instantaneous effects) of $x^1_t$ in graph $\mathcal{G}$.

We consider $S = s$ with $p(s) > 0$. Lemma 37 in Peters et al. [38] states that if $p(x_t)$ is generated according to the SEM models as follows:

$$x^i_t = f_i\left(\mathbf{Pa}^i_{\mathcal{G}}(< t), \mathbf{Pa}^i_{\mathcal{G}}(t), \epsilon^i_t\right), i \in \{1, 2, \cdots, d\}, x^i_t \in V$$

with corresponding DAG $\mathcal{G}$, then for a variable $x^i_t \in V$, if $S \subseteq \mathbf{ND}^{\mathcal{G}}_{x^i_t}$ then $\epsilon^i_t \perp\!\!\!\perp S$. Our TRHNM can be viewed one specific class of the SEM in the aforementioned equation. Hence, Lemma 37 holds under our TRHNM and applying it to $x^1_t$ renders $\epsilon^1_t \perp\!\!\!\perp (x^2_t, S)$ and $x^2_t \, \epsilon^2_t \perp\!\!\!\perp (x^1_t, S)$.

Using now Lemma 36 in Peters et al. [38], and we denote $x^{1,*}_t := x^1_t \mid S = s$ and $x^{2,*}_t := x^2_t \mid S = s$. We have:

$$
\begin{aligned}
x^{1,*}_t|_{h_t} &= f^1\left(x^{2,*}_t|_{h_t}\right) + g^1\left(x^{2,*}_t|_{h_t}\right) \cdot \epsilon^1_t \\
x^{2,*}_t|_{h_t} &= f^2\left(x^{1,*}_t|_{h_t}\right) + g^2\left(x^{1,*}_t|_{h_t}\right) \cdot \epsilon^2_t,
\end{aligned}
\tag{18}
$$

where $x^{i,*}_t|_{h_t}$ is $x^i_t$ conditioned on $h_t, S$ and $h_t = \mathbf{Pa}^1_{\mathcal{G}}(< t) \cup \mathbf{Pa}^2_{\mathcal{G}}(< t)$ in this case, which is also equal to $h_t = \mathbf{Pa}^1_{\mathcal{G}'}(< t) \cup \mathbf{Pa}^2_{\mathcal{G}'}(< t)$, because we proved identifiability of time lagged parents. Eq(18) raise a contradiction because, having these forward and backward models imply the verification of PDE 17. But we chose $s$ such that this PDE is not verified hence contradiction. Then, the identifiability of our Temporal Restricted Heteroscedastic noise.

### H.4    Proof of theorem 4.3

In this section, we want to prove the identifiability of mixture of Temporal causal models either in the case of Temporal Heteroscedastic Gaussian noise, Temporal Restricted Heteroscedastic noise and Homoscedastic NonGaussian noise.

*Proof.* Let $\mathcal{F}$ be a family of identifiable temporal causal models either from TGHNM or TNGNM, $\mathcal{F} = (p_{\theta^r}(\cdot|\cdot, \mathcal{G}^r))_{r \in \mathbb{N}^*}$ that are linearly independent and let $\mathcal{M}_K$ be the family of all $K$-finite mixtures of elements from $\mathcal{F}$, i.e.

$$\mathcal{M}_K = \left\{ p(x_t|x_{<t}) = \sum_{r=1}^K \pi_t(\omega^r) p_{\theta^r}(x_t|x_{<t}, \mathcal{G}^r), p_{\theta^r}(\cdot|\cdot, \mathcal{G}^r) \in \mathcal{F}, \forall t \in \mathcal{T} : \pi_t(\omega^r) > 0 \text{ and } \sum_{r=1}^K \pi_t(\omega^r) = 1 \right\}$$

First, we introduce a result from Yakowitz & Spragins [58] that established a necessary and sufficient condition for the identifiability of finite mixtures of multivariate distributions.

**Theorem H.11** (Identifiability of finite mixtures of distributions, Yakowitz & Spragins [58])**.** . *Let $\mathcal{F} = \{F(x; \alpha), \alpha \in \mathbb{R}^m, x \in \mathbb{R}^n\}$ be a finite mixture of distributions. Then $\mathcal{F}$ is identifiable if and only if $\mathcal{F}$ is a linearly independent set over the field of real numbers.*

We will further assume that it exists two distribution such that $\forall(x_t, x_{<t})$ covering the space value of random variables $(X_t, X_{<t})$:

$$
\begin{aligned}
p(x_t|x_{<t}) &= \sum_{r=1}^K \pi_t(\omega^r) p_{\theta^r}(x_t|x_{<t}, \mathcal{G}^r) \\
p(x_t|x_{<t}) &= \sum_{r=1}^{\tilde{K}} \pi_t(\tilde{\omega}^r) \tilde{p}_{\tilde{\theta}^r}(x_t|x_{<t}, \tilde{\mathcal{G}}^r)
\end{aligned}
\tag{19}
$$

Our objective is to show first that $K = \tilde{K}$, it exists a permutation $\sigma$ and a translation function $\varrho : \mathbb{R}^2 \to \mathbb{R}^2 \colon (\theta^r, \mathcal{G}^r) = (\tilde{\theta}^{\sigma(r)}, \tilde{\mathcal{G}}^{\sigma(r)})$ and $\boldsymbol{\omega}^r = \varrho(\tilde{\boldsymbol{\omega}}^{\sigma(r)})$.

Using that $\mathcal{F} = (p_{\theta^r}(\cdot|\cdot, \mathcal{G}^r))_{r=1}^K$ are linearly independent and fixing $t$:

$$\sum_{r=1}^K \underbrace{\pi_t(\boldsymbol{\omega}^r)}_{a_r} p_{\theta^r}(\boldsymbol{x}_t | \boldsymbol{x}_{<t}, \mathcal{G}^r) = \sum_{r=1}^{\tilde{K}} \underbrace{\pi_t(\tilde{\boldsymbol{\omega}}^r)}_{b_r} \tilde{p}_{\tilde{\theta}^r}(\boldsymbol{x}_t | \boldsymbol{x}_{<t}, \tilde{\mathcal{G}}^r)$$

$$\sum_{r=1}^K a_r p_{\theta^r}(\boldsymbol{x}_t | \boldsymbol{x}_{<t}, \mathcal{G}^r) = \sum_{r=1}^{\tilde{K}} b_r \tilde{p}_{\tilde{\theta}^r}(\boldsymbol{x}_t | \boldsymbol{x}_{<t}, \tilde{\mathcal{G}}^r),$$

and this true $\forall (\boldsymbol{x}_t, \boldsymbol{x}_{<t})$ covering the space value of the random variable $\boldsymbol{Y} = (\boldsymbol{X}_t, \boldsymbol{X}_{<t})$. By using theorem H.11, we can conclude that: $K = \tilde{K}$ and it exists a permutation $\sigma$ such that: $(\theta^r, \mathcal{G}^r) = (\tilde{\theta}^{\sigma(r)}, \tilde{\mathcal{G}}^{\sigma(r)})$ and $\forall t \in \mathcal{T} : \pi_t(\boldsymbol{\omega}^r) = \pi_t(\tilde{\boldsymbol{\omega}}^r)$.

To proof our identifiability as defined in definition 4.1, we still need to prove that $\boldsymbol{\omega}^r = \varrho(\tilde{\boldsymbol{\omega}}^{\sigma(r)})$. We have $\forall t \in \mathcal{T} : \pi_t(\boldsymbol{\omega}^r) = \pi_t(\tilde{\boldsymbol{\omega}}^r)$, we take two indices $r, s \in [|1 : K|]$ :

$$\begin{cases} \pi_t(\boldsymbol{\omega}^r) = \pi_t(\tilde{\boldsymbol{\omega}}^{\sigma(r)}) \\ \pi_t(\boldsymbol{\omega}^s) = \pi_t(\tilde{\boldsymbol{\omega}}^{\sigma(s)}) \end{cases} \tag{20}$$

To handle time varying weights identifiability, we will consider ratios of mixture weights:

$$\begin{cases} \dfrac{\pi_t(\boldsymbol{\omega}^r)}{\pi_t(\boldsymbol{\omega}^s)} = \dfrac{\frac{\exp(\omega_1^r \cdot t + \omega_0^r)}{\sum_{j=1}^K \exp(\omega_1^j \cdot t + \omega_0^j)}}{\frac{\exp(\omega_1^s \cdot t + \omega_0^s)}{\sum_{j=1}^K \exp(\omega_1^j \cdot t + \omega_0^j)}} \\[3ex] \dfrac{\pi_t(\boldsymbol{\omega}^{\sigma(r)})}{\pi_t(\boldsymbol{\omega}^{\sigma(s)})} = \dfrac{\frac{\exp\left(\omega_1^{\sigma(r)} \cdot t + \omega_0^{\sigma(r)}\right)}{\sum_{j=1}^K \exp\left(\omega_1^{\sigma(j)} \cdot t + \omega_0^{\sigma(j)}\right)}}{\frac{\exp\left(\omega_1^{\sigma(s)} \cdot t + \omega_0^{\sigma(s)}\right)}{\sum_{j=1}^K \exp\left(\omega_1^{\sigma(j)} \cdot t + \omega_0^{\sigma(j)}\right)}} \end{cases}$$

By Equation 20:

$$\frac{\pi_t(\boldsymbol{\omega}^r)}{\pi_t(\boldsymbol{\omega}^s)} = \frac{\pi_t(\boldsymbol{\omega}^{\sigma(r)})}{\pi_t(\boldsymbol{\omega}^{\sigma(s)})}$$

$$\Leftrightarrow \exp\left[(\omega_1^r - \omega_1^s)\, t + (\omega_0^r - \omega_0^s)\right] = \exp\left[\left(\omega_1^{\sigma(r)} - \omega_1^{\sigma(s)}\right) t + \left(\omega_0^{\sigma(r)} - \omega_0^{\sigma(s)}\right)\right]$$

$$\Leftrightarrow \forall t \in \mathcal{T} : (\omega_1^r - \omega_1^s)\, t + (\omega_0^r - \omega_0^s) = \left(\omega_1^{\sigma(r)} - \omega_1^{\sigma(s)}\right) t + \left(\omega_0^{\sigma(r)} - \omega_0^{\sigma(s)}\right)$$

As a consequence of the last equation, we have for all the indices:

$$\begin{cases} \omega_1^r - \omega_1^s = \omega_1^{\sigma(r)} - \omega_1^{\sigma(s)} \\ \omega_0^r - \omega_0^s = \omega_0^{\sigma(r)} - \omega_0^{\sigma(s)} \end{cases}$$

$$\begin{cases} \omega_1^r - \omega_1^{\sigma(r)} = \omega_1^s - \omega_1^{\sigma(s)} = \Delta_1 \\ \omega_0^r - \omega_0^{\sigma(r)} = \omega_0^s - \omega_0^{\sigma(s)} = \Delta_0 \end{cases}$$

Hence it exists a translation function $\varrho : \mathbb{R}^2 \to \mathbb{R}^2$, such that $\forall r \in [|1 : K|]$:

$$\boldsymbol{\omega}^r = \varrho(\tilde{\boldsymbol{\omega}}^{\sigma(r)}).$$

Hence our mixture of temporal causal models is identifiable as defined in definition 4.1.

