# OpenReview forum: "Flow based approach for Dynamic Temporal Causal models with non-Gaussian or Heteroscedastic Noises"
_NeurIPS.cc/2025/Conference — NeurIPS 2025 poster_

### Official Review · Reviewer_FQMP · 2025-06-12

**Clarity:** 2
**Significance:** 4
**Originality:** 4
**Rating:** 5
**Confidence:** 3

**Summary:**

This paper introduces FANTOM, a Bayesian framework for causal discovery in multivariate time series that exhibit nonstationarity, regime shifts, and complex noise structures such as heteroscedasticity. Unlike existing methods that assume stationarity and homoscedastic Gaussian noise, FANTOM jointly learns the number of latent regimes, their temporal boundaries, and a separate causal graph (DAG) for each regime. It does so using a Bayesian EM algorithm that integrates conditional normalizing flows to model input-dependent noise. The authors establish theoretical identifiability guarantees under heteroscedastic (Gaussian and non Gaussian) noise settings and for mixtures of identifiable causal models, ensuring the recoverability of the true graph structure. Experimental results on synthetic and real-world data show that FANTOM consistently outperforms prior approaches, especially in challenging nonstationary or heteroscedastic conditions.

**Questions:**

1.	Initialization sensitivity and “regime-collapse” heuristics. The E-step begins with an Nw >K equal-window initialization and then prunes or merges regimes via heuristic rules (e.g., discarding windows with fewer than ζ points and re-assigning their samples). While the qualitative intuition is clear, the paper does not quantify how sensitive FANTOM’s performance is to the initial choice of Nw, window length, or the pruning threshold. Canyou provide insights on this?

2.	Clarification of ELBO derivation from Equation (2) to Equation (6) The derivation from the general ELBO expression in Equation (2) to the explicit M-step optimization objective in Equation (6) is not fully detailed in the main text. Several substitutions appear to be made (e.g., replacing the latent regime distribution p(zt∣xt,x<t) with soft assignments βtr, separating the graph prior and entropy terms, and incorporating the time-varying regime prior πt(ωr)but these steps are not explicitly shown. Can you provide details of derivation?

3.	Clarification on neural network design for fi,r and the functional constraint. In Section 3, the authors state that the structural function fi,r is designed to satisfy the condition ∂ fi,r / ∂xt−τ j= 0 and they propose a flexible neural architecture in Equation (4) to enforce this. However, the design and enforcement of this partial derivative constraint within the neural network is not clearly explained. Could the authors clarify how the architecture in Equation (4) ensures that with respect to a non-parent input is exactly zero?

**Ethical Concerns:**

["NO or VERY MINOR ethics concerns only"]

**Final Justification:**

This paper presents FANTOM, a novel Bayesian framework for causal discovery in multivariate time series with nonstationarity and heteroscedastic noise. The work is technically rigorous, integrates regime segmentation and graph learning in a unified probabilistic model, and is supported by identifiability guarantees and strong empirical results. While some aspects (e.g., initialization heuristics, clarity of derivations) could be improved, the rebuttal addressed my main concerns satisfactorily. Overall, I find this a solid and impactful contribution and recommend acceptance.

**Limitations:**

No, the authors have not adequately addressed the limitations of their work nor any potential societal impact. They did mention Theoretical guarantees (e.g., identifiability) are currently restricted to stationary settings, while empirical results are shown under nonstationary conditions. While the method is general-purpose, it would be valuable to include a short discussion on potential societal implications—especially when applying causal discovery to sensitive domains (e.g., public policy, healthcare, or financial modeling)—and how to responsibly interpret or validate discovered causal structures.

**Paper Formatting Concerns:**

i think it is fine

**Quality:**

3

**Strengths And Weaknesses:**

Strengths:

•	Quality: The paper demonstrates a high level of technical rigor, offering a unified probabilistic model that addresses both structure learning and temporal regime segmentation. The use of conditional normalizing flows to capture heteroscedasticity is well-justified, and the variational EM formulation is grounded in a clearly articulated ELBO. The proofs of identifiability under heteroscedastic noise and their extension to mixtures of regimes further strengthen the theoretical foundations.

•	Originality: The work is novel in its comprehensive treatment of causal discovery for multivariate time series under both nonstationary and heteroscedastic conditions. The idea of integrating regime inference, graph learning, and flow-based noise modeling in a single Bayesian framework is a substantial innovation compared to prior works that treat these steps separately or rely on simplistic assumptions.

•	Significance: The proposed method addresses a major limitation in the causal discovery literature—namely, its sensitivity to regime shifts and complex noise. This makes the contribution relevant to real-world domains such as finance, climate, and neuroscience, where such assumptions are frequently violated. The empirical results demonstrate clear gains over strong baselines.

•	Clarity: Overall, the paper is decently written and structured, with clear motivation and appropriate mathematical detail.

Weaknesses:
Clarity: the notations are not well explained. For example some aspects of the ELBO derivation from eqn (2) to (6) are not explained,  and the main Figure 2 can be more clearly explained- the terms in eqn 4.

Technical: 1) The initialization strategy and merging procedure in the E-step rely on heuristic pruning and regime collapse rules, which, while practical, introduce some ad hoc elements. The sensitivity of the results to this initialization (e.g., number of initial regimes $N_w$) is not deeply explored.

2)	While the identifiability results are a strong theoretical asset, they are derived under stationary settings. The implications for nonstationary time series—where regime transitions are latent and evolving—are not theoretically addressed and are instead treated empirically.

---

> ### Author Rebuttal · Authors · 2025-07-30
>
> We thank the reviewer for their valuable feedback and suggestions. We are pleased they find that our paper is novel, demonstrates a high level of rigor, and our choices are well-justified. The reviewer finds that our method addresses a major limitation in causal discovery and is supported by theoretical results for the single and multi regime settings. Below, we address their specific comments and suggestions. We hope that our responses are convincing.
>
> >Clarification of ELBO derivation from Equation (2) to Equation (6)
>
> We thank the reviewer for pointing out this question, we will add a section in our appendices in which we will clarify the mathematical derivation Eq.(2) to Eq.(6) and the intuition behind it.
>
> We perform the maximization of ELBO presented in proposition 3.1, using a BEM procedure, where we alternate between E-step (updating posterior probabilities while fixing all the parameters $\Theta = \\{\theta^r,\phi^r, \omega^r\\}\_{r=1}\^{N_w}$) and M step (Updating the parameters while using the posteriors learned in the E-step), we can summarize the process as follows:
>
> * In the Estep: we learn $\beta\_t^r = p(z_t|x_t,x_{<t}, \Theta\_{old})$, where $\Theta\_{old}$ is $\Theta$ of the previous iteration.
> * In the M-step: we fix the learned posterior probabilities to the values $\beta\_t^r $ and we update $\Theta$.
> * By fixing the posterior probabilities in the M-step, the entropy of these probabilities $H\left(p(z_t \mid x_t, x_{<t},\Theta_{\mathrm{old}})\right)$ is a constant, which allow us to discard it.
>
> The detail derivation from Eq.(2) to Eq.(6) is the following:
>
> $\\mathrm{ELBO}(\Theta) = \\sum_{t=1}^{|T|}\\mathbb{E}\_{q\_{\phi}(\mathcal{G})} \Big[\\mathbb{E}\_{p(z_t \mid x_t, x_{<t})}\big(\log p_{\theta_{z_t}}(x_t \mid x_{<t}, \mathcal{G}^{z_t})+ \log p(z_t) \big)+ H\big(p(z_t \mid x_t, x_{<t})\big)\Big]+ \sum\_{r=1}^{N_w}
> \\mathbb{E}\_{q\_{\phi_r}(\mathcal{G}^r)}\left[\log p(\mathcal{G}^r)\right]+ H\left(q_{\phi_r}(\mathcal{G}^r)\right)$
>
> $= \sum_{t=1}^{|T|} \\mathbb{E}\_{q\_{\phi}(\mathcal{G})}\Big[\sum_{z_t} p(z_t \mid x_t, x_{<t}, \Theta_{\mathrm{old}})\big(\log p_{\theta_{z_t}}(x_t \mid x_{<t}, \mathcal{G}^{z_t}) + \log p(z_t)\big) \Big] + \\underbrace{H\left(p(z_t \mid x_t, x_{<t},\Theta_{\mathrm{old}})\right)}_{\text{Cte}} + \\sum\_{r=1}^{N_w} \\mathbb{E}\_{q\_{\phi_r}(\mathcal{G}^r)}\left[\log p(\mathcal{G}^r)\right]+ H\left(q\_{\phi_r}(\mathcal{G}^r)\right)$
>
> We replace $p(z_t \mid x_t, x_{<t}, \Theta_{\mathrm{old}})$ by $\beta_t^{r}$, $p(z_t)$ by our prior choice $\pi_t(\omega^r)$, and we discard the entropy of the posterior because it is a constant in these steps. Hence, we got the Eq (6):
>
> $=\\mathbb{E}\_{q\_{\phi}(\mathcal{G})}\Big[\\sum\_{t=1}^{|T|}\sum_{r=1}^{N_w}\beta_t^r \big(\log p_{\theta_r}(x_t \mid x_{<t},\mathcal{G}^r) + \log \pi_t(\omega^r)\big)\Big]  + \sum_{r=1}^{N_w} \\mathbb{E}\_{q\_{\phi_r}(\mathcal{G}^r)}\left[\log p(\mathcal{G}^r)\right]+ H\left(q_{\phi_r}(\mathcal{G}^r)\right).$ Eq(6)
>
> After the maximization of Eq.(6), FANTOM update the posteriors $p(z_t \mid x_t, x_{<t}, \theta_{\mathrm{old}})$ following Eq.(7).
>
> >The initialization strategy and merging procedure in the E-step rely on heuristic pruning and regime collapse rules, which, while practical, introduce some ad hoc elements. The sensitivity of the results to this initialization (e.g., number of initial regimes Nw) is not deeply explored.
> While the qualitative intuition is clear, the paper does not quantify how sensitive FANTOM’s performance is to the initial choice of Nw, window length, or the pruning threshold. Can you provide insights on this?
>
> We thank the reviewer for this suggestion. FANTOM uses two hyperparameters: (i) the window initialization that gives rise to initial regimes, and (ii) ζ which cancels short regimes during the BEM training loop. The choice of window length determines the number of initial regimes, starting with small windows will give rise to a bigger number of regimes. For example, in the presence of two unknown groundtruth regimes [3000,2500] a window length = 1000 will give rise to **five initial regimes, hence N_w=5** [1000,1000,1000,1000,1500], then FANTOM will smoothly converges to the two ground-truth regimes.
>
> To quantify the sensitivity of FANTOM to the choice of window length and ζ, we conducted new experiments that shows the robustness of FANTOM towards the choices of these hyperparamerters. To show efficiency under long-horizon conditions, we used regimes with lengths [2000, 3000, 2000, 2500], and we also show that FANTOM can scale to more than 3 regimes with different initializations and for different number of nodes, the results are presented in these tables:
>
> **Table 1: Performance of FANTOM with varying hyperparameters (window size and ζ) on 10‑node graphs with 2, 3, and 4 regimes.**
> |                   |   |       |      |         | Heteroscedastic noise |      |     |      |        |
> |:-----------------:|---|:-----:|------|---------|:---------------------:|:----:|:---:|:----:|:------:|
> | (window, $\zeta$) | K | Nodes | Numb | Running |         Inst.         |      | Lag |      | Regime |
> |                   |   |       | iter | Time    |          SHD          |  F1  | SHD |  F1  |  Acc.  |
> |     (1000,900)    | 2 |   10  | 4    | 17'8s   |           5           | 91.2 |  13 | 85.7 |  96.5  |
> |    (1500,1200)    | 2 |   10  | 4    | 10'12s  |           1           | 97.9 |  9  | 89.8 |  95.1  |
> |     (1000,900)    | 3 |   10  | 4    | 30'33s  |           5           | 94.2 |  8  | 94.4 |  96.4  |
> |    (1500,1200)    | 3 |   10  | 4    | 16'2s   |           3           | 96.7 |  10 | 91.9 |  91.8  |
> |     (1000,900)    | 4 |   10  | 4    | 40'2s   |           5           | 95.8 |  12 | 92.7 |  92.9  |
> |    (1500,1200)    | 4 |   10  | 4    | 25'38s  |           3           | 97.5 |  16 | 90.5 |  91.5  |
>
> **Table 2: Performance of FANTOM with varying hyperparameters (window size and ζ) on 40‑node graphs with 2, 3, and 4 regimes.**
> |                   |   |       |      |         | Heteroscedastic noise |      |     |      |        |
> |:-----------------:|---|:-----:|------|---------|:---------------------:|:----:|:---:|:----:|:------:|
> | (window, $\zeta$) | K | Nodes | Numb | Running |         Inst.         |      | Lag |      | Regime |
> |                   |   |       | iter | Time    |          SHD          |  F1  | SHD |  F1  |  Acc.  |
> |     (1000,900)    | 2 |   40  | 4    | 22'15s  |           33          | 81.9 |  27 | 91.9 |   100  |
> |    (1500,1200)    | 2 |   40  | 4    | 10'20s  |           35          | 83.8 |  25 | 91.7 |   100  |
> |     (1000,900)    | 3 |   40  | 4    | 34'25s  |           49          | 85.6 |  58 | 88.6 |  99.8  |
> |    (1500,1200)    | 3 |   40  | 4    | 15'52s  |           43          | 87.3 |  72 | 87.3 |  99.9  |
> |     (1000,900)    | 4 |   40  | 5    | 66'30s  |           34          | 86.5 |  53 | 91.9 |  99.7  |
> |    (1500,1200)    | 4 |   40  | 4    | 21'38s  |           32          | 87.1 |  54 | 91.8 |  99.6  |
>
> From the table, FANTOM scales to 40 nodes and converges to four regimes under different initializations. **With initialization = 1500 and ζ =1200**, it starts with **six initial regimes** [1500,1500,1500,1500,1500,2000] and converges smoothly to four in 21 min 38 s. With **initialization = 1000 and ζ =900**, it starts with **nine initial regimes** [1000,1000,1000,1000,1000,1000,1000,1000,1500] and converges to four with 99.7% accuracy in 66 min 30 s. The choices of hyperparameters does not affect FANTOM performances, it only affects the runtime of the method. Also, in our real world experiment for **Epilepsy detection**, FANTOM is initialized with **eight initial regimes** and reliably converges to the **two ground-truth states [non seizure, seizure]**.
>
> >Clarification on neural network design for fi,r and the functional constraint.
>
> We thank the reviewer for this question. We propose flexible functional designs for $f^{i, r}$, which must respect the relations encapsulated in $\mathcal{G}^r$. Namely, if $x\_{t-\tau}^j \notin \\mathbf{P a}\_{\\mathcal{G}^r}^i(<t) \cup \\mathbf{Pa}\_{\mathcal{G}^r}^i(t)$, then $\partial f^{i, r} / \partial x_{t-\tau}^j=0$.
>
> We instantiate $f\_{\theta^r}^{i, r}$ as:
>
> $f\_{\theta^r}^{i, r}\left(\mathbf{P a}\_{\mathcal{G}^r}^i(<t), \mathbf{P a}\_{\mathcal{G}^r}^i(t)\right)=\psi^r\left(\sum\_{\tau=0}^L \sum\_{j=1}^d G\_{\tau, j i}^r \cdot \vartheta^r\left(x\_{t-\tau}^j, e\_{\tau, j}^r\right), e\_{0, i}^r\right)$,
>
> where $\theta^r$ includes the learnable parameters of the neural network $\vartheta^r$ and $\psi^r$ and the learnable embedding $e\_{\tau, j}^r$ and $e\_{0, j}^r$.
>
> $G\_{\tau, j i}^r$ encodes whether the edge $j \rightarrow i$ is present at lag $\tau$ in regime $r$. When $x\_{t-\tau}^j \notin \\mathbf{P a}\_{\\mathcal{G}^r}^i(<t) \cup \\mathbf{Pa}\_{\mathcal{G}^r}^i(t)$, we have $G\_{\tau, j i}^r =0$; consequently, variations in $x\_{t-\tau}^j$ do not impact the value of $f\_{\theta^r}^{i, r}$, hence $\partial f^{i, r} / \partial x_{t-\tau}^j=0$.
>
> >No, the authors have not adequately addressed the limitations of their work nor any potential societal impact.
>
> We thank the reviewer for bringing to our attention that we did not mention that a limitation section is introduced in our appendices. In our revised version, we will explicitly reference the Limitations section in the main text, and we will add a paragraph on societal impacts.

---

### Official Review · Reviewer_KqQF · 2025-06-23

**Clarity:** 3
**Significance:** 3
**Originality:** 3
**Rating:** 5
**Confidence:** 4

**Summary:**

The authors present a novel method for causal discovery in multivariate time series. It can cope with heteroscedastic models (for arbitrary noise distributions), lagged and instantaneous effects as well as stationarity and non-stationarity. The method is based on a Bayesian Expectation Maximization algorithm which learns both the different regimes of the time series and the underlying causal structure(s). Moreover, identifiability results, a detailed empirical comparison with further causal discovery methods, and real data examples are provided.

**Questions:**

-	Did you investigate whether the method is robust to scaling? E.g., standardizing the data, which is often a problem in likelihood-based methods (Reisach et al. *Beware of the simulated dag! causal discovery benchmarks may be easy to game*, 2021).
- In Eq. (9) (Appendix C) should small $k$ be $\tau$?
-	Can you elaborate a bit more on how you carried out the maximization of the ELBO in the M-step?
-	What about $L > 1$? Line 323 references Appendix F.1 for such cases, but I could not find results in that section.

**Ethical Concerns:**

["NO or VERY MINOR ethics concerns only"]

**Final Justification:**

Since the authors could answer my questions in a satisfactory manner, I increased my score.

**Limitations:**

Yes (except scaling; see questions).

**Quality:**

3

**Strengths And Weaknesses:**

Strengths:

-	The authors fill a gap in the literature developing a causal structure learning algorithm for time series that can handle (non-) stationary heteroscedastic and non-Gaussian settings.
-	The authors back up their method with convincing empirical (experiments) and theoretical results (identifiability, building upon proofs for the stationary setting).
-	A detailed explanation of the methodology is provided, including sufficient intuition, and supported by clear and illustrative figures.

Weaknesses (minor):

-	More details on how exactly the “joint” maximization of Eq. (6) in 3.3 is carried out could be incorporated.
-	Within the main text, a more thorough explanation of Theorem 4.3 would be appreciated.
-	Please explain why the entropy of $p(z_t|x_t,x_{<t})$ does not show up in Eq (6).
-	The writing can be improved. For instance, the authors alternate between “non-Gaussian”, “non Gaussian”, “log likelihood”, “log-likelihood” and “non-stationary” and “non stationary”, … There should be consistency in this regard. Also usage of “can not” and “definition H.1” instead of “Definition”, …
-	The placement of Table 2 should be changed to satisfy the formatting requirements.

---

> ### Author Rebuttal · Authors · 2025-07-30
>
> We thank the reviewer for the valuable feedback. The reviewer observes that the proposed method fills an important gap in the literature, is rigorously benchmarked, and delivers substantial empirical and theoretical contributions. They also acknowledge the clarity of the exposition, including intuitive explanations and illustrative figures. Below, we address their specific comments and questions. We hope that our responses are satisfactory so that the rating can yet be increased.
>
> >More details on how exactly the “joint” maximization of Eq. (6) in 3.3 is carried out could be incorporated.
>
> We thank the reviewer for the suggestion and the question about computing Eq. (6). We will add the following discussion to the revised manuscript: Eq. (6) splits into two **independent maximization problems**: (i) graph learning and (ii) regime alignment. The colors in Eq. (6) marks the terms belonging to each subproblem. **We solve these two maximizations separately.** We would like to mention that the regime‑alignment step is computationally light and runs in only a few seconds.
>
> >Can you elaborate a bit more on how you carried out the maximization of the ELBO in the M-step (Eq 6)?
>
> After estimating the posterior probabilities $\beta^r_t$ in the E‑step, we apply hard thresholding, which assigns each timestamp to a single regime and makes $\\beta^r_t$ binary ($\\forall t \in \\mathcal{T}: \\sum\_{r=1}^{N_w}\beta^r_t =1$ and $\\beta^r_t \\in \\{0,1\\}$). We then optimize the two objectives separately.
> * For regime alignment, we use SGD to learn the parameter $\omega^r$ in a softmax of an affine function of the time index.
> * For graph learning, we use an augmented Lagrangian to enforce the acyclicity prior and optimize with Adam, following Geffner et al. (2022) [1] and Gong et al. [2]. The procedure alternates between: (i) optimizing the graph objective with fixed penalty parameters $\rho$ and $\alpha$ (defined in our graph prior line 240 and 247) for about 2000 steps, and (ii) updating $\rho$ and $\alpha$. We repeat (i) (ii) until convergence or until hitting the maximum number of outer updates (typically 3).
>
> [1] Geffner, Tomas, et al. "Deep End-to-end Causal Inference." Transactions on Machine Learning Research.
>
> [2] Gong, Wenbo, et al. "Rhino: Deep Causal Temporal Relationship Learning with History-dependent Noise." The Eleventh International Conference on Learning Representations.
>
> >Within the main text, a more thorough explanation of Theorem 4.3 would be appreciated.
>
> We thank the reviewer for this constructive suggestion, which improves the quality of our paper. We will add the following explanation to the revised manuscript:
>
> *The theorem states that for identifiable temporal causal models, if we choose K linearly independent component models, then their mixture with time‑varying weights is identifiable, up to a permutation of the mixture components and up to a translation (additive shift) of the time‑varying weights. This establishes identifiability for mixtures with time‑varying weights and generalizes the standard case with static weights.*
>
> >Please explain why the entropy of $p(z_t|x_t,x_{<t})$ does not show up in Eq (6).
>
> We thank the reviewer for pointing out this question, we will add a section in our appendices in which we will clarify the mathematical derivation and the intuition behind it.
>
> We perform the maximization of ELBO presented in proposition 3.1, using a BEM procedure, where we alternate between E-step (updating posterior probabilities while fixing all the parameters $\Theta = \\{\theta^r,\phi^r, \omega^r\\}\_{r=1}\^{N_w}$) and M step (Updating the parameters while using the posteriors learned in the E-step), we can summarize the process as follows:
>
> * In the Estep: we learn $\beta\_t^r = p(z_t|x_t,x_{<t}, \Theta\_{old})$, where $\Theta\_{old}$ is $\Theta$ of the previous iteration.
> * In the M-step: we fix the learned posterior probabilities to the values $\beta\_t^r $ and we update $\Theta$.
> * By fixing the posterior probabilities in the M-step, the entropy of these probabilities is a constant, which allow us to discard it.
>
> The detail derivation from Eq.(2) to Eq.(6) is the following:
>
> $\\mathrm{ELBO}(\Theta) = \\sum_{t=1}^{|T|}\\mathbb{E}\_{q\_{\phi}(\mathcal{G})} \Big[\\mathbb{E}\_{p(z_t \mid x_t, x_{<t})}\big(\log p_{\theta_{z_t}}(x_t \mid x_{<t}, \mathcal{G}^{z_t})+ \log p(z_t) \big)+ H\big(p(z_t \mid x_t, x_{<t})\big)\Big]+ \sum\_{r=1}^{N_w}
> \\mathbb{E}\_{q\_{\phi_r}(\mathcal{G}^r)}\left[\log p(\mathcal{G}^r)\right]+ H\left(q_{\phi_r}(\mathcal{G}^r)\right)$
>
> $= \sum_{t=1}^{|T|} \\mathbb{E}\_{q\_{\phi}(\mathcal{G})}\Big[\sum_{z_t} p(z_t \mid x_t, x_{<t}, \Theta_{\mathrm{old}})\big(\log p_{\theta_{z_t}}(x_t \mid x_{<t}, \mathcal{G}^{z_t}) + \log p(z_t)\big) \Big] + \\underbrace{H\left(p(z_t \mid x_t, x_{<t},\Theta_{\mathrm{old}})\right)}_{\text{Cte}} + \\sum\_{r=1}^{N_w} \\mathbb{E}\_{q\_{\phi_r}(\mathcal{G}^r)}\left[\log p(\mathcal{G}^r)\right]+ H\left(q\_{\phi_r}(\mathcal{G}^r)\right)$
>
> We replace $p(z_t \mid x_t, x_{<t}, \Theta_{\mathrm{old}})$ by $\beta_t^{r}$, $p(z_t)$ by our prior choice $\pi_t(\omega^r)$, and we discard the entropy of the posterior because it is a constant in these steps. Hence, we got the Eq (6):
>
> $=\\mathbb{E}\_{q\_{\phi}(\mathcal{G})}\Big[\\sum\_{t=1}^{|T|}\sum_{r=1}^{N_w}\beta_t^r \big(\log p_{\theta_r}(x_t \mid x_{<t},\mathcal{G}^r) + \log \pi_t(\omega^r)\big)\Big]  + \sum_{r=1}^{N_w} \\mathbb{E}\_{q\_{\phi_r}(\mathcal{G}^r)}\left[\log p(\mathcal{G}^r)\right]+ H\left(q_{\phi_r}(\mathcal{G}^r)\right).$ Eq(6)
>
> After the maximization of Eq.(6), FANTOM update the posteriors $p(z_t \mid x_t, x_{<t}, \theta_{\mathrm{old}})$ following Eq.(7).
>
> >The writing can be improved. Format
>
> We thank the reviewer for their careful reading, their remarks and suggestions that enhance the quality of our paper. We will correct all these inconsistencies in our updated version. Regarding table 2 we will correct the format.
>
> >Did you investigate whether the method is robust to scaling? E.g., standardizing the data, which is often a problem in likelihood-based methods (Reisach et al. Beware of the simulated dag! causal discovery benchmarks may be easy to game, 2021).
>
> Thank you for raising this important and challenging question about causal discovery with optimization‑based models. We investigated it. **FANTOM is robust to data standardisation** because it uses **Bayesian structure learning** to estimate a distribution over plausible graphs, and **conditional normalizing flows** to model complex distributions. In contrast, other models like CASTOR are sensitive to standardization and fails even when we provide ground truth regime labels. The table below summarizes the results.
>
> **Table 1: Robustness of FANTOM to data standardization**
> | data         | K | model  | Inst SHD | Inst F1 | Lag SHD | Lag F1 | Reg Acc |
> |--------------|---|--------|----------|---------|---------|--------|---------|
> | Raw          | 2 | CASTOR with regime labels| 90       | 26.9    | 28      | 29.7   | x       |
> | standardized | 2 | CASTOR with regime labels| 28       | 0.0     | 33      | 0.0    | x       |
> | Raw          | 2 | FANTOM | 4        | 93.3    | 7       | 90.9   | 97.1    |
> | standardized | 2 | FANTOM | 5        | 91.5    | 7       | 90.9   | 91.1    |
> | Raw          | 3 | CASTOR with regime labels| 135      | 24.1    | 135     | 29.6   | x       |
> | standardized | 3 | CASTOR with regime labels| 37       | 0.0     | 48      | 0.0    | x       |
> | Raw          | 3 | FANTOM | 3        | 96.1    | 10      | 91.5   | 92.2    |
> | standardized | 3 | FANTOM | 2        | 97.3    | 9       | 92.3   | 92.2    |
>
> >In Eq. (9) (Appendix C) should small $k$ be $\tau$?
>
> We thank the reviewer for pointing out this typo. In Eq. (9), τ should replace k. We will correct this in the revised manuscript.
>
> >What about L>1? Line 323 references Appendix F.1 for such cases, but I could not find results in that section.
>
> We want to thank the reviewer for this remark, we acknowledge that due to the extensive experiments we conducted: *ablation studies of the hyperparameters, 6 tables for non-Gaussian and 6 tables for heteroscedastic settings.* We missed adding the results for L>1. We apologize for this oversight and will add them in the revised version.
>
> We also want to clarify that in real world scenarios we used FANTOM with L=8 for causal chambers and Epilepsy experiments. Here are the results on synthetic data with heteroscedastic noise and L=2:
>
> **Table 2: Performance of FANTOM on $𝐿=2$ time series under varying node counts $d$ 10,20,40 and regime settings $K$ 2 and 3.**
> | K | d  | Inst SHD | Inst F1 | Lag SHD | Lag F1 | Reg Acc |
> |---|----|-----------|----------|---------|--------|----------|
> | 2 | 10 | 1         | 97.7     | 0       | 100.0  | 94.3     |
> | 2 | 20 | 14        | 87.0     | 4       | 98.7   | 99.0     |
> | 2 | 40 | 15        | 94.5     | 3       | 99.5   | 98.9     |
> | 3 | 10 | 1         | 98.5     | 1       | 99.5   | 91.5     |
> | 3 | 20 | 5         | 97.5     | 11      | 97.4   | 97.9     |
> | 3 | 40 | 43        | 88.6     | 3       | 99.0   | 99.0     |
>
> FANTOM maintains similar performance with a larger lag ($L=2$). It achieves an F1 score above 87% on instantaneous links for 10, 20, and 40 nodes in MTS with 2 or 3 regimes. For time‑lagged links, we evaluate using the summary graph of lagged relations; the table shows that FANTOM detects them effectively.

---

> > ### Comment · Reviewer_KqQF · 2025-08-04
> >
> > Thank you for the clarifications. I updated my review.

---

> > > ### Author Response · Authors · 2025-08-04
> > > **Official Comment by the authors**
> > >
> > > Dear Reviewer,
> > >
> > > Thank you for taking the time to review our paper. We hope our rebuttal has addressed all of your concerns. Please let us know if any clarification is needed.
> > >
> > > We would also appreciate it if you could let us know whether you are inclined to support the acceptance of our paper.
> > >
> > > Kind regards,
> > >
> > > The Authors

---

### Official Review · Reviewer_4BQU · 2025-06-30

**Clarity:** 2
**Significance:** 2
**Originality:** 2
**Rating:** 5
**Confidence:** 3

**Summary:**

The paper introduces FANTOM, a causal discovery framework that handles multi-regime multivariate time series (MTS), homoscedastic non-Gaussian and heteroscedastic noise. FANTOM is capable of learning the regime indices even when its unknown. The authors also provide identifiability guarantees for both heterscedastic and homoscedastic noise settings. Given the MTS multi-regime data, FANTOM learns the regimes and the causal graph via Bayesian expectation maximization (BEM) optimizing evidence lower bound (ELBO).

**Questions:**

1. In assumption 2.2, the MTS system is assumed to be stationary for $\zeta$ consecutive steps. Is the $\zeta$ assumed to known apriori? Is the initial regime assignments based on the knowledge of $\zeta$?
2. How would the initialization of $N_w$ guaranteed to be greater than $K$ when $K$ unknown?
3. Given that FANTOM runs until a maximum number of iterations are reached (mentioned in line 176), how is the maximum number of iterations chosen?
4. How does FANTOM compare to the baseline (specifically CASTOR) in terms of computational cost? How much does modeling heteroscedastic and non-Gaussian noise affect the computational cost when compared to CASTOR?

If the authors can satisfactorily address the questions above, I would be inclined to raise my overall recommendation.

(Minor comment) It would help to define the function $H$ after introducing it in equation (2).

**Ethical Concerns:**

["NO or VERY MINOR ethics concerns only"]

**Final Justification:**

All the points I raised were addressed during the rebuttal.

**Limitations:**

Yes

**Quality:**

3

**Strengths And Weaknesses:**

Quality: The paper is technically sound, the main claims of the paper (causal discovery under MTS multi regime setting) is supported by the theoretical results established in section 4, and experimentally in section 5 where the proposed method shows improved performance over the baselines in certain settings.

Clarity: The presentation is sound. However, the paper is dense with technical details making it a bit challenging to read.

Significance: The paper addresses the difficult task of causal discovery from time series data multiple regime changes and heteroscedastic noise. The experimental results shows that the the proposed method, FANTOM, outperforms the baselines in the heteroscedastic setting, and in the real world data experiments. However, the proposed work assumes that the minimum regime width is assumed to known beforehand, which may not always be practical.

Originality: The proposed work extends the results of Rahmani and Frossard (2025) [1] (CASTOR), both follow a similar framework with FANTOM allowing for heteroscedastic noise as well as non-Gaussian noise compared to CASTOR.

[1] Rahmani, Abdellah, and Pascal Frossard. "Causal Temporal Regime Structure Learning".

---

> ### Author Rebuttal · Authors · 2025-07-30
>
> We thank the reviewer for their valuable feedback and interesting questions. We are pleased that they find our paper technically sound, supported by theoretical results, deem the task we address difficult and our performance superior to existing baselines. Below, we address their specific comments and questions. We hope that our responses are satisfactory so that the rating can yet be increased.
>
> >Originality: The proposed work extends the results of Rahmani and Frossard (2025) [1] (CASTOR)
>
> We thank the reviewer for their feedback, It s true that FANTOM and CASTOR uses the same time varying prior for the latent factors  as well as the initialization trick, however FANTOM first employs Conditional Normalizing flows (CNF) to handle complex noise distributions (heteroscedastic noises and Non-Gaussian noises). In such cases CASTOR fails to detect the regimes as well as learning the graphs (**see Table 1 in our paper**). Moreover FANTOM uses Bayesian structure learning instead of structure learning (used by CASTOR) and this with CNF helps solving **the problem of data standardization** faced by CASTOR, Dynotears and other models. We conducted new experiments to show this phenomena, here are the results:
>
> **Table 1: Robustness of FANTOM to data standardization**
> | data         | K | model  | Inst SHD | Inst F1 | Lag SHD | Lag F1 | Reg Acc |
> |--------------|---|--------|----------|---------|---------|--------|---------|
> | Raw          | 2 | CASTOR with regime labels| 90       | 26.9    | 28      | 29.7   | x       |
> | standardized | 2 | CASTOR with regime labels| 28       | 0.0     | 33      | 0.0    | x    |
> | Raw          | 2 | FANTOM | 4        | 93.3    | 7       | 90.9   | 97.1    |
> | standardized | 2 | FANTOM | 5        | 91.5    | 7       | 90.9   | 91.1    |
> | Raw          | 3 | CASTOR with regime labels| 135      | 24.1    | 135     | 29.6   | x       |
> | standardized | 3 | CASTOR with regime labels| 37       | 0.0     | 48      | 0.0    | x       |
> | Raw          | 3 | FANTOM | 3        | 96.1    | 10      | 91.5   | 92.2    |
> | standardized | 3 | FANTOM | 2        | 97.3    | 9       | 92.3   | 92.2    |
>
> From the Table, FANTOM achieves the same results on raw and standardized data while CASTOR fails in both cases even when we give it access to the ground truth regime labels. In the case of standardized data, CASTOR predicts adjacency matrices full of zeros due to its incapability of handling such scenarios. We also want to highlight that in real world scenarios where the noise distribution is complex CASTOR fails to learn meaningful graphs (In both our presented real world data sets CASTOR collapses to one regime).
>
> > In assumption 2.2, the MTS system is assumed to be stationary for ζ consecutive steps. Is the ζ assumed to known apriori? Is the initial regime assignments based on the knowledge of ζ?
>
> * **Minimum regime duration (ζ).** As in CASTOR, we treat ζ as a hyperparameter that sets the minimum length of a regime. When domain knowledge is available, it should guide this choice. For example, epileptic seizures typically last minutes to hours; setting ζ≈3 or 4 minutes will allow FANTOM to detect the meaningful regimes. In climate applications, seasons span months; setting ζ≈1 month is appropriate.
>
> * **Initialization and pruning.** FANTOM uses two hyperparameters: (i) the window initialization that gives rise to initial regimes, and (ii) ζ which cancels short regimes during the BEM training loop. We choose the window slightly larger than ζ to avoid early collapse to a single regime (e.g., window = 1000, ζ = 900 in our experiments).
>
> * **No domain knowledge available.** If no prior knowledge is available, taking a reasonable small (window, ζ) will help learning the meaningful regime smoothly. This choice will not affect the performance of FANTOM but it will affect running time.
>
> Here we conducted some ablation studies that show that FANTOM is robust to the (window size, ζ ) choices, the regime length are in this order [2000,3000,2000,2500], we tried to choose long horizons in order to show that FANTOM is scalable to long horizons regardless of the choice of the hyper parameters:
>
> **Table 2: Performance of FANTOM with varying hyperparameters (window size and ζ) on 10‑node graphs with 2, 3, and 4 regimes.**
> |                   |   |       |      |         | Heteroscedastic noise |      |     |      |        |
> |:-----------------:|---|:-----:|------|---------|:---------------------:|:----:|:---:|:----:|:------:|
> | (window, $\zeta$) | K | Nodes | Number | Running |         Inst.         |      | Lag |      | Regime |
> |                   |   |       | of iter | Time    |          SHD          |  F1  | SHD |  F1  |  Acc.  |
> |     (1000,900)    | 2 |   10  | 4    | 17'8s   |           5           | 91.2 |  13 | 85.7 |  96.5  |
> |    (1500,1200)    | 2 |   10  | 4    | 10'12s  |           1           | 97.9 |  9  | 89.8 |  95.1  |
> |     (1000,900)    | 3 |   10  | 4    | 30'33s  |           5           | 94.2 |  8  | 94.4 |  96.4  |
> |    (1500,1200)    | 3 |   10  | 4    | 16'2s   |           3           | 96.7 |  10 | 91.9 |  91.8  |
> |     (1000,900)    | 4 |   10  | 4    | 40'2s   |           5           | 95.8 |  12 | 92.7 |  92.9  |
> |    (1500,1200)    | 4 |   10  | 4    | 25'38s  |           3           | 97.5 |  16 | 90.5 |  91.5  |
>
> **Table 3: Performance of FANTOM with varying hyperparameters (window size and ζ) on 40‑node graphs with 2, 3, and 4 regimes.**
>
> |                   |   |       |      |         | Heteroscedastic noise |      |     |      |        |
> |:-----------------:|---|:-----:|------|---------|:---------------------:|:----:|:---:|:----:|:------:|
> | (window, $\zeta$) | K | Nodes | Number | Running |         Inst.         |      | Lag |      | Regime |
> |                   |   |       | of iter | Time    |          SHD          |  F1  | SHD |  F1  |  Acc.  |
> |     (1000,900)    | 2 |   40  | 4    | 22'15s  |           33          | 81.9 |  27 | 91.9 |   100  |
> |    (1500,1200)    | 2 |   40  | 4    | 10'20s  |           35          | 83.8 |  25 | 91.7 |   100  |
> |     (1000,900)    | 3 |   40  | 4    | 34'25s  |           49          | 85.6 |  58 | 88.6 |  99.8  |
> |    (1500,1200)    | 3 |   40  | 4    | 15'52s  |           43          | 87.3 |  72 | 87.3 |  99.9  |
> |     (1000,900)    | 4 |   40  | 5    | 66'30s  |           34          | 86.5 |  53 | 91.9 |  99.7  |
> |    (1500,1200)    | 4 |   40  | 4    | 21'38s  |           32          | 87.1 |  54 | 91.8 |  99.6  |
>
> In the **four‑regime setting** with ζ=900 and window = 1000, FANTOM initializes with **nine regimes: [1000,1000,1000,1000,1000,1000,1000,1000,1500]** and converges smoothly to the **four ground‑truth regimes: [2000,3000,2000,2500], with 99.7%**, this configuration requires 66 min 30 s. With initialization = 1500 and ζ =1200, it starts with **six initial regimes: [1500, 1500, 1500, 1500, 1500, 2000]** and converges smoothly to **four: [2000,3000,2000,2500], with 99.6%** accuracy in 21 min 38 s. Overall, Regime and Graph learning performances has not been affected by hyperparameters choices.
>
> >How would the initialization of Nw guaranteed to be greater than K when K unknown?
>
> We thank the reviewer for the question. As discussed earlier, our method has two hyperparameters: the window size and ζ. By choosing them carefully, you can ensure $N_w \geq K$. In practice, set (window,ζ) as small as possible while keeping enough data per window will both satisfy $N_w \geq K$ and help FANTOM converge reliably to the true regimes. In our earlier ablations, any (window,ζ) smaller than (2000,1900) and larger than ~(800,700) will work, since the Temporal GNN and CNF need few data to learn the graph effectively. We also showed that initialization does not affect FANTOM’s performance; it only changes runtime (see the previous table 2 and 3).
>
> >Given that FANTOM runs until a maximum number of iterations are reached (mentioned in line 176), how is the maximum number of iterations chosen?
>
> The number of iterations is a hyperparameter. In general, increasing it does not degrade FANTOM’s performance: once FANTOM reaches the optimum, it stabilizes, and additional iterations leave the solution unchanged.
>
> >How does FANTOM compare to the baseline (specifically CASTOR) in terms of computational cost? How much does modeling heteroscedastic and non-Gaussian noise affect the computational cost when compared to CASTOR?
>
> We thank the reviewer for the suggestion. We ran additional experiments comparing the computational cost of CASTOR and FANTOM in the case of heteroscedastic noise. We set the regime lengths to [2000, 3000, 2000], with ζ = 1200, window = 1500, and d = 10. The results are as follows:
>
> | K=3, d=10 |         |
> |-----------|---------|
> | model     | runtime |
> | CASTOR    | 14'59   |
> | FANTOM    | 16'2s   |
>
> CASTOR runs ~2 minutes faster than FANTOM, but it collapses to a single regime at the first iteration which makes it model the MTS as only one regime and reduces the computational cost in the next iterations. Moreover, CASTOR cannot learn the ground-truth regimes and graphs in heteroscedastic settings, where FANTOM performs significantly better.
>
> In terms of memory: All results in the paper were obtained on an NVIDIA GTX TITAN with 12 GB RAM, indicating that FANTOM has modest memory requirements.
>
> >Reviewer comment: (Minor comment) It would help to define the function H after introducing it in equation (2).
>
> We thank the reviewer for their comment. We will introduce the entropy H with its mathematical formulation in our updated version.

---

> > ### Comment · Reviewer_4BQU · 2025-08-01
> >
> > Thank you for your response and clarifications. I have raised my rating.

---

> > > ### Author Response · Authors · 2025-08-02
> > > **Official comment by the Authors**
> > >
> > > We’re delighted that our rebuttal addressed your concerns and that you’ve raised your rating. Thank you for supporting the acceptance of our work.
> > >
> > > Kind regards,
> > >
> > > The Authors

---

### Official Review · Reviewer_yhgx · 2025-07-02

**Clarity:** 3
**Significance:** 2
**Originality:** 3
**Rating:** 4
**Confidence:** 3

**Summary:**

The paper presents FANTOM, a method for causal discovery in multivariate time series that may switch among an unknown number of regimes and exhibit non-Gaussian or heteroscedastic noise. FANTOM combines a Bayesian EM procedure with conditional normalising flows. The authors prove identifiability for both single-regime and multi-regime settings. Experiments on synthetic data and two real-world benchmarks show that FANTOM recovers regime boundaries and regime-specific causal graphs more accurately than several existing methods.

**Questions:**

Can the authors provide additional information mentioned in weakness?

**Ethical Concerns:**

["NO or VERY MINOR ethics concerns only"]

**Final Justification:**

During the rebuttal, most of the concerns were resolved. Although I still have concerns about the risk of spurious causality, especially in real-world scenarios, I will keep my positive comment on this work.

**Limitations:**

- Scalability / Efficiency – report wall-clock training time, GPU hours, and memory for the 40-variable experiment, and discuss how cost grows with more variables, longer horizons, or >3 regimes.
- Hyper-parameter sensitivity – show how results change with the initial regime count, pruning threshold, and normalising-flow depth; document failure cases (e.g., over-pruning).
- Risk of spurious causality – recommend practical safeguards (expert vetting, interventional checks) before acting on the learned graphs, especially in medical or financial domains.
- Privacy & misuse – clarify whether the method can train on anonymised / differentially-private data and outline precautions when applied to sensitive EEG or industrial logs.

**Quality:**

3

**Strengths And Weaknesses:**

## Strengths
- Rigorous methodology. The Bayesian EM routine cleanly separates regime assignment (E-step) and DAG/noise learning (M-step), with conditional normalising flows providing flexible likelihoods.
- Provable guarantees. The paper supplies identifiability theorems for both single- and multi-regime settings, lending theoretical credibility to the approach.
- Provable guarantees. The paper supplies identifiability theorems for both single- and multi-regime settings, lending theoretical credibility to the approach.

## Weaknesses
- Ablation gaps. There is no study isolating the importance of heteroscedastic modelling, flow architecture, or the pruning step, so robustness remains uncertain.
- Compute cost unreported. Training time and GPU usage for the flow-based M-step are omitted, making efficiency hard to judge.

---

> ### Author Rebuttal · Authors · 2025-07-30
>
> We thank the reviewer for their valuable feedback and suggestions. We are pleased they find our paper rigorous and supported by theoretical results for the single and multi regime settings. Below, we address their specific comments and suggestions. We hope that our responses are satisfactory so that the rating can yet be increased.
>
> >Ablation gaps. There is no study isolating the importance of heteroscedastic modelling, flow architecture, or the pruning step, so robustness remains uncertain.
>
> We thank the reviewer for the helpful suggestions. To assess the contribution of heteroscedastic modelling and the flow architecture, we ran an ablation on MTS comparing three FANTOM variants: (i) Gaussian output (no flow, FANTOM-Gauss), (ii) a simple coupling‑spline flow (FANTOM-Spline), and (iii) the full model with a conditional normalizing flow (CNF, FANTOM). In the single‑regime setting, removing or simplifying the CNF consistently degrades performance. In the multi‑regime setting, only the CNF variant remains stable and converges to the two ground truth regimes; the alternatives collapse to one regime. Results are reported in the table below:
>
> **Table 1: Importance of heteroscedastic modelling and flow architecture. - means the method collapses to 1 regime.**
>
> |   |               |      | Heteroscedastic |     |      |          |
> |:-:|---------------|------|:---------------:|:---:|:----:|----------|
> |   |               | Inst |                 | Lag |      | Regime   |
> | K | model         | SHD  | F1              | SHD | F1   | Accuracy |
> | 1 | FANTOM-Gauss  | 39   | 0.0             | 44  | 42.5 | only 1 regime        |
> |   | FANTOM-Spline | 17   | 0.0             | 18  | 0    | only 1 regime         |
> |   | FANTOM        | 0    | 100             | 0   | 100  | only 1 regime        |
> | 2 | FANTOM-Gauss  | -    | -               | -   | -    | -        |
> |   | FANTOM-Spline | -    | -               | -   | -    | -        |
> |   | FANTOM        | 1    | 97.8            | 1   | 98.9 | 98.6     |
>
> We evaluated robustness to the pruning step, minimum regime duration (ζ), and regime initialization. Across these variations, FANTOM’s predictive performance remains stable. We find that changing (window size, ζ) primarily affects runtime, especially at long horizons, without materially impacting graph and regime learning performances. To show efficiency under long-horizon conditions, we used regimes with lengths [2000, 3000, 2000, 2500].
>
> **Table 2: Performance of FANTOM with varying hyperparameters (window size and ζ) on 10‑node graphs with 2, 3, and 4 regimes.**
> |                   |   |       |      |         | Heteroscedastic noise |      |     |      |        |
> |:-----------------:|---|:-----:|------|---------|:---------------------:|:----:|:---:|:----:|:------:|
> | (window, $\zeta$) | K | Nodes | Number | Running |         Inst.         |      | Lag |      | Regime |
> |                   |   |       | of iter | Time    |          SHD          |  F1  | SHD |  F1  |  Acc.  |
> |     (1000,900)    | 2 |   10  | 4    | 17'8s   |           5           | 91.2 |  13 | 85.7 |  96.5  |
> |    (1500,1200)    | 2 |   10  | 4    | 10'12s  |           1           | 97.9 |  9  | 89.8 |  95.1  |
> |     (1000,900)    | 3 |   10  | 4    | 30'33s  |           5           | 94.2 |  8  | 94.4 |  96.4  |
> |    (1500,1200)    | 3 |   10  | 4    | 16'2s   |           3           | 96.7 |  10 | 91.9 |  91.8  |
> |     (1000,900)    | 4 |   10  | 4    | 40'2s   |           5           | 95.8 |  12 | 92.7 |  92.9  |
> |    (1500,1200)    | 4 |   10  | 4    | 25'38s  |           3           | 97.5 |  16 | 90.5 |  91.5  |
>
> **Table 3: Performance of FANTOM with varying hyperparameters (window size and ζ) on 40‑node graphs with 2, 3, and 4 regimes.**
> |                   |   |       |      |         | Heteroscedastic noise |      |     |      |        |
> |:-----------------:|---|:-----:|------|---------|:---------------------:|:----:|:---:|:----:|:------:|
> | (window, $\zeta$) | K | Nodes | Number | Running |         Inst.         |      | Lag |      | Regime |
> |                   |   |       | of iter | Time    |          SHD          |  F1  | SHD |  F1  |  Acc.  |
> |     (1000,900)    | 2 |   40  | 4    | 22'15s  |           33          | 81.9 |  27 | 91.9 |   100  |
> |    (1500,1200)    | 2 |   40  | 4    | 10'20s  |           35          | 83.8 |  25 | 91.7 |   100  |
> |     (1000,900)    | 3 |   40  | 4    | 34'25s  |           49          | 85.6 |  58 | 88.6 |  99.8  |
> |    (1500,1200)    | 3 |   40  | 4    | 15'52s  |           43          | 87.3 |  72 | 87.3 |  99.9  |
> |     (1000,900)    | 4 |   40  | 5    | 66'30s  |           34          | 86.5 |  53 | 91.9 |  99.7  |
> |    (1500,1200)    | 4 |   40  | 4    | 21'38s  |           32          | 87.1 |  54 | 91.8 |  99.6  |
>
> From the table, FANTOM scales to 40 nodes and converges to four regimes under different initializations. With initialization = 1500 and ζ =1200, it starts with **six initial regimes** and converges smoothly to **four with 99.6% accuracy** in 21 min 38 s. With initialization = 1000 and ζ =900, it starts with **nine initial regimes** and converges to **four with 99.7% accuracy** in 66 min 30 s. All rebuttal experiments were run on a Tesla V100‑SXM2 (26 GB), whereas the results in the paper were obtained on an NVIDIA GTX TITAN (12 GB). These results indicate that FANTOM scales without requiring large GPU memory.
>
> >Scalability / Efficiency – report wall-clock training time, GPU hours, and memory for the 40-variable experiment, and discuss how cost grows with more variables, longer horizons, or >3 regimes.
>
> **Table 4: Runtime for different number of nodes and under different number of regimes.**
>
> |       |      |         | (window, $\zeta$)=(1500,1200) |         |      |         |
> |:-----:|------|---------|:-----------:|:-------:|:----:|:-------:|
> |       | K=2  |         | K=3         |         | K=4  |         |
> | Nodes | Number | Running | Number | Running | Number | Running |
> |       | of iter | Time    | of iter        | Time    | of iter | Time    |
> | 10    | 4    | 10'12s  | 4           | 16'2s   | 5    | 25'38s  |
> | 20    | 4    | 9'50s   | 4           | 15'3s   | 5    | 21'10s  |
> |   40  | 4    | 10'20s  | 4           | 15'52   | 5    | 21'38s  |
>
> We thank the reviewer for these suggestions. To test scalability, we ran FANTOM on a **40-node dataset with four regimes and long horizons** (lengths 2000, 3000, 2000, 2500). As expected, additional regimes and longer horizons increase runtime. With **40 nodes and two regimes**, training finishes in **10 min 20 s**, whereas the **four-regime setting requires 21 min 38 s**.
>
> >Hyper-parameter sensitivity – show how results change with the initial regime count, pruning threshold, and normalising-flow depth; document failure cases (e.g., over-pruning).
>
> We thank the reviewer for suggesting enriching experiments. We showed in the table above (Table 2 and 3) that FANTOM is robust to the initialization parameter (**window size that impacts directly the initial regime count and $\zeta$ the pruning threshold**), e.g. in the case of **40 nodes with 4 regimes**, with window initialization = 1500 and ζ =1200, it starts with **six initial regimes** and converges smoothly to four in 21 min 38 s. With initialization = 1000 and ζ =900, it starts with **nine initial regimes** and converges to four with 99.6% accuracy in 66 min 30 s.
>
> We also conducted new experiments to explore the sensitivity of FANTOM to normalizing flow depth. We pick K=2 regimes and d=10 nodes.
>
> **Table 4: FANTOM performance with different normalizing flow depth.**
>
> | K=2, d=10               |      | Heteroscedastic |     |      |          |
> |-------------------------|------|:---------------:|:---:|:----:|----------|
> |                         | Inst |                 | Lag |      | Regime   |
> | Normalizing flows depth | SHD  | F1              | SHD | F1   | Accuracy |
> | 16                      | 0    | 100             | 1   | 98.9 | 97.1     |
> | 32                      | 0    | 100             | 0   | 100  | 96.6     |
> | 64                      | 0    | 100             | 1   | 98.9 | 94.8     |
> | 128                     | 0    | 100             | 0   | 100  | 97.2     |
>
> FANTOM shows robustness to the choice of the depth of NF, regardless of the number of bins, it achieves the same range of performances.
>
> Regarding over pruning, we want to thank you for this suggestion, we will add a discussion in our manuscript, where we will clarify that over pruning will affect the performance of FANTOM.
>
> >Risk of spurious causality – recommend practical safeguards (expert vetting, interventional checks) before acting on the learned graphs, especially in medical or financial domains.
>
> We thank the reviewer for this suggestion. We will add the following paragraph to our updated version:
> Spurious causality is a practical risk, so we advocate using the learned graphs as decision support rather than as ground truth. Before acting on them, especially in medical or financial settings, practitioners should adopt safeguards: (i) expert vetting of proposed edges and mechanisms; (ii) robustness checks (e.g., stability under resampling or perturbations, alternative specifications); and (iii) interventional validation when feasible.
>
> >Privacy & misuse – clarify whether the method can train on anonymised / differentially-private data and outline precautions when applied to sensitive EEG or industrial logs.
>
> We will add a paragraph to our updated paper to clarify that our method trains on anonymised data and clarifies the precautions when applied to EEG data.

---

> > ### Comment · Reviewer_yhgx · 2025-08-02
> >
> > Thank you for your response. I will keep my score.

---

> > > ### Author Response · Authors · 2025-08-02
> > > **Official Comment by the authors**
> > >
> > > Dear Reviewer yhgx,
> > >
> > > Thank you for taking the time to review our paper.
> > >
> > > In our rebuttal, we addressed all concerns in your review and ran all the new experiments you suggested. We show that FANTOM is robust to 𝜁, window initialization, and the number of initial regimes. We also demonstrate the importance of flow modeling and heteroscedastic modeling. In addition, we show that FANTOM scales beyond three regimes with 40 nodes, report wall-clock training times for all experiments, and verify robustness to the choice of normalizing flow depth.
> > >
> > > Could you please point us to any remaining limitations you see? We would be happy to address them during the discussion period.
> > >
> > > Best regards,
> > >
> > > The Authors

---

> > > > ### Author Response · Authors · 2025-08-05
> > > > **Official Comment by the authors**
> > > >
> > > > Dear Reviewer yhgx,
> > > >
> > > > Thank you for taking the time to review our paper.
> > > >
> > > > In our rebuttal, we addressed all concerns in your review and ran all the new experiments you suggested.
> > > >
> > > > Could you please point us to any remaining limitations you see? We would be happy to address them in the extended discussion period.
> > > >
> > > > Kind regards,
> > > >
> > > > The Authors

---

### Decision · Program_Chairs · 2025-09-17

**Decision:**

Accept (poster)

**Comment:**

The authors propose a causal discovery framework in the realistic setting of non-stationary, multivariate time series data. Overall, the contribution is appreciated by the reviewers and they all recommend acceptance of the paper.  The novelty, experiments, and the presentation were all found to be adequate.